# Prion protein conversion at two distinct cellular sites precedes fibrillisation

Juan Manuel Ribes [1,3], Mitali P. Patel [1,3], Hazim A. Halim [1], Antonio Berretta[1], Sharon A. Tooze [2] & Peter-Christian Klöhn [1]✉

The self-templating nature of prions plays a central role in prion pathogenesis and is associated with infectivity and transmissibility. Since propagation of proteopathic seeds has now been acknowledged a principal pathogenic process in many types of dementia, more insight into the molecular mechanism of prion replication is vital to delineate specific and common disease pathways. By employing highly discriminatory anti-PrP antibodies and conversion-tolerant PrP chimera, we here report that de novo PrP conversion and formation of fibril-like PrP aggregates are distinct in mechanistic and kinetic terms. De novo PrP conversion occurs within minutes after infection at two subcellular locations, while fibril-like PrP aggregates are formed exclusively at the plasma membrane, hours after infection. Phenotypically distinct pools of abnormal PrP at perinuclear sites and the plasma membrane show differences in N-terminal processing, aggregation state and fibril formation and are linked by exocytic transport via synaptic and large-dense core vesicles.

The "protein only hypothesis"[1] proposes that an abnormal conformer of PrP causes self-templating protein aggregation by converting host-encoded cellular prion protein (PrP$^c$)[2–4], but the cellular mechanism of conversion has remained elusive. A major roadblock for the study of self-templating protein aggregation in prion diseases is the lack of anti-PrP antibodies that discriminate PrP$^c$ from abnormal PrP, here termed disease-associated PrP (PrP$^d$). Since the vast majority of anti-PrP antibodies characterised to date are non-discriminatory or "pan" antibodies[5], reliance on the enzymatic or chemical removal of PrP$^c$ has greatly limited progress in the investigation of PrP conversion.

Pivotal questions concerning the biology of PrP conversion have remained a matter of controversy. Firstly, the cellular site of prion replication remains unsolved and putative sites include the endocytic recycling pathway[6,7], the endosomal-lysosomal pathway[8–10], the Golgi apparatus[11,12] and the plasma membrane[13,14]. Secondly, the kinetics of de novo PrP conversion, a comparatively unexplored question in prion biology, is a matter of debate. Evidence that PrP converts within minutes after infection[13] has been disputed by others[15], who claim that proteinase K-resistant PrP (PrP$^{Sc}$) is not detectable before 24 h after infection, raising the question whether

distinct oligomeric states of abnormal PrP aggregates may explain these inconsistencies. Thirdly, early studies of prion-infected mice showed evidence of "prion-amyloid filaments" in the extracellular space beneath the ependyma[16] and recent studies characterised ex vivo isolated prion rods[17,18], but the formation and elongation of prion fibrils has not been recapitulated in neuronal cells. A recent study reported the detection of lipid raft-associated amyloid strings and webs of PrP, but the molecular mode of amyloid formation is unknown[14].

To address these current challenges, we identified highly discriminatory anti-PrP antibodies and conversion-tolerant, tagged *Prnp* chimera to examine aggregation phenomena in freshly and persistently prion-infected neuronal cells.

We provide evidence that (i) PrP conversion occurs within minutes after infection at two distinct cellular sites and precedes the formation of abnormal PrP conformers, (ii) fibril-like PrP$^d$ aggregates are formed and elongated at the plasma membrane, while PrP$^d$ fibrils are absent intracellularly, (iii) the prion infectious state is dependent on functional dynamins and Cdc42, (iv) PrP$^d$ segregates into synaptic and large-dense core vesicles of the regulated secretory pathway.

[1]Medical Research Council Prion Unit at UCL, Institute of Prion Diseases, University College London, London W1W 7FF, UK. [2]Molecular Cell Biology of Autophagy Laboratory, the Francis Crick Institute, London NW1 1BF, UK. [3]These authors contributed equally: Juan Manuel Ribes, Mitali P. Patel. ✉e-mail: p.kloehn@prion.ucl.ac.uk

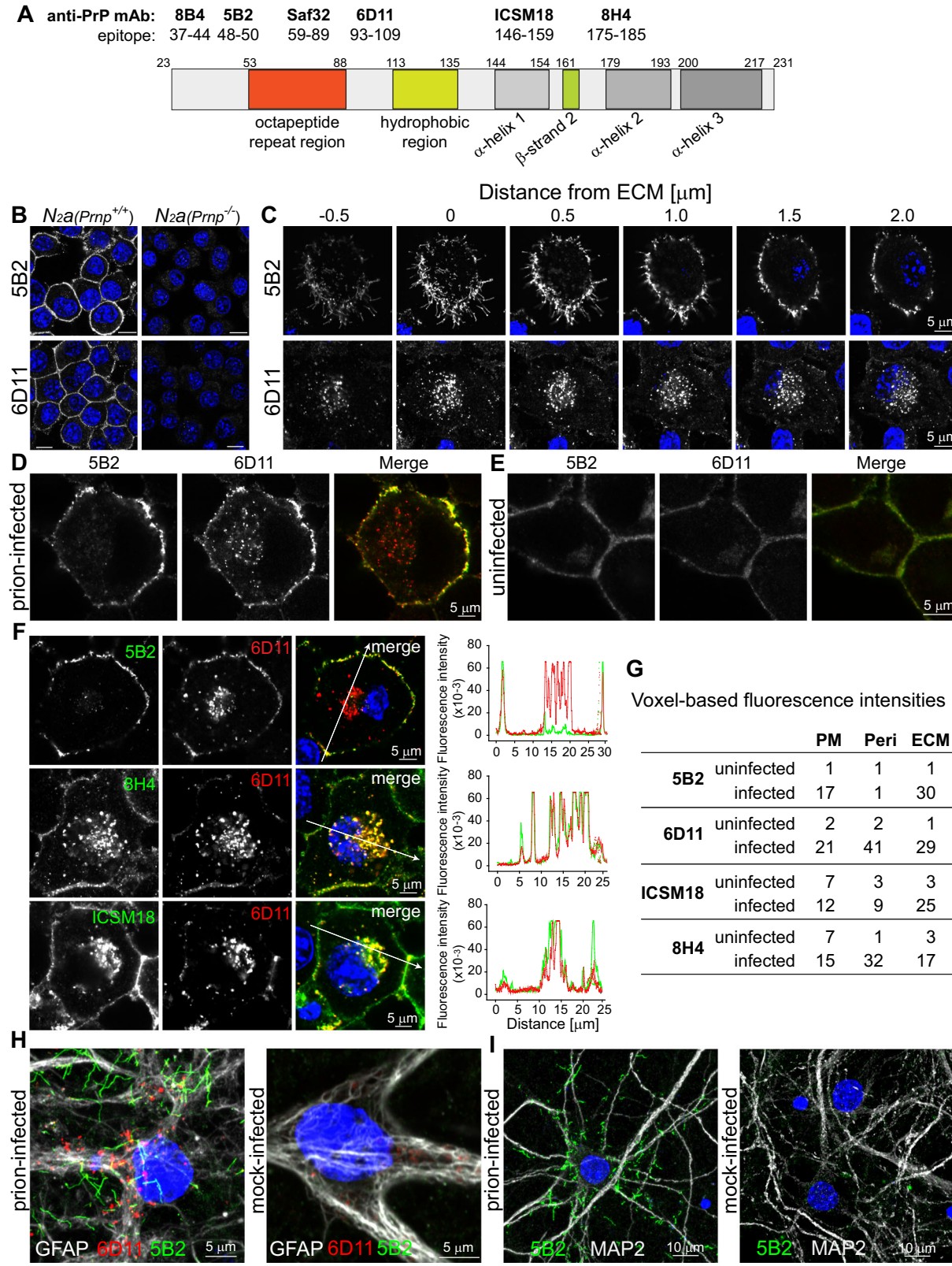

**G**

Voxel-based fluorescence intensities

|          |             | PM | Peri | ECM |
|----------|-------------|----|------|-----|
| **5B2**  | uninfected  | 1  | 1    | 1   |
|          | infected    | 17 | 1    | 30  |
| **6D11** | uninfected  | 2  | 2    | 1   |
|          | infected    | 21 | 41   | 29  |
| **ICSM18** | uninfected | 7  | 3    | 3   |
|          | infected    | 12 | 9    | 25  |
| **8H4**  | uninfected  | 7  | 1    | 3   |
|          | infected    | 15 | 32   | 17  |

## Results

### Phenotypically distinct PrP^d aggregates in prion-infected cells identified by monospecific anti-PrP antibodies

To identify highly validated anti-PrP antibodies, we characterised 18 frequently used monoclonal antibodies (mAbs) with respect to their target-specificity and propensity to discriminate PrP$^d$ from PrP$^c$ (Fig. 1A). We deleted *Prnp* in N2a cells to unequivocally demonstrate that mAbs are target-specific as shown for example for 5B2 and 6D11 (Fig. 1B). Seven mAbs were target-specific, also termed monospecific[19], while 4 mAbs showed moderate and 7 mAbs showed significant off-target effects (Supplementary Fig. 1A and Supplementary Table 1). Validated mAbs used in this study recognise distinct PrP regions which include the N-terminus (5B2, 8B4), the octapeptide repeat region (Saf32), charge cluster 2 (6D11), α-helix 1 (ICSM 18) and α-helix 2 (8H4; Fig. 1A).

**Fig. 1 | Identification of phenotypically distinct PrP$^d$ aggregates in prion-infected cells using monospecific anti-PrP antibodies. A** Schematic diagram of validated anti-PrP monoclonal antibodies (mAbs) and their putative epitopes on the context of mouse PrP domains[76]. **B** Exemplary data for anti-PrP mAb validation using N2a (Prnp$^{-/-}$) wild-type (Prnp$^{+/+}$) cells. For the full data set of anti-PrP mAbs tested see Supplementary Fig. 1A. Scale bars correspond to 10 μm. **C** Serial confocal sections of prion-infected S7 cells, labelled with anti-PrP mAbs 5B2 and 6D11, respectively. Cells were grown for extended cultures times, i.e. 6 days (see "Methods" section, "extended TC protocol"). **D, E** Co-labelling of prion-infected (**D**) and uninfected (**E**) S7 cells with 5B2 and 6D11. **F** Detection of perinuclear PrP$^d$ by core, but not by N-terminal anti-PrP mAbs in prion-infected S7 cells. Arrows in merged images depict orientation and placement of fluorescence intensity profiles, shown at the right hand side of image panels. For complete data set with all monospecific mAbs see Supplementary Fig. 2. **G** MAb-dependent differences in the detection of PrP$^d$ versus PrP$^c$, mapped as normalised voxel-based intensities in cellular loci where PrP$^d$ deposits (PM plasma membrane, Peri perinuclear region, ECM extracellular matrix). For data see Supplementary Table 2. **H** Glial fibrillary acidic protein (GFAP)-positive astrocytes in cultures of primary neuronal cells from embryonic e17 mouse brains, infected with 10$^{-5}$ dilutions of the prion strain RML (10% brain homogenate, w/v) and uninfected CD1 (10% brain homogenate, w/v, mock), triple-labelled with 5B2, 6D11 and anti-GFAP. **I** Primary hippocampal neuronal cultures from embryonic e17 mouse brains were isolated, cultured for 6 days and incubated with 1 μM AraC. The following day, cells were infected with a 10$^{-5}$ dilution of RML (10%, w/v) or uninfected CD1 (10%, w/v, mock-infected) brain homogenate, respectively. Three weeks after infection, cells were fixed and double-labelled with 5B2 and anti-Map2. Source data are provided as a Source Data file.

Unexpectedly, monospecific mAbs showed phenotypic differences in single-labelled prion-infected cells of the same culture (Supplementary Fig. 1B-E). Fibril-like PrP$^d$ aggregates were detected with 5B2 at the extracellular matrix (ECM) in infected cells (Supplementary Fig. 1B), while 6D11 labelling revealed a punctate staining predominantly at perinuclear regions (Supplementary Fig. 1C). ICSM18 detected PrP$^d$ aggregates at the ECM (Supplementary Fig. 1D) and 8H4 labelled aggregates were reminiscent of the punctate 6D11 phenotype (Supplementary Fig. 1E). This finding prompted us to investigate mAb-dependent differences of PrP$^d$ phenotypes in more detail.

### Fibril-like PrP$^d$ aggregates are tethered to the plasma membrane and extend to the ECM
Unless otherwise specified, all labelling experiments were conducted under denaturing conditions, i.e. in presence of guanidinium thiocyanate (GTC), where the majority of anti-PrP mAbs detect abnormal PrP[5,20,21]. Prolonged cultures (≥ 6 days) of prion-infected S7 cells displayed extensive fibril-like 5B2-positive PrP$^d$ aggregates that are tethered to the plasma membrane and extend to the ECM (Fig. 1C, upper panel). To normalise the distance of focal planes in z-direction, we used in-focus detection of the ECM-resident protein focal adhesion kinase (FAK), arbitrarily denoted "zero μm", as focal reference level as elaborated in "Methods" section and Supplementary Fig. 1F. Serial sections from the ECM to the focal plane of the nucleus demonstrate that 5B2 labelled PrP$^d$ at the plasma membrane, while the cytosolic face of the cell remained void with no 5B2-positive PrP$^d$ detected intracellularly. In contrast, 6D11 labelled punctate PrP$^d$ aggregates at perinuclear sites, at the plasma membrane and extracellularly (Fig. 1C, lower panel). Double-labelling experiments with 5B2 and 6D11 corroborate these phenotypic differences. Whilst both antibodies colabelled PrP$^d$ aggregates at the plasma membrane, 5B2 failed to detect perinuclear PrP$^d$ (Fig. 1D). In uninfected control cells, both antibodies weakly labelled the plasma membrane (Fig. 1E). In contrast to 5B2, mAbs against α-helix 1 (ICSM 18) and α-helix 2 (8H4) colabelled 6D11-positive PrP$^d$ aggregates at perinuclear sites as depicted in fluorescence intensity profiles (Fig. 1F). Double-labelled images for all selected monospecific mAbs are shown in Supplementary Fig. 2. While failure to label perinuclear PrP$^d$ with 5B2 is confirmed with the N-terminal mAb 8B4, all other mAbs labelled perinuclear PrP$^d$ aggregates.

To identify anti-PrP mAbs that best discriminate PrP$^d$ from PrP$^c$, we quantified the relative fluorescence intensities of mAbs in prion-infected and uninfected cells at all cellular compartments where PrP$^d$ is detected, i.e. the plasma membrane, perinuclear regions and the ECM. Voxel-based fluorescence intensities in Fig. 1G are normalised and provide a quantitative scale for their discriminatory propensities at the specified subcellular locations (see Supplementary Table 2 for data). Notably, at the plasma membrane, only 5B2 and 6D11 showed 10-fold or higher fluorescence intensities in infected versus uninfected cells, while a more intense PrP$^c$ label with ICSM18, Saf32 and 8H4 limits their use to detect PrP$^d$ at the plasma membrane. All mAbs, except 8H4, were sufficiently discriminatory to call PrP$^d$ at the ECM, while low differential fluorescence intensities for ICSM18 limit its use in detecting PrP$^d$ at perinuclear regions.

Phenotypic differences between intracellular and extracellular PrP$^d$ aggregates, detected with 5B2 and 6D11, respectively are not limited to mouse neuroblastoma cell lines, as shown in Fig. 1H, I. Primary neuronal cultures from embryonic e17 mouse brains, infected with RML prions showed 5B2-positive long fibril-like PrP$^d$ aggregates at the cell periphery of GFAP-positive astrocytes with 6D11-positive PrP$^d$ puncta at perinuclear sites at three weeks post infection (Fig. 1H), thus corroborating the phenotypic features of the mAb pair in neuroblastoma cells. When glial proliferation in primary neuronal cultures was inhibited with 1 μM cytosine arabinoside (AraC), followed by infection with RML or uninfected CD1 control homogenates, Map2-positive neurons with fibril-like PrP$^d$ aggregates were visible upon RML infection (Fig. 1I), demonstrating that this fibrillar PrP$^d$ phenotype is prevalent in neurons as well.

### Elongation, average length and structural features of fibril-like PrP$^d$ aggregates
To characterise the time-dependence of PrP$^d$ fibril elongation, we infected primary neuronal cultures from brains of FVB wild-type (wt) and prion-replication deficient FVB Prnp$^{-/-}$ mice with the mouse RML prions. In cultures from FVB wt mouse brains, fibril length continuously and significantly increased over three weeks, at mean fibril elongation rates of 190 nm per day, reaching lengths of up to 15 μm, while no fibrils were detected at the periphery of astrocytes from brains of FVB Prnp$^{-/-}$ mice (Fig. 2A, B).

To investigate the phenotype of fibril-like PrP$^d$ at subdiffraction resolution, we co-labelled PrP$^d$ at the ECM of prion-infected S7 cells with 5B2 and 6D11 and imaged labelled cells using structured illumination microscopy (SIM, Fig. 2C). Notably, extended segments of PrP$^d$ fibrils did not show co-labelling under denaturing conditions. Instead, 5B2 immuno-positivity prevailed with 6D11-positive puncta along the lengths and at ends of fibrils. Where 6D11 puncta were detected in close proximity to 5B2-positive fibrils, areas of colocalisation were apparent (see insets in Fig. 2C). This result is consistent with the observed punctate and fibril-like phenotype of PrP$^d$ aggregates in single-labelled prion-infected cells (Fig. 1C) and suggests (i) that different aggregation states may co-exist during PrP$^d$ fibril formation and (ii) that the charge cluster 2 region where 6D11 binds may be cryptic in highly ordered fibrillar aggregates.

### Identification of conformational and truncated PrP$^d$ variants in prion-infected cells
While it is well documented that some anti-PrP mAbs that recognise PrP$^c$ under native conditions fail to detect PrP$^d$ aggregates in prion-infected cells, unless cells are treated with denaturing agents after fixation[5,20,21], this phenomenon has not been broadly investigated with panels of validated anti-PrP mAbs. Our initial results show that prion-infected cells, labelled with the monospecific core mAbs 6D11, ICSM18

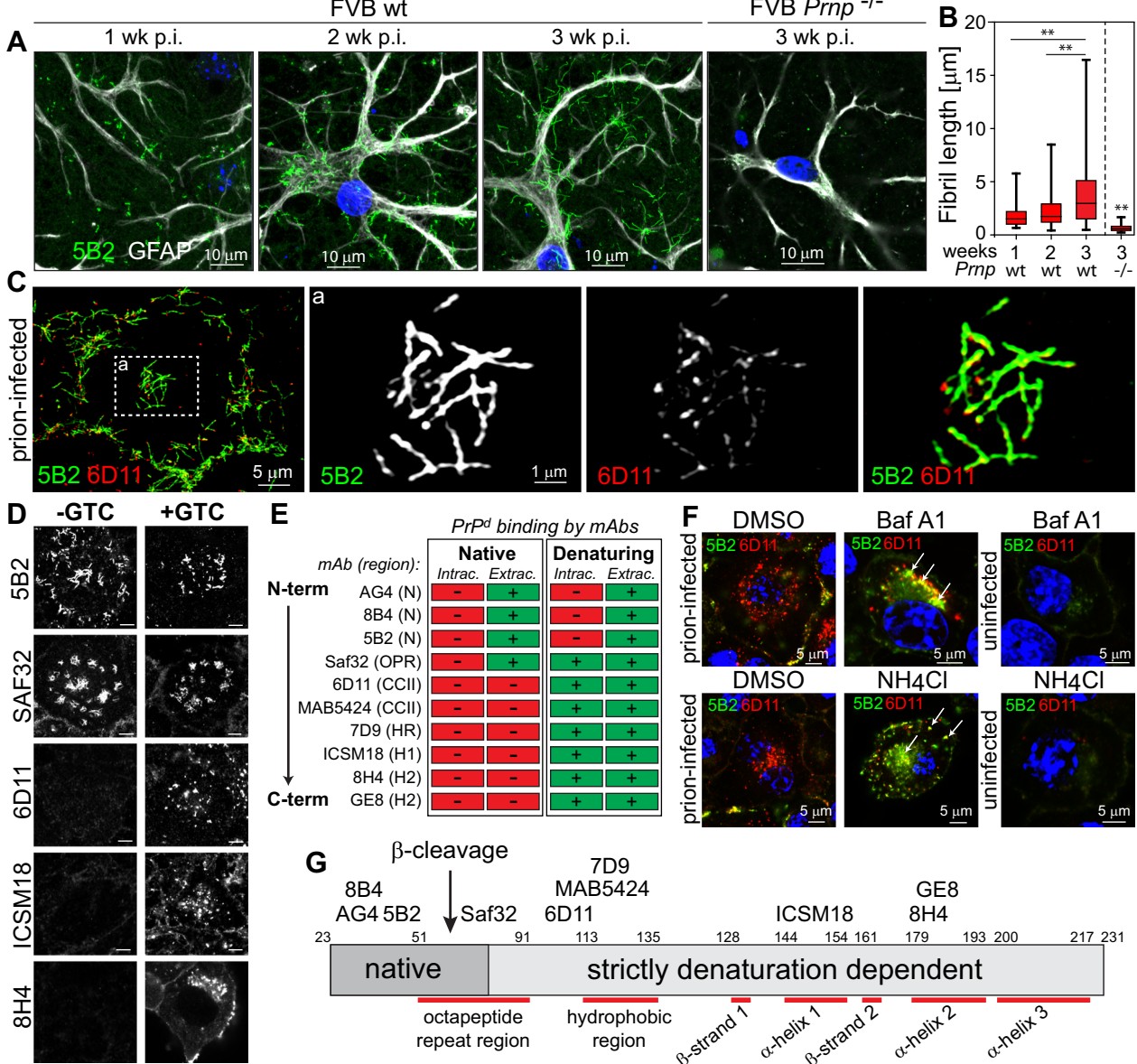

**Fig. 2 | Elongation and structural features of fibril-like PrP$^d$ aggregates and crypticity of PrP$^d$ epitopes. A** Time-dependent growth of 5B2-positive PrP$^d$ fibrils in primary astrocytes of FVB-wt (*Prnp$^{+/+}$*), but not in those of FVB-ko (*Prnp$^{-/-}$*) mice, following infection with RML. **B** Lengths of fibril-like PrP$^d$ aggregates in FVB-wt versus FVB-ko mice. Significance levels for time-dependent fibril growth in FVB-wt (Prnp$^{+/+}$) mice and for FVB-wt versus FVB-ko were assessed from 2 independent experiments with at least 30 replicates per group using Kruskal–Wallis test with Dunn's multiple comparisons with **$p < 0.001$. The statistical difference in rod length between FVB-wt versus FVB-ko mice at 3 weeks was determined by unpaired Mann–Whitney U Test; ** denotes a *p*-value < 0.001. For definition of boxplot elements see "Methods" section. **C** SIM images of fibril-like PrP$^d$ aggregates at the ECM of prion-infected S7 cells, cultured according to "standard TC protocol" in

"Methods" section, labelled with 5B2 and 6D11 after guanidinium thiocyanate (GTC) treatment. Magnified area (a) denoted by dashed box. **D** Identification of cryptic epitopes in PrP$^d$-bearing cells. Prion-infected S7 cells were incubated with anti-PrP mAbs in absence and presence of GTC. For complete image data set see Supplementary Fig. 4. Scale bars are 5 μm. **E** Summary of anti-PrP mAb binding to PrP$^d$ aggregates under native (-GTC) and denaturing (+GTC) conditions with corresponding PrP domains (for abbreviations of PrP regions see Supplementary Fig. 4). **F** Detection of intracellular 5B2-positive PrP$^d$ aggregates following a 16 h incubation with 6 nM bafilomycin A1 (BafA1) and 16 mM ammonium chloride (NH$_4$Cl), respectively. Arrows denote intracellular 5B2-positive PrP$^d$ aggregates. **G** Model depicting cryptic, exposed and putative PrP-cleavage sites, respectively (modified from Rouvinski et al.[14]). Source data are provided as a Source Data file.

and 8H4 in absence of GTC are phenotypically indistinguishable from uninfected cells (Supplementary Fig. 3), suggesting that those PrP epitopes are cryptic under native conditions. This notion however does not explain why 5B2 fails to detect intracellular PrP$^d$ under denaturing conditions (Fig. 1C, D, F).

We thus examined monospecific anti-PrP mAbs under native and denaturing conditions (Fig. 2D) at all sites of PrP$^d$ deposits (Supplementary Fig. 4). The binding characteristics of all investigated mAbs are summarised in Fig. 2E. In keeping with the binding characteristics

of 5B2 in Fig. 1, all N-terminal mAbs detected fibril-like PrP$^d$ at extracellular sites (ECM and plasma membrane), but not intracellularly. MAbs that bind to the "core" PrP region (90-231)[22] failed to detect PrP$^d$ in prion-infected cells under native conditions, while PrP$^d$ aggregates were readily detected following denaturation. Saf32, an antibody that has multiple binding sites at the octapeptide repeat region[23] showed notable propensities, in that it detected extracellular fibril-like PrP$^d$ under native conditions, like N-terminal mAbs and intracellular PrP$^d$ under denaturing conditions, unlike N-terminal mAbs, suggesting that

extracellular and intracellular PrP$^d$ species are conformationally distinct at the octapeptide repeat region (Fig. 2E and Supplementary Fig. 4). Alternatively, inherent conformational differences may exist between N-proximal and C-proximal octapeptide repeats that render C-proximal octapeptide repeats cryptic and invariant to limited proteolysis. Notably, a considerable body of evidence shows that PrP is cleaved at the level of the OPR, giving rise to the C2 fragment[24,25].

To investigate whether failure to detect intracellular PrP$^d$ with N-terminal mAbs is due to proteolytic processing, we tested whether an increase in the endosomal/lysosomal pH may restore intracellular PrP$^d$ labelling by 5B2. Activation of endosomal/lysosomal proteinases is tightly regulated by the intraorganellar pH[26] and bafilomycin A1 (BafA1), an inhibitor of V-ATPases prevents luminal acidification[27]. Notably, dissipation of the luminal pH with BafA1 and the lysosomotropic agent NH$_4$Cl gave rise to intracellular 5B2-positive aggregates in prion-infected cells (Fig. 2F), suggesting that truncation of full-length (FL-)PrP$^d$ is blocked upon inhibition of pH-dependent proteinases, thus excluding crypticity as cause for the failure to detect intracellular FL-PrP$^d$. Data on cryptic epitopes and the putative β-cleavage site of PrP are summarised in Fig. 2G.

In summary, we provide evidence that PrP$^d$ undergoes proteolytic processing in a pH-sensitive manner, while core mAbs fail to detect abnormal PrP under native conditions due to crypticity of their corresponding binding sites. In contrast, elongating PrP$^d$ fibrils at the plasma membrane can be labelled with N-terminal mAbs under native conditions in agreement with Rouvinski et al.[14]. These results suggest existence of two distinct aggregated PrP$^d$ species in prion-infected cells, FL-PrP$^d$ which is associated with fibril elongation at the plasma membrane and a truncated (TR-) PrP$^d$ type at perinuclear sites. Evidence that these two PrP$^d$ types are conformationally distinct at the level of the octapeptide repeat region (OPR) is suggested by the binding propensities of the monospecific anti-PrP mAb Saf32 which labels FL-PrP$^d$ at the plasma membrane under native conditions, but TR-PrP$^d$ at perinuclear sites under denaturing conditions.

## Asynchronous changes in intra- and extracellular PrP$^d$ pools after cell dissociation

Dynamic shifts in prion steady-state levels have been reported after cell dissociation of persistently infected cells[28]; an initial rapid drop of PrP$^{Sc}$, measured as a proxy for prion levels was followed by gradual recovery of PrP$^{Sc}$ levels[28]. We used this experimental paradigm to further characterise the relationship between intra- and extracellular PrP$^d$ pools. Confluent layers of chronically infected S7 cells were resuspended and replated into fresh medium (Supplementary Fig. 5A). Following cell trituration, long 5B2-positive PrP$^d$ fibrils remained attached to the substrate (Supplementary Fig. 5B). Attempts to immuno-isolate PrP$^d$ fibrils from decellularised ECM (dECM) following stringent lysis with RIPA buffer failed (Supplementary Fig. 5C). Under these conditions, 5B2-positive fibrils were still visible at the dECM (Supplementary Fig. 5D). Unexpectedly, when cells were resuspended and replated according to Supplementary Fig. 5A, 5B2-positive PrP$^d$ aggregates were absent in cells within the first two days after replating, while punctate 6D11-positive PrP$^d$ aggregates were detected at the ECM and at perinuclear sites (Supplementary Fig. 5E). 5B2-positive PrP$^d$ aggregates at the plasma membrane were first detected at day three, after which a rapid rise of 5B2- and co-labelled PrP$^d$ aggregates was evident (Supplementary Fig. 5E, F). This result suggests that cell dissociation leads to a quantitative clearance of FL-PrP$^d$ and that fibril growth at the plasma membrane commences after a considerable lag time following cell dissociation. To provide further evidence that FL-PrP$^d$ is cleaved under these conditions, we exploited our previous observation that dissipation of the luminal pH by BafA1 blocked truncation of FL-PrP$^d$ in persistently infected cells (Fig. 2F).

Notably, BafA1 treatment of infected S7 cells led to (i) detection of abundant intracellular 5B2-positive FL-PrP$^d$ at 2 days after BafA1 treatment (Supplementary Fig. 5G) and (ii) a significant rise in 5B2 fluorescence intensity in prion-infected, compared to uninfected cells (Supplementary Fig. 5H). Prion propagation in S7 cells is associated with proteolytic processing of PrP, giving rise to truncated PrP conformers as evident in cell lysates after deglycosylation with PNGase (Supplementary Fig. 5I); bands, representing full-length (FL-) and truncated (TR-) PrP are indicated. As evident on Western blots of cell lysates, BafA1 treatment of infected (Supplementary Fig. 6A), but not of uninfected cells (Supplementary Fig. 6C) led to higher levels of FL-PrP. Proteinase K (PK) digestion of cell lysates from infected cells gave rise to PK-resistant PrP (PrP$^{Sc}$), while PrP$^c$ in cell lysates from uninfected cells was fully digested (band 5 in Supplementary Fig. 6A). Notably, BafA1 treatment of infected cells resulted in higher levels of PrP$^{Sc}$, when compared to mock-treated cells. Since endolysosomes have been proposed as a potential site where abnormal PrP is N-proximally truncated[8], we enriched lysosomes of BafA1 and vehicle-treated cells by immunoprecipitation (Supplementary Fig. 6D). A band shift from TR-PrP to FL-PrP upon BafA1 treatment is apparent in Lamp1-enriched lysosomes (Supplementary Fig 6D). In summary, we provide evidence that cell dissociation leads to asynchronous changes in extracellular and intracellular PrP$^d$ pools. The slow kinetics in PrP$^d$ fibril formation is unexpected and suggests that considerable cellular or energetic barriers have to be overcome for PrP$^d$ fibrils to form.

## Formation of PrP$^d$ fibrils at the plasma membrane is associated with synaptic vesicle trafficking and sensitive to cholesterol-lowering

We next examined whether the antibody pair 5B2/6D11 may provide further insight into the mode of fibril formation at the plasma membrane. At subdiffraction resolution, 6D11-positive puncta can be detected at close proximity to 5B2-positive fibrils, suggesting that TR-PrP$^d$ aggregates are recruited to membrane areas where PrP$^d$ fibril form (Fig. 3A). A pivotal question is thus whether TR-PrP$^d$ reaches the plasma membrane by passive diffusion or by vesicular transport. Triple labelling of prion-infected cells with Synapsin 1 (Syn1), a marker for synaptic vesicles and anti-PrP mAbs 6D11/5B2 provides evidence that PrP$^d$ may reach the plasma membrane by vesicular transport (Fig. 3B). Single-, double- and triple-labelled puncta can be detected at focal planes above the nucleus and underscore the heterogeneity of overlapping distribution patterns. Double-labelling of PrP$^d$ and integrin β1, a membrane spanning protein which favourably visualises plasma membrane boundaries demonstrates that PrP$^d$ is detected proximal to (short hatched arrows) and at the level of (long straight arrows) the plasma membrane (Fig. 3C), suggesting that PrP$^d$ may dock and fuse with the plasma membrane.

Since lowering cholesterol impairs the docking and fusion of transport vesicles with the plasma membrane[29,30], we incubated cells with the cholesterol-lowering agent methyl-β cyclodextrin (mβCD) in a dose-dependent manner to reduce cellular cholesterol. While cellular cholesterol levels decreased by 30-40% under these conditions, cytotoxic effects, measured by a sensitive ATP assay, were absent below 1 mM mβCD (Fig. 3D). Notably, at 0.4 mM mβCD, the fluorescence intensities of 6D11 and 5B2 decreased considerably at the plasma membrane, while 6D11-positive PrP$^d$ aggregates were still detected intracellularly at perinuclear sites (Fig. 3E). Quantitative analysis of fluorescence areas confirmed a significant reduction of 5B2 fluorescence and 6D11/5B2 colabelling at a concentration range between 0.4 mM and 0.8 mM mβCD (Fig. 3F). To exclude the possibility that mβCD treatment leads to an extraction of PrP$^c$ under the experimental conditions of Fig. 3E, F, we treated uninfected cells with mβCD concentrations of up to 2 mM (Fig. 3G), confirming that PrP$^c$ levels are not reduced at mβCD concentrations below 2 mM. A slight increase in PrP$^c$

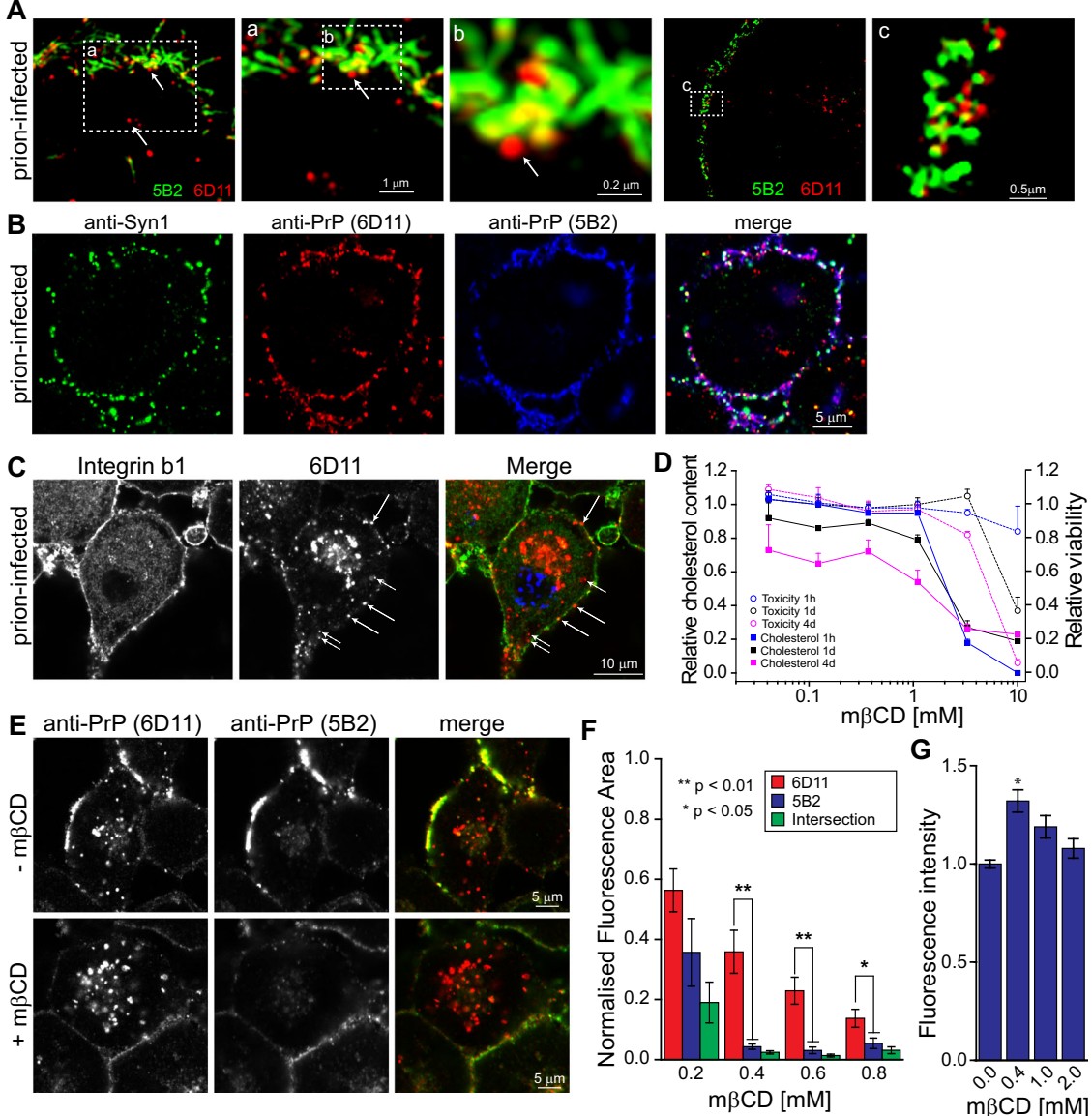

**Fig. 3 | The formation of PrP$^d$ fibrils at the plasma membrane is associated with vesicular trafficking and is sensitive to cholesterol-lowering. A** SIM images of prion-infected S7 cells, cultured according to "standard TC protocol" in "Methods" section, double-labelled with 5B2 and 6D11 at the level of the plasma membrane. Magnified areas a, b and c are denoted by dashed boxes. Arrows denote punctate 6D11-positive PrP$^d$ in juxtaposition with 5B2-positive PrP$^d$ fibrils. Estimated lateral resolution SIM: 100 nM. **B** Triple-labelling of prion-infected S7 cells with anti-Syn1, 6D11 and 5B2. For quantification of colocalisation, see Fig. 4G. **C** Prion-infected S7 cells, labelled with anti-integrin β1 and 6D11 show PrP$^d$ aggregates proximal to (short hatched arrows) and at the level of (long straight arrows) the plasma membrane. **D** Dose-response relationship of mβCD on cholesterol-lowering and cell viability in prion-infected S7 cells at the time points and mβCD concentrations specified. **E** Colabelling of PrP$^d$ with 6D11 and 5B2 at the plasma membrane in

presence and absence of 0.4 mM mβCD at 16 h following mβCD addition. **F** Quantification of mβCD effects on total fluorescence above threshold and colabelling of PrP$^d$ with 6D11 in red, 5B2 in blue and 6D11/5B2 intersection in green at 16 h after addition of specified inhibitor concentrations. Data is normalised to vehicle control (1.0) and represents mean and SEM. Statistical significance was determined by ANOVA. **G** Dose-dependent effects of mβCD on PrP$^c$ levels at the plasma membrane during a 16 h-incubation as specified in "Methods section" "Quantifying relative PrP$^d$ fluorescence intensities". Mean values and SEM of three independent experiments are shown. Statistical significance was assessed by ANOVA with Bonferroni multiple comparison test with at least 148 images per condition analysed. * denotes *p*-value < 0.01. Source data are provided as a Source Data file.

surface levels between 0.4 and 1 mM mβCD may be due to a higher cell surface retention of PrP$^c$. In summary, the drop of 6D11/5B2 colabelling following the lowering of cellular cholesterol levels suggests (i) that intra- and extracellular pools of PrP$^d$ can be uncoupled and (ii) that blocking of PrP$^d$ transport to the plasma membrane inhibits the formation of FL-PrP$^d$ at the plasma membrane. Alternatively to inhibition of vesicle transport, a perturbation of lipid rafts by mβCD could account for the reduced 5B2/6D11 colocalisation at the plasma membrane. Lipid rafts are thought to play an important role in the formation and propagation of prions[31–33].

## PrP$^d$ is segregated into vesicles of the regulated secretory pathway

Experimental evidence that PrP$^d$ colocalises with a marker for synaptic vesicles (Fig. 3B) prompted us to further investigate the contribution of other secretory pathways to PrP$^d$ trafficking. Perinuclear PrP$^d$ colocalises with protein markers of large dense-core (Secretogranin 2 (Scg2)), synaptic (Syn1) and immature (Vesicle-associated membrane protein 4 (Vamp4)) vesicles (Fig. 4A–C), providing evidence that PrP$^d$ egresses cells by regulated exocytosis. In contrast, no colocalisation between PrP$^d$ and Collagen 4 (Col4), an extracellular matrix protein

that is constitutively secreted, was observed (Fig. 4D). To map the route of PrP$^d$ aggregates from perinuclear sites to the plasma membrane, we determined the degree of 6D11 colocalisation, expressed as Pearson correlation coefficient, with markers of the ER (P4hb), Golgi network (β-Cop, Grasp55, Gm130), secretory vesicles (Syp, Vamp4, Syn1, Scg2, Chga, Col4) and endosomal/lysosomal markers (Eea1, Lamp1) (Fig. 4F, G). Representative images for all studied marker proteins, co-labelled with 6D11 can be found in Supplementary Fig. 7. This result confirms a role of the vesicular secretory pathways and the endosomal/lysosomal system in PrP$^d$ trafficking. To further examine whether intracellular TR-PrP$^d$ segregates into regulated exocytosis, we triggered membrane depolarisation with potassium chloride (KCl). Depolarisation led to a fast depletion of Syn1-positive synaptic vesicles (Fig. 4H) and a significant increase in PrP$^d$ aggregates at the plasma membrane (Fig. 4I, J), confirming that PrP$^d$ reaches the plasma membrane by regulated exocytosis. In contrast, in uninfected S7 cells, KCl-evoked depolarisation did not lead to an increase in PrP$^c$ levels at the plasma membrane (Fig. 4K).

## PrP$^d$ internalisation is dependent on functional dynamin and Cdc42

We next investigated whether perturbation of endocytosis affects the prion steady-state level of persistently infected cells. While evidence suggests that PrP$^c$ is internalised via clathrin-dependent endocytosis, mediated by its N-terminal domain[34,35], the mode of PrP$^d$ endocytosis remains obscure. We here targeted known regulators of clathrin-dependent endocytosis (CME) and clathrin-independent endocytosis (CIE) pathways, using pharmacological inhibitors and RNA interference. While CME is considered the main endocytic pathway for "housekeeping functions" in cells, a growing number of fast, non-canonical pathways, classified as CIE, has been characterised and include Cdc42-, RhoA-, Arf6-regulated pathways[36,37].

Absence of toxicity was carefully monitored for all inhibitor studies and toxic thresholds are reported in Supplementary Table 3. Two pharmacological dynamin inhibitors, dynasore[38] and dynole[39] showed a strong dose-dependent decrease of cellular prion steady-state levels by as much as 90% over three days, as determined by Scrapie Cell assay (SCA, Fig. 5A). However, the clathrin inhibitor Pitstop 2[40] did not affect prion steady-state levels. Further, the macropinocytosis inhibitor Amiloride[41] and the actin polymerisation inhibitor Cytochalasin D showed moderate effects on prion steady-state levels (Fig. 5A). Two frequently used inhibitors, the CIE inhibitor EGA and Vacuolin which induces vacuole formation by homotypic fusion of endosomes and lysosomes[42], reduced prion steady-state levels significantly by 60% and 80%, respectively (Supplementary Table 4), but the molecular targets of either inhibitor remains unknown.

To corroborate a role of CIE pathways in the maintenance of cellular prion steady-state levels, we next targeted key regulators of endocytic pathways using RNA interference. To effectively silence gene expression, we used pools of 30 custom-designed siRNAs (siPools)[43] per gene target. We first tested the use of siPools by silencing *Prnp*, since a loss of *Prnp* expression directly compromises prion propagation. Knockdown of *Prnp* leads to a 86% decrease in *Prnp* expression levels, measured by qPCR, when compared to a non-targeting control (NT control, Fig. 5B). To verify that transient *Prnp* knockdown persists over 3 days of culture, we next examined the time-dependence of gene knockdown, following conventional versus reverse transfection of cells. Reverse transfection, where siPools are directly mixed with suspended cells leads to a fast and sustained knockdown of *Prnp* levels over the duration of 72 h, while conventional transfection, where siPools are added to adherent cells 16 h after plating leads to higher variance and lower knockdown efficacies (Fig. 5C). That RNA silencing of *Prnp* by reverse transfection leads to efficient curing of prion-infected cells is corroborated by labelling cells

with discriminatory anti-PrP mAbs (Fig. 5D). Following proof of concept for the use of siPools in modulating prion steady-state levels, we next targeted key regulatory genes involved in endocytosis. Levels of knockdown for individual gene targets are reported in Supplementary Table 5. Despite a 98% knockdown of gene expression, clathrin a (Clta) loss-of-function did not affect prion steady-state levels (Fig. 5E). Knockdown of the dynamin isoforms Dnm1 and Dnm2 led to a decrease of prion steady-state levels by 20-30% in single knockdown experiments, while double-knockdown resulted in a 50% decrease in prion steady-state levels (Fig. 5E). A highly efficient knockdown of cell division cycle 42 (Cdc42) by 97% led to a 50% drop of cellular prion titres, suggested that Cdc42-mediated CIE plays an important role in maintaining prion steady-state levels. Conversely, depletion of ADP ribosylation factor 6 (Arf6), Ras homologue family member A (Rhoa), Arf1 and Caveolin 1 (Cav1) did not affect prion steady-state levels (Fig. 5E). In summary, we provide evidence that prion propagation in N$_2$a cells is dependent on functional dynamin and Cdc42, but independent of clathrin.

## Evaluating the minimum contact time required for productive infection

Due to the lack of PrP$^d$-specific anti-PrP mAbs, the kinetics of de novo PrP conversion is one of the most unexplored questions. While Goold et al., who expressed myc-tagged PrP to distinguish between PrP$^c$ and PrP$^{Sc}$ suggested that de novo conversion commences within minutes after infection[13], Yamasaki et al., who used a PrP$^{Sc}$-preferring mAb suggested that PrP$^{Sc}$ is formed within days after infection[44].

To investigate discordances in the kinetics of PrP$^d$ formation, we first determined the minimum contact time for prions with cells required for productive infection to occur, henceforth termed "contact time". Productive infection is here defined as an infectious event that leads to persistent prion infection. We thus monitored incremental changes of the cell infectious state over several passages after infection in relation to the initial contact time using the SCA. As depicted in the schematic of Fig. 5F we infected S7 cells with the mouse-adapted prion strain RML on a time-scale between 2 min and 24 h in parallel, followed by gentle washing of cells to remove the inoculum. Cells were then grown to confluence and passaged three times before the proportion of infected cells in the total cell population was determined. The relationship between contact time and the percentage of infected cells, following two weeks of cell culture is depicted in Fig. 5G and the corresponding mean percent infection rates were included in the lower panel of Fig. 5F. This confirms that a contact time of 2 min suffices to infect neuronal cells, while the mean contact time required for infecting 50% of the cell population, defined as PI$_{50}$ (PI for productive infection) is 4 h. Furthermore, a 2- to 4-fold increase in the percentage of infected cells between subsequent cell passages confirms stable propagation of prions (Fig. 5H). When plotted as a linear-log graph, a highly correlated linear relationship between the logarithm of contact time and the percentage of infected cells is apparent from 30 min to 24 h, while a minor cell population undergoing fast infection after 2 min of contact time is in the non-linear region of the plot (Fig. 5I).

We next addressed the problem of residual inoculum. Crude brain homogenates from mice at disease end stage, typically used for infections, are a non-physiological source of infectivity and may present unspecific effects. We thus tested alternative prion sources. Exosomes are biological nanoparticles that are formed from late endosomes by inward budding into multivesicular bodies[45,46] and are constitutively shed into the cell supernatant by prion-infected cells[47]. When compared to RML, infectious exosomes showed a 10-fold lower immuno-positive background (residual inoculum) following infection (Supplementary Fig. 8A) and lower toxicity at high infectious doses (Supplementary Fig. 8B). Following infection of

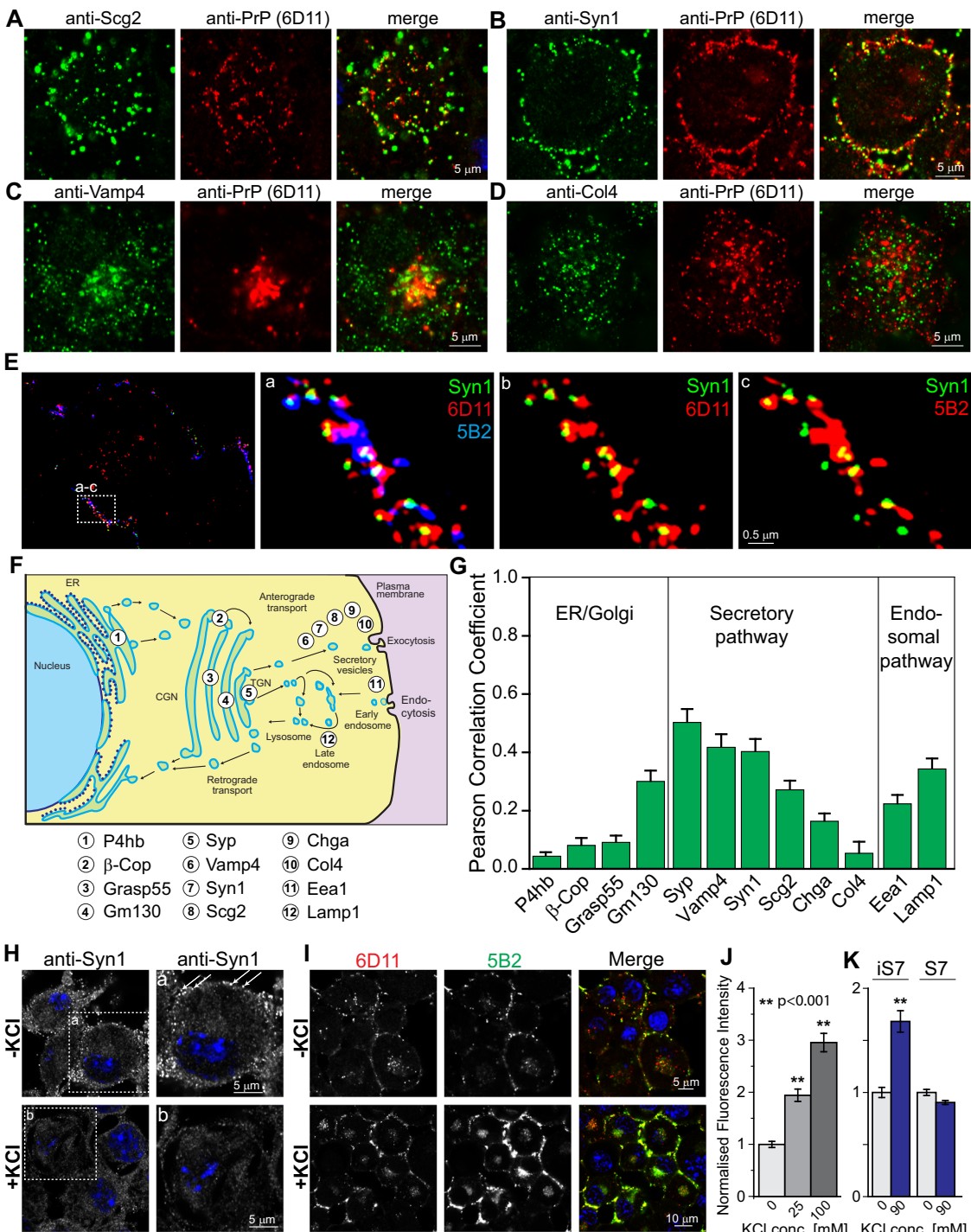

**Fig. 4 | PrP[d] is segregated into exocytotic vesicles of the regulated secretory pathway. A–D** Persistently prion-infected S7 cells were co-labelled with 6D11 and vesicular markers (**A**) Scg2, (**B**) Syn1 and (**C**) Vamp4 as well as with constitutively secreted Col4 (**D**). **E** SIM image of plasma membrane, triple-labelled with Syn1, 6D11 and 5B2. Magnified areas (a–c) are denoted by a dashed box in the first image. **F** Diagram of marker proteins used to map trafficking routes of 6D11-positive PrP[d]. **G** Levels of colocalisation between 6D11 and organelle/vesicular markers, expressed as Person correlation coefficients. Data from two independent experiments with at least 10 per protein are shown. For representative images and gene names we refer to Supplementary Fig. 7. **H** Neuronal depolarisation with 25 mM KCl led to rapid depletion of Syn1-positive vesicles in S7 cells. Cells were fixed at 5 min after incubation with KCl. Arrows in magnified areas denote synaptic vesicles. **I** Increased

levels of co-labelled PrP[d] aggregates at the plasma membrane, following KCl-evoked depolarisation. Cells were fixed at 5 min after incubation with KCl. **J** Quantitative changes of co-labelled PrP[d] after depolarisation in dependence of the KCl concentration normalised to untreated cells from three independent experiments with at least 36 replicates per condition. For statistical analysis, ANOVA with Bonferroni correction for multiple comparisons was conducted. **K** Quantitative changes in surface PrP levels after KCl-evoked depolarisation in uninfected S7 versus prion-infected iS7 cells. Following fixation, cells were stained with anti-PrP antibody 8H4. Data from three independent experiments with at least 48 images per experiment were analysed. Data represent mean values ± SEM. Statistical significance was evaluated by Student's t-test ($p < 0.01$). Source data are provided as a Source Data file.

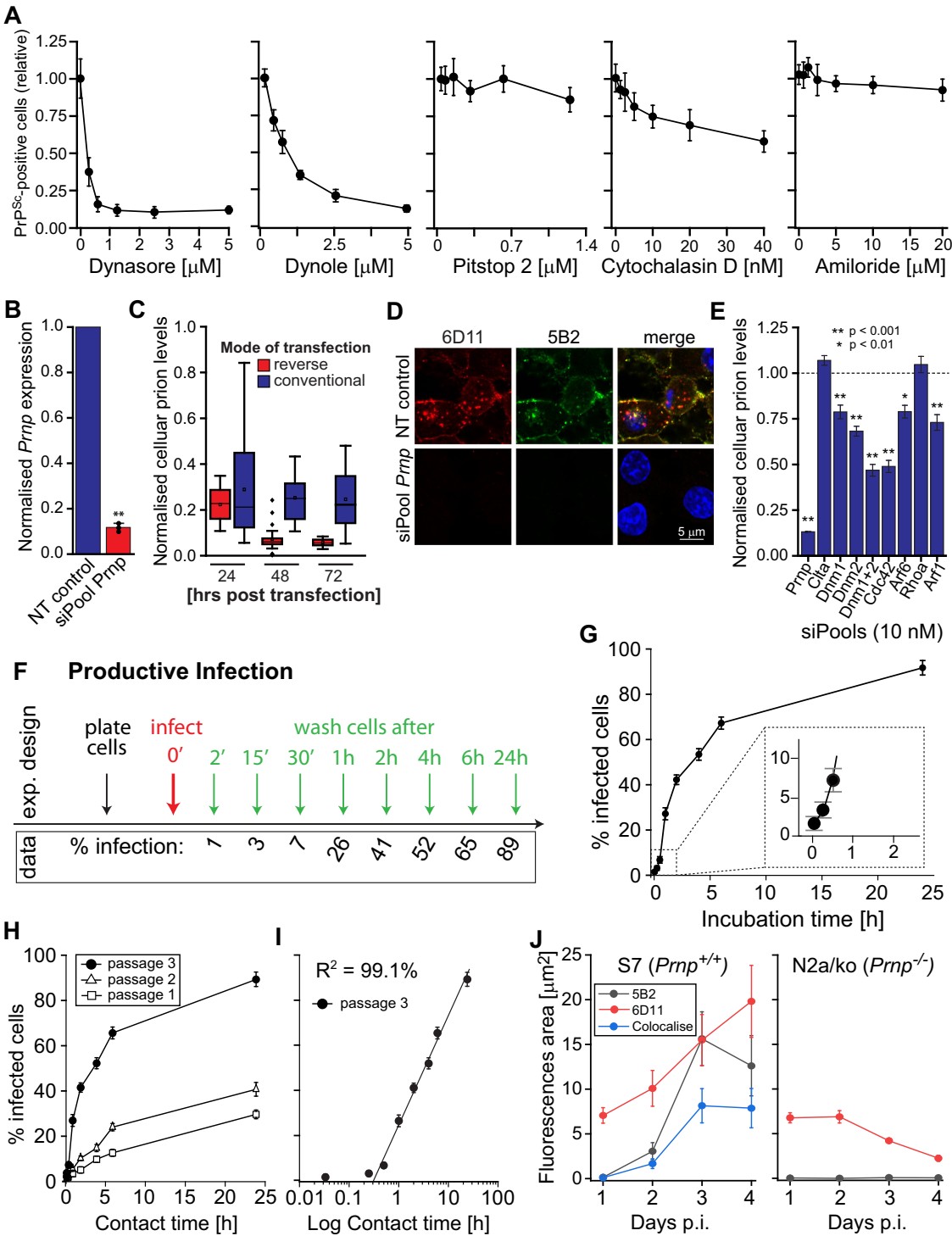

replication-incompetent Prnp[-/-] cells with exosomes from chronically infected S7 cells, 6D11-positive spots were detected intracellularly 1, 2 and 3 days after infection (Supplementary Fig. 8C). Remarkably, 5B2-positive FL-PrP[d] was not detected in susceptible S7 cells prior to 2 day after infection (Fig. 5J and Supplementary Fig. 8C), indicating that the formation of fibril-like PrP[d] at the plasma membrane significantly lags behind infection.

## Prion infection leads to PrP conversion at two distinct cellular locations and precedes the formation of FL-PrP[d]

While a contact time of 2 min suffices to infect neuronal cells (Fig. 5F–I), early events after prion infection cannot be resolved using

anti-PrP mAbs. We therefore generated a cell model expressing myc-tagged versions of PrP to distinguish newly formed abnormal PrP from inoculum. Myc-tagged Prnp chimera were expressed in PK1-10 cells where endogenous Prnp is stably silenced with a small hairpin RNA targeting the 3'-UTR of Prnp[13]. Our data on truncation and crypticity of PrP epitopes (Fig. 2D–G) provides guiding information for a rational design of N-terminally tagged PrP variants and suggests that the OPR meets critical prerequisites for monitoring de novo PrP conversion. To test this hypothesis, we generated the myc-PrP fusion proteins Gly30-myc PrP (henceforth G30 PrP), G45 PrP, G70 PrP (OPR) and Q90 PrP (CC2) as depicted in Fig. 6A and characterised the tagged fusion proteins. Susceptibility to prion infection of myc-Prnp expressing cells,

**Fig. 5 | Identification of PrP$^d$ internalisation pathways and the minimum contact time for productive cell infection. A** Persistently prion-infected S7 cells were incubated with endocytosis inhibitors for 3 days and changes in prion levels were determined. Mean values ± SEM of at least three independent experiments are shown (n = 12). For toxic threshold levels, SCA data and significance levels see Supplementary Tables 3 and 4. **B** Gene silencing of *Prnp* in S7 cells using pooled siRNA, determined by RT-qPCR; mean values + SD of three independent experiments shown (*p* < 0.01). **C** Time-dependent effect of *Prnp* silencing on prion levels following conventional (blue bars) and reverse (red bars) transfection of cells from two independent experiments. For definition of boxplot elements see "Methods" section. **D** Effect of *Prnp* knockdown on PrP$^d$ aggregates in chronically infected cells following reverse transfection with siPools against *Prnp* and a NT control. **E** Silencing of gene targets associated with endocytosis in persistently infected S7 cells and its effect on prion levels. For knockdown efficacies and gene names see Supplementary Table 5. **F** Schematic for assessing the minimum contact time for

productive prion infection. S7 cells were plated out in parallel and infected with clarified RML homogenates (see "Methods" section "Minimum contact time for productive infection") at a $10^{-4}$ dilution (v/v, red arrow), followed by gentle washing of cells with PBS at the specified time points (green arrows) to remove inoculum. Cells were grown to confluence and the proportion of infected cells determined. **G** Proportion of prion-infected S7 cells, determined by SCA, plotted against contact times after infection with RML. Inset: magnified area of the proportion of infected cells at early time points, 2′, 15′ and 1 h. **H** Relationship between contact time and proportion of infected cells following infection with RML during subsequent cell passages. **I** Replotted data from (**G**) as a semi-logarithmic (linear-log) graph. **J** Prion replication-permissive (S7) and -refractory (N2a/ko) cells were infected with prion-infected exosomes and the formation of 6D11-, 5B2-positive and double-positive PrP$^d$ determined. Data represent average values ± SD of three independent experiments. Source data are provided as a Source Data file.

inferred from the number of PrP$^{Sc}$-positive cells following infection with RML prions, increased in the order G30 PrP, G45 PrP to G70 PrP, while Q90 PrP was refractory to infection (Fig. 6B). Notably, failure in detecting intracellular myc-tagged PrP$^d$ in persistently infected G30 and G45, but not in G70 PrP-expressing cells under denaturing conditions confirms that PrP$^d$ is N-terminally truncated prior to the third octapeptide repeat (Supplementary Figs. 8D and 9A, B). Remarkably, the myc tag is rendered cryptic in G70 PrP$^d$ which can be unambiguously shown in persistently infected G70 PrP-expressing cells treated in absence and presence of GTC (Supplementary Fig. 8E). While myc-tagged PrP$^d$ can be distinguished from PrP$^c$ in uninfected cells by fluorescence intensity and the granularity of PrP$^d$ aggregates, respectively (Supplementary Fig. 8D), we colabelled cells with the discriminatory anti-PrP mAb 6D11 and anti-myc to unambiguously detect infected cells (Fig. 6C, D and Fig. 7A, B).

Next, we interrogated de novo PrP conversion in G70 *Prnp*-expressing cells, infected with RML or exosomes, according to the experimental design in Fig. 5F. Notably, infection of cells with exosomes gave rise to myc-tagged PrP$^d$ in two distinct subcellular sites, the plasma membrane and perinuclear sites as early as 2 min after infection (Fig. 6C). We next identified the sites of PrP conversion at 2 and 15 min after infection. Irrespective of the source of inoculum, myc PrP$^d$ aggregates were detected in perinuclear regions (Peri), at the plasma membrane (PM) or in either location as depicted by Venn diagrams in Fig. 6D. All experiments were scored independently by three investigators and data is summarised in Supplementary Table 6. This suggests that abnormal PrP is formed rapidly after infection at two distinct cellular locations, the plasma membrane and an unidentified location at perinuclear sites of neuronal cells.

We next determined the proportion of phenotypic cells at early time points after infection with RML prions (Fig. 7A). Cells bearing de novo converted PrP (6D11/myc double-labelled) account for less than 0.2% of the total cell population at 2 and 15 min and less than 1% at 4 h after infection. For comparison, we reblotted the number of prion-infected cells following 2 weeks of tissue culture, representing perpetuating prion propagation after infection (Fig. 5G). An up to 80-fold increase in the number of prion-infected cells at 2 weeks, compared to the initial myc PrP$^d$-bearing cells suggests that an unexpectedly low number of cells becomes infected initially, while prion replication and cell-to-cell transmission account for the rapid spread of prions.

To address the discrepancy between the fast kinetics of PrP conversion and the delayed formation of FL-PrP$^d$, we infected G70 PrP-expressing cells according to Fig. 7A and labelled them with the discriminatory anti-PrP mAb pair 5B2/6D11 (Fig. 7B). Notably, 5B2-positive cells could not be detected before 24 h after infection, suggesting that de novo PrP conversion precedes FL-PrP$^d$ fibril formation at the plasma membrane.

The low frequency of de novo PrP converting cells (Fig. 7A) greatly limits investigation of the underpinning mechanism of PrP conversion.

However, since prion steady-state levels in persistently infected cells are highly dependent on functional dynamins and Cdc42, we further explored whether infection of cells can be inhibited by transcriptional silencing of dynamin and Cdc42, respectively (Fig. 7C). In fact, single knockdown of Cdc42, double-knockdown of Dnm1 and Dnm2 (Dnm1/2) and triple-knockdown (Dnm1/2/Cdc42) significantly blocks infection, suggesting that internalisation of the infectious seed is critical in establishing prion infection.

## Discussion

The self-perpetuating replication of prions is a central tenet of prion pathogenesis and represents a prototypic non-nucleic acid-based mechanism of infection and heritability. While long believed to be unique to prions, self-assembly and aggregation of proteopathic seeds is now considered a common trait of amyloids in many neurodegenerative diseases, including Alzheimer's and Parkinson's disease. It is therefore pivotal to identify common and specific disease pathways to assess potential risk and therapeutic strategies alike. This study has advanced our knowledge on the molecular underpinning of prion propagation in several ways.

Built on our initial observation of distinct PrP$^d$ phenotypes in neuronal cells and primary astrocytes, we provide evidence that extracellular fibril-like FL-PrP$^d$ is exclusively formed and elongated at the plasma membrane. In contrast, proteolytically truncated TR-PrP$^d$ aggregates reside at perinuclear sites and segregate into the regulated exocytotic pathway, where they dock and fuse with the plasma membrane, thus providing evidence for a directional transport of PrP$^d$ seeds, opposed to passive diffusion. We further provide evidence that FL-PrP$^d$ gives rise to distinct aggregation states, where the charge cluster 2 region remains cryptic under denaturing conditions over extended segments, suggesting that amorphous and fibrillar aggregates may co-exist during fibril formation at the plasma membrane. Notably, N-terminally truncated TR-PrP$^d$ at perinuclear sites fails to form fibrillar aggregates which suggests that the PrP N-terminus might be necessary for fibril formation. This is in agreement with the notion that the N-terminus mediates higher-order aggregation processes[48–50]. Owing to the lack of fibril formation, TR-PrP$^d$ may represent amorphous aggregates or oligomers.

That the prion state of cells is dependent on functional dynamins and Cdc42 implies that reuptake of PrP$^d$ aggregates in prion-infected cells is critical to maintain prion propagation. While PrP$^c$ is believed to be internalised via CME[35], we provide evidence that PrP$^d$ reuptake is mediated by CIE. This notion is associated with another important aspect of PrP$^d$ turnover, proteolytic processing. In keeping with the observation that FL-PrP$^d$ is absent intracellularly, unless cells are treated with BafA1, proteolytic processing of FL-PrP$^d$ following reuptake may result in fibril fragmentation, a cellular process that is considered critical to maintain prion replication. In contrast, prion propagation in yeast is believed to be strongly dependent on chaperon activity[51,52].

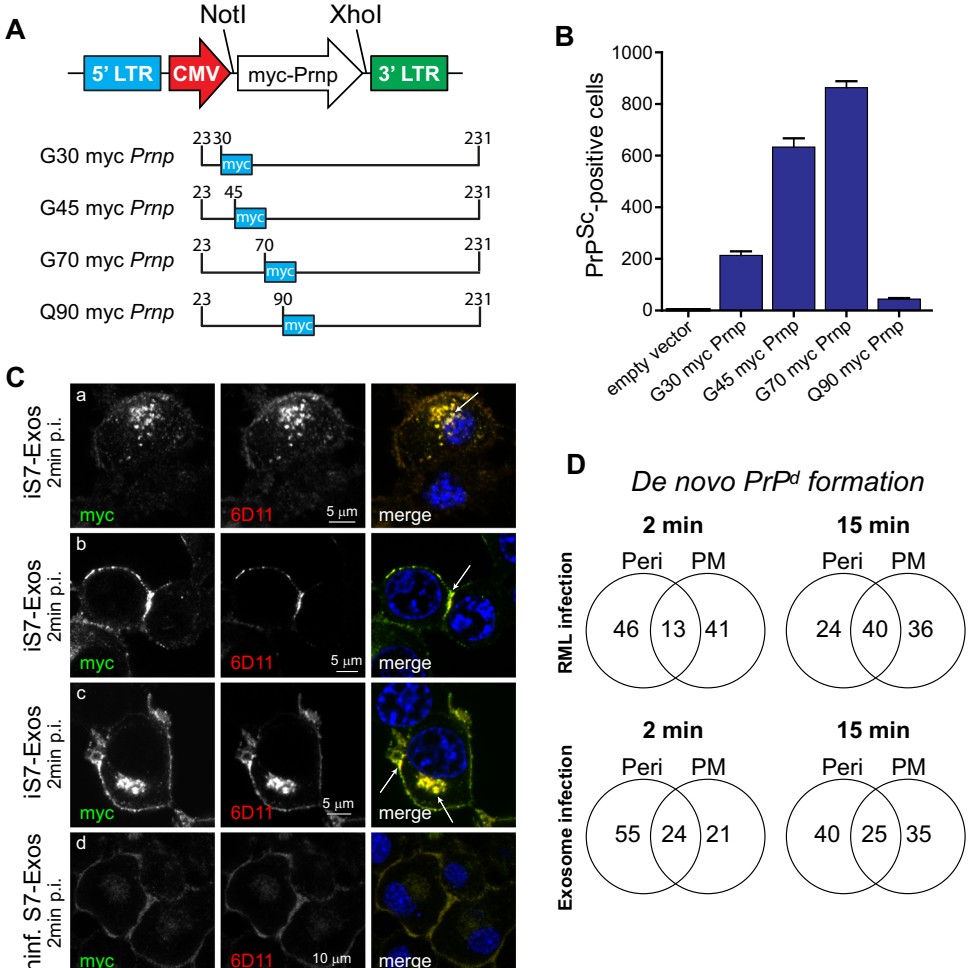

**Fig. 6 | Prion infection leads to PrP conversion at two distinct cellular locations.**
**A** Cloning of N-terminal versions of myc-tagged *Prnp* into the retroviral vector pLNCX2. The position of insertion of the myc tag is depicted. *Prnp* chimera were expressed in PK1-10 Si8 *Prnp* double-knockdown cells (see "Methods" section). **B** Comparative susceptibilities of myc-tagged *Prnp* expressing cells after infection with a 10⁻⁴ dilution of RML prions (titre: 10^8.4 LD50 units/g). An empty vector confirms that cells transcriptionally silenced with hairpins against the 3'UTR of *Prnp* are refractory to prion infection. Data represents one experiment with 12 repeats. **C** G70 PrP-expressing cells, fixed at 2 min following infection with infected exosomes were colabelled with AF488-conjugated anti-myc and the discriminatory 6D11 mAbs. Detection of *myc* PrP^d at distinct cellular locations is depicted by arrows in merged images: perinuclear (**A**), plasma membrane (**B**), perinuclear and plasma membrane (**C**) and mock-infected control cells (**D**). **D** Identifying the cellular sites of de novo conversion. Detection of de novo converted PrP in G70 PrP expressing cells at 2 min and 15 min after infection with RML and exosomes at perinuclear sites (Peri), the plasma membrane (PM), or both sites as represented by Venn diagrams in percent of total. At least 50 myc PrP^d positive cells were analysed per condition by three investigators. Data represent the percentage occurrence of PrP conversion at the specified cellular loci. Source data are provided as a Source Data file.

We provide evidence that PrP conversion occurs within minutes after infection at two distinct cellular locations, the plasma membrane and an unidentified perinuclear site. This result accommodates opposing views on the primary sites of PrP conversion[6,9,13], but also supports the notion of a fast rate of PrP conversion[13]. We think that labelling of cells under non-denaturing condition may explain why intracellular PrP conversion was not observed in the Goold study[13]. Our study provides two further important aspects of seeded PrP^d aggregation.

Firstly, despite the fast rate of PrP conversion, formation of FL-PrP^d aggregates was not detected prior to 24 h after infection. This suggests (i) that the formation of truncated PrP species is energetically favoured and (ii) that the infectious seeds that trigger the fast formation of TR-myc PrP^d at both cellular conversion sites are N-terminally truncated. Notably, non-fibrillar amorphous oligomers are thought to represent the most infectious seeds for prion propagation[53,54], although their state of proteolytic processing remains unknown. While the presence of truncated versions of PrP^Sc and PrP^c in prion diseases

has been widely reported[55–59], this study provides insight into the formation and the molecular relationship between truncated and full-length versions of PrP^d. In extension to current models of PrP^d formation (for review see ref. 60), we propose that the conformational transition from PrP seeds to fibrils is favoured by neutral pH conditions and can be blocked by retaining PrP^d seeds at an acidic luminal pH.

Secondly, the frequency of de novo PrP conversion is much lower than previously suggested[13], as verified by correlative imaging and prion titre output methods for various contact times. While de novo conversion was observed at a frequency of only 0.1%, 0.2% and 0.6% for a contact time of 2 min, 15 min and 4 h, respectively, prion titres increased up to 80-fold when cells were serially passaged for 2 weeks. This suggests that cell-to-cell transmission is a crucial cellular mechanism that underpins prion propagation.

We provide evidence that PrP^d segregates into synaptic and large-dense core vesicles of the regulated exocytosis pathway. Notably, peptide hormones and neuropeptides are thought to be sorted into immature secretory granules by virtue of protein aggregation[61–64],

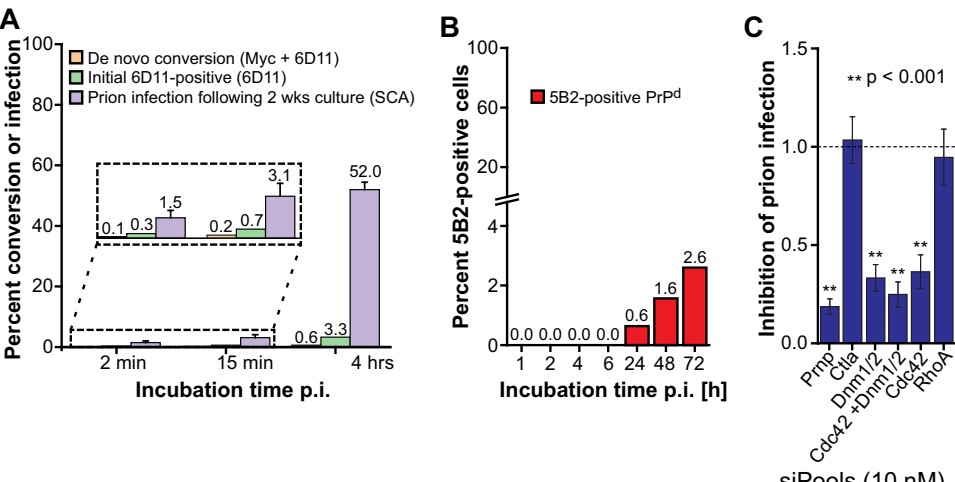

**Fig. 7 | De novo PrP conversion precedes the formation of FL-PrPd and is inhibited by transcriptional silencing of Cdc42 and dynamins. A** Quantitative analysis of contact times against the proportion of cells with evident de novo PrP conversion (Myc + 6D11) and initially infected (6D11) cells, respectively. G70 PrP-expressing cells were infected with RML for the specified times (post infection: p.i.), gently washed to remove the inoculum, fixed and labelled. For comparison, the proportion of infected cells after two weeks of tissue culture following exposure with prions at the specified contact times, data reblotted from Fig. 5G, are shown (violet bars). Mock-infected background-corrected data from 81 frames of about 60 cells per frame were scored as specified in "Methods" section. **B** Relationship between cell contact times after RML infection and the detection of 5B2-positive FL-PrP$^d$. G70 PrP-expressing cells were infected with RML brain homogenate at a $10^{-4}$ dilution for the time periods specified above. Cells were subsequently fixed and labelled with 5B2 and 6D11, followed by fluorescence conjugated secondary antibodies. As above, the number of double-positive cells was scored. **C** Inhibition of prion infection by gene silencing of dynamins and Cdc42. G70 myc *Prnp*-expressing cells were transfected with siPools against the specified targets, followed by infection with exosomes. Levels of prion infection were determined from 4 independent experiments with at least 24 replicates per gene target by SCA and expressed as mean values ± SEM, relative to cells transfected with non-targeting control (1.0 ± 0.013). For statistical analysis Kruskal–Wallis test with Dunn's multiple comparisons was conducted (**$p < 0.001$). For gene names see Supplementary Table 5. Source data are provided as a Source Data file.

suggesting that prions may be sorted into the secretory pathway by default. Importantly, the self-assembly of functional amyloids is reversible and more information on the reversibility of the amyloid-like aggregation state of peptide hormones and neuropeptides[62] will be critical to understand proteopathic seeds. Our finding that PrP$^d$ aggregates are detected in synaptic vesicles may further provide an important link between prion propagation and pathogenesis, since synaptic dysfunction is considered an early pathogenic process in neurodegeneration[65].

## Methods
### Antibodies and reagents
The antibodies used in this study were summarised as follows (clone, company, catalogue number). Unless otherwise stated, antibodies were used at a 1:1,000 dilution. Mouse monoclonal anti-PrP (8H4, Sigma, P0110), rat monoclonal Lamp1 (1D4B, Santa Cruz Biotechnology (SCB), sc-19992), mouse monoclonal anti-PrP (5B2, SCB, sc-47730, used at a 1:500 dilution), mouse monoclonal anti-PrP (AH6, SCB, sc-69896), mouse monoclonal anti-PrP (8B4, SCB, sc-47729), mouse monoclonal anti-Syp (7.2, Synaptic Systems, 101 011), rabbit polyclonal anti-Vamp4 (Synaptic Systems, 136 002), rabbit polyclonal anti-Chga (Synaptic Systems, 259 003), rabbit polyclonal anti-phospho FAK (Tyr925, Cell Signaling Technology, 3284), rabbit polyclonal anti-beta Cop (Thermo Fisher Scientific, PA1-061), rabbit polyclonal anti-Scg2 (Abcam, ab12241), mouse monoclonal anti-PrP (7D9, Abcam, ab14219), rabbit polyclonal anti Col4 (AbD Serotech, 2150-1470), rat monoclonal anti-CD29 (9EG7, BD Biosciences, 553715), mouse monoclonal anti-GM130 (35, BD Biosciences, 610822), mouse monoclonal anti-PrP (6D11, BioLegend, 808002), mouse monoclonal anti-PrP (Saf32, Cayman, 189720), rabbit monoclonal anti-PDI (C81H6, Cell Signaling Technology, 3501 S), rabbit monoclonal anti-Eea1 (C45B10, Cell Signaling Technology, 3288), rabbit polyclonal anti-Lc3a (Cell Signaling Technology, 4599), rabbit polyclonal anti-GFAP (DAKO, Z0334, used at a 1:10,000 dilution), Mouse monoclonal anti-PrP (3F4, Merck,

MAB1562), mouse monoclonal anti-PrP (MAB5424, Merck, MAB5424), mouse monoclonal anti-myc, AF 488-conjugated (9E10, Merck, 16-308), rabbit polyclonal anti-Syp1 (Millipore, AB1543), mouse monoclonal anti-Snap25 (SP14, Millipore, MAB331), rabbit polyclonal anti-Gorasp2 (Grasp55) (Proteintech, 10598-1-AP), mouse monoclonal anti-PrP (AG4, TSE Resources Centre, TSE RC, RC 059), mouse monoclonal anti-PrP (GE8, TSE Resources Centre, TSE RC, RC 061), ICSM18 and ICSM35 (UCL Institute of Prion Diseases), chicken polyclonal anti-Map2 (Abcam, ab5392). The reagents in this study were summarised as follows (company, catalogue number): Dynasore (Cambridge Bioscience, 1900-25), Pitstop 2 (Generon, AOB3600-25), Bafilomycin A1 (Insight Biotechnology, BIV-1829-250), Dynole (Insight Biotechnology, sc-362731), Cytochalasin D (Cambridge Bioscience, B1176-5), Cell Titre Glo luminescent cell viability assay (Promega, G7571), Lipofectamine RNAimax transfection reagent (Thermo Fisher Scientific, 13778150), Pierce LDH cytotoxicity assay (Thermo Fisher Scientific, C20301), Amplex Red assay (Thermo Fisher Scientific, A12216), One Shot TOP10 chemically competent E.Coli (Thermo Fisher Scientific, C404010).

### Cell lines and tissue culture
Neuro2a (N2a, CCL-131) cells were purchased from American Type Culture Collection (ATCC). Prion-infected and uninfected mouse neuroblastoma N2a-derived cell lines were maintained in OptiMEM, containing 10% foetal calf serum (FBS) and 1% penicillin/streptomycin (OFCS). The highly prion-susceptible cell line S7 was derived from N2aPK1 cells as previously reported in Marbiah et al.[66] and was split twice weekly for cell maintenance. *Extended TC protocol*: Where S7 cells were grown for extended culture times (Fig. 1C), the serum concentration was reduced by 50%, while the culture medium was changed daily from 3 days of culture. Under these conditions, the build-up of fibril-like PrP$^d$ aggregates at the plasma membrane is observed from three days onwards. *Standard TC protocol*: Daily replacement of ½ the conditioned medium with fresh OFCS from 3 days of culture was alternatively used for extended tissue culture times where specified (Figs. 2C and 3A).

## Primary neuronal cultures

Primary cortico-hippocampal cultures were prepared from embryonic e17 FVB mouse brains essentially as described previously[67]. These cultures contain GFAP-positive prion-susceptible astrocytes which, when infected with RML prions, form 5B2-positive PrP$^d$ fibrils (Fig. 1H). To culture neuronal monocultures (Fig. 1I), primary hippocampal cultures were treated with 1 μM AraC at 6 days after isolation. Following infection with RML prions, PrP$^d$ fibrils were observed after three weeks (Fig. 1I). Primary neuronal cultures were maintained in Neurobasal medium, supplemented with 2% (v/v) B27, 0.25% (v/v) GlutaMAX and 1% penicillin/streptomycin (Pen/Strep). Where primary neuronal cultures were maintained up to four weeks, half the medium was replaced with fresh medium twice a week. Primary hippocampal cultures were conducted essentially as reported in Ramsberger et al.[68].

## Safety and ethical declaration

All experimental procedures involving mouse-adapted prion strains were carried out in microbiological containment level 2 facilities with strict adherence to safety protocols and guidelines. All procedures involving animals were performed under approval and license granted by the UK Home Office (Animals (Scientific Procedures) Act 1986), project license number 70/9022 in compliance with UCL institutional guidelines and Animal Research: Reporting of In Vivo Experiments (ARRIVE) guidelines (www.nc3rs.org.uk/ARRIVE/). FVB mice were purchased from Envigo Ltd (London, UK) and were housed under specific pathogen-free conditions in a temperature and light-controlled environment (12 h light/dark cycles) and had free access to food and water.

## Prion infection of cultured cells

Crude brain homogenates (10% w/v) were prepared essentially as described by Hill et al.[69]. To improve homogeneity, samples were ribolysed as reported previously[70]. Briefly, crude brain homogenates were transferred to 2 ml microtubes (Sarstedt Ltd., Leicester, UK), containing zirconium beads, Protease Inhibitor Cocktail Set I (Pierce, Leicester, UK) and 25 U benzonase (Novagen, Madison, WI) using a Ribolyser (Hybaid, Cambridge, UK) at maximum speed for two cycles of 45 seconds. To infect prion-permissive cells, aliquots of $5 \times 10^4$ S7 cells/ml OFCS were plated into wells of 96-well plates. Sixteen hours later, cells were infected with serially 1:10 diluted ribolysed RML brain homogenates. Cells were typically infected with $10^{-4}$ or serial 1:10 dilutions of 10% brain homogenates (w/v) from prion-diseased mice at clinical disease with a titre of $10^{8.4}$ LD50 units/g. Following infection, cells were grown to confluence and split three times 1:8 to dilute out PrP$^{Sc}$-positive inoculum. At each split after reaching confluence, the proportion of PrP$^{Sc}$-positive cells was determined by the Scrapie cell assay (SCA) as specified in "Quantification of prion infection rates". If not otherwise specified, persistently prion-infected S7 cells were used in this manuscript. Prion-permissive S7 or myc-tagged S7 cells were freshly infected in Figs. 5F–J and 6C–D, 7A–C and Supplementary Fig. 8A–D.

## Minimum contact time for productive infection

To determine the minimum contact time for productive infection (Fig. 5F–I), prion-susceptible S7 cells were infected with clarified RML brain homogenates to reduce the amount of residual immune-positive inoculum, detected as background in the SCA and diluted out by serial cell passages[71]. RML brain homogenates were clarified by centrifugation at $300 \times g$ for 10 min. To monitor the amount of residual inoculum in clarified RML homogenates, prion replication-refractory Prnp$^{-/-}$ cells (see "Generation of Prnp knockout cells using Crispr-Cas9" below) were plated out in parallel experiments and infected with clarified RML brain homogenate alongside with S7 cells. Residual inoculum, detected in Prnp$^{-/-}$ cells was less than 10% of the total signal in prion-susceptible cells after the first passage and less than 1% after the

second passage, confirming that residual inoculum in clarified RML homogenates is rapidly diluted out during sequential passaging of cells. To assess the minimum contact time for productive infection, aliquots of $5 \times 10^4$ S7 cells/ml OFCS were plated into wells of 96-well plates and infected with a $1 \times 10^{-4}$ dilution of clarified RML brain homogenate the following day. After varying time points after infection, ranging from 2 min to 24 h (Fig. 5F), cells were carefully washed with PBS to remove the inoculum, grown to confluency and the number of infected cells was determined in subsequent cell passaged (Fig. 5 G, H).

## Quantification of prion infection rates

The number of prion-infected cells following infection with prion-containing media was determined using the SCA as described previously[71]. The SCA is based on the microscopic detection of proteinase K-resistant PrP (PrP$^{Sc}$) in prion-permissive cells in an automated manner using a Biomek FX liquid handling robot. Suspended cells (approximately 25,000 cells) were transferred onto ELISPOT (Multi Screen Immobilon-P, Millipore, UK) plates. ELISPOT plates were vacuum-drained and dried at 50 °C prior to storage at 4 °C until further processing. Where prion titres in persistently infected cells were determined, suspended cells were diluted in PBS 1:10 and approximately 2000 cells per well were transferred onto ELISPOT plates. The number of cells was determined by Trypan blue as specified below. The revelation and quantification of PrP$^{Sc}$-positive cells was done essentially as described in Schmidt et al.[72] using a Bioreader 5000-Eß system (BioSys, Karben, Germany). Changes in the prion steady-state levels of persistently infected cells following incubation with small inhibitory molecules or siRNA were also quantified by SCA. Where the percentage of infected cells on a population level was quantified in subsequent passages (Fig. 5G–I), the total number of cells was determined in parallel experiments by transferring ~1000 cells per ml onto ELISPOT plates. Subsequently, plates were stained with Trypan blue in cell lysis buffer (50 mM Tris HCl, 150 mM NaCl, 0.5% (w/v) sodium deoxycholate and 0.5% (v/v) Triton-X 100, pH 8.0) and counted with the Bioreader 5000-Eß to determine the total number of transferred cells.

We further used crude exosome fractions as a source of prion infectivity. To isolate exosomes, chronically infected S7 cells were grown to confluence in 15-cm Petri dishes. After removing the growth medium, cell layers were overlaid with fresh medium and conditioned medium was collected after 24 h for 4 consecutive days. Cell supernatants were stored at 4 °C until further processing. Combined supernatants were cleared from whole cells and cell debris by centrifugations at 300 x g for 5 min and at 4500 x g for 10 min, respectively. Clarified supernatants were then centrifuged at 32,000 x rpm for 2 h in a Beckman ultracentrifuge, equipped with SW 32 Ti swinging-bucket rotors (Beckman Coulter UK, High Wycombe, UK). Supernatants were discarded and pellets resuspended in residual medium. Concentrated exosome-containing suspensions were aliquoted and kept at −80C until use. Infectious titres of the exosome concentrates were determined using the SCA, essentially described previously[71]. Typically, one litre of cell supernatant from prion-infected S7 cells yielded 2-3 millilitres of concentrated exosomal pellet with an infectious titre of $10^{7.88}$ Tissue culture infectious units (TCIU)/ml.

## Cell curing assay

Effects of small molecule inhibitors on the prion steady-state levels were determined by the SCA while cytotoxicity was carefully monitored. Stock solutions of inhibitors were prepared in DMSO unless specified otherwise in the manufacturer's guidelines. Since 1:1,000 dilutions of DMSO did not affect the cell doubling rates and the prion steady-state levels of S7 cells, the concentration of DMSO-soluble inhibitor stock solutions was adjusted to allow dilutions of at least 1:1,000 or higher. To determine effects of small molecule inhibitors on prion steady-state levels, 300 μl aliquots of chronically infected S7 cells

were plated into wells of 96-well plates at a concentration of $5 \times 10^4$ cells per ml OFCS. After 16 h, cells were incubated with serially diluted inhibitors. After 3 days, cells were resuspended, diluted 1:10 in PBS and 100 µl of the cell suspension was transferred onto ELISPOT plates (Cat#. MSIPN4550, Merck Millipore Ltd, Tullagreen, Ireland) and prion titres were determined. Data in Supplementary Table 4 represent mean values of at least three independent experiments and are normalised to vehicle (DMSO)-treated cells.

### Determination of cell viability and cytotoxic thresholds

Cytotoxicity was assessed by measuring LDH release (Pierce LDH cytotoxicity assay kit, Thermo Fisher Scientific) and ATP levels by Luminescent Cell Viability Assay (Cell Titre Glo, Promega). To take experimental variation into account, cytotoxic effects were scored positive, when a threshold of 10% was exceeded, when compared to mock controls. Since cytotoxicity of small molecule inhibitors is highly correlated with the cell density, we determined cytotoxic thresholds of inhibitors for distinct cell densities (see Supplementary Table 3). In the ATP assay, luminescence values of inhibitor-treated cells were normalised with those of mock-treated cells and changes in ATP levels were recorded. In the LDH assay, the levels of toxicity were calculated as below,

$$\% cytotoxicity = \frac{LDH\ activity\ (treated) - LDH\ activity\ (unspecific)}{maximal\ LDH\ activity - LDH\ activity\ (unspecific)} \times 100$$

where unspecific LDH activity represents absorbance values of cells treated with 10% water (v/v) and maximal LDH represents absorbance values following cell lysis.

To determine the effects of methyl-β-cyclodextrin on cell viability, cholesterol and $PrP^d$ levels, chronically infected S7 cells were plated into 96-well plates (Cat#: 3505,Scientific Laboratory Supplies Ltd, Nottingham, UK) at a concentration of $5 \times 10^4$ cells per ml OFCS. After 16 h, cells were incubated with methyl-β-cyclodextrin (Cat#: Sc-215379, Santa Cruz Biotechnology, Heidelberg, Germany), dissolved in culture media without FBS, at concentrations specified in Fig. 3D. To determine the cellular cholesterol levels, cell layers were washed with PBS and lysed in 100 µl TE buffer (10 mM Tris and 1 mM EDTA, pH 8.0) per well. Aliquots of 50 µl lysates were transferred into black 96-well microplates (Cat#: 655077, Greiner Bio One Ltd, Gloucestershire, UK) and cholesterol levels where determined using the Amplex red assay (Cat#: A12216, Thermo Fisher Scientific, Loughborough, UK) according to the manufacturer's specification. While cholesterol levels were not normalised for cell numbers in Fig. 3D, the effects of cholesterol lowering on the level of colabelling between 5B2 and 6D11 (Fig. 3E, F) were determined for non-toxic mβCD concentrations (<1 mM) only. Cell viabilities were analysed by CellTiter-Glo luminescent cell viability assay (Cat#: G7571, Promega, Southampton, UK).

### Processing of cells for imaging

For immunolabelling, cells were processed as specified previously[66]. Briefly, cells were fixed with 3.7% formalin (Thermo Fisher Scientific, Loughborough, UK) in PBS for 12 min. Unless otherwise specified, cells were washed with chilled acetone for 1 min. After rehydration with PBS, cells were incubated with 3.5 M guanidine thiocyanate (GTC, Melford) for 10 min. Cells were washed at least five times with PBS prior to labelling with primary antibodies, diluted into Superblock (Thermo Fisher Scientific)/PBS (1:4), supplemented with 10% Pen/Strep and incubated at 4 °C overnight. After washing with PBS, cells were labelled with highly cross-adsorbed fluorescence-conjugated secondary antibodies (AffiniPure, Jackson ImmunoResearch) and incubated at 4 °C overnight. Cells were imaged after a final wash with PBS.

### Image acquisition using laser-scanning microscopy

Images were acquired with a Zeiss LSM710 laser-scanning microscope, equipped with a ×63 objective (1.4 oil, Plan-Apochromat) using immersion oil Immersol 518 F (Carl Zeiss, Cambridge, UK). Cells were fixed and treated as described above. Cells were labelled with fluorescence-conjugated secondary antibodies (AffiniPure, Jackson ImmunoResearch) and Alexa Fluor 488 and Rhodamine Red-X fluorescence was measured using a 460-540 nm and 565–640 nm bandpass filters, respectively, following excitation with an argon laser at 488 nm and a diode-pumped solid state laser at 561 nm, respectively. To unambiguously distinguish $PrP^d$ deposits at distinct cellular sites, i.e. the plasma membrane, perinuclear regions and the extracellular matrix (ECM), we conducted serial z-stacks and normalised the focal planes using ECM-resident focal adhesion kinase (FAK). Notably, in-focus detection of FAK, labelled with an anti-FAK antibody (Tyr925, Cell Signaling Technology, 3284) coincides with in-focus detection of PrP and was arbitrarily denoted "zero µm" as a focal reference level with an estimated standard deviation of ± 0.2 µm. At least 3 independent experiments were conducted for all immunofluorescence experiments and representative images are shown.

### Structural illumination microscopy

To image $PrP^d$ fibrils at the plasma membrane and ECM, cells were grown for extended culture times (5-7 days) with daily medium changes from day 4. Cells were processed as specified in "Processing of cells for imaging" and labelled with anti-PrP antibodies 5B2 and 6D11, followed by highly cross-adsorbed secondary antibodies, anti-mouse IgG1-Rhodamine and anti-mouse IgG2a-Alexa Fluor 488 (AffiniPure, Jackson ImmunoResearch), respectively. Images were acquired by Structured Illumination Microscopy (SIM) using a Zeiss ELYRA PS.1 microscope with Plan-Apochromat DIC M27 63X objective and 1.4 immersion oil. Images were acquired using 5 phase shifts and 3 grid rotations for the 561 nm and 488 nm lasers at a 5-10% output. Images were acquired using a sCMOS (pco.edge sCMOS) camera. Channels were aligned using a slide with multi-coloured beads (100 nm, Tetra-Speck microspheres T7279, Thermo Fisher Scientific) with the same image acquisition settings. After SIM reconstruction and channel alignment, images were concatenated into a single image stack. Images were processed with ZEN black version 11.0.2 software (Zeiss).

### Quantifying relative $PrP^d$ fluorescence intensities

Chronically prion-infected and uninfected cells were seeded into 8-well chamber slides and grown to confluence prior to processing as described in "Processing of cells for imaging". Quantitative analysis of $PrP^d$ aggregates at perinuclear and extracellular sites was conducted following double-labelling of cells with anti-PrP antibodies 5B2 and 6D11 overnight at 4 °C. After washing with PBS, cells were labelled with highly cross-adsorbed isotype-specific fluorescence-conjugated secondary antibodies (AffiniPure, Jackson ImmunoResearch) for 12-24 h. Randomly selected fields with 20 to 40 cells were analysed from at least three replicate experiments using Volocity. To quantify fluorescence intensities of $PrP^d$ aggregates, threshold levels of $PrP^c$ fluorescence were determined in uninfected cells and used as baseline levels. Intersection denotes colabelled areas where pixels are detected in more than one fluorescence channel. To determine levels of colocalisation of 6D11-positive $PrP^d$ with intracellular markers at perinuclear sites, stacks of 10-15 images in z-direction above the basement membrane were recorded with a step size of 0.5 µm and levels of colocalisation between $PrP^d$ and organelle markers analysed using imaging software Volocity. Data of at least 20 single cells were analysed using the Volocity cropping function. Cells were double-labelled with primaries for 24 h and washed with PBS. Cells were subsequently labelled with highly cross-adsorbed fluorescence-conjugated secondary antibodies (AffiniPure, Jackson ImmunoResearch) for 24 h. Levels of colocalisation were determined by Pearson's correlation with Costes

threshold correction[73] according to the specification of Perkin Elmer, the manufacturer of Volocity. To examine whether mβCD extracts PrP$^c$ from the plasma membrane, we treated confluent layers of uninfected cells with 0.4 to 2 mM mβCD for 16 h at 37 °C. Cells where then fixed and labelled with anti-PrP mAb 8H4, followed by AF488-conjugated secondary antibody. Prior to imaging, cells were permeabilised with 0.04% TX-100 and labelled with DAPI for 15 min to reveal nuclei. Changes in fluorescence intensities of total PrP$^c$ were determined as described above.

### Validating the specificity of anti-PrP antibodies

To validate the target specificity of anti-PrP antibodies, we generated *Prnp* knockout cells. Briefly, the *Prnp* open reading frame (ORF) in N$_2$a cells was deleted by CRISPR-Cas9 according to Ran et al.[74]. Single guide RNAs (sgRNAs) were selected using the web-based "sgRNA designer" from the Broad Institute and cloned into the Addgene vector pX459 (Ref: 62988) as specified below. Out of four sgRNAs tested for their efficacy of *Prnp* knockdown, sgRNA-*PrnP*−69, which recognises the complementary *Prnp* sequence 53-ATGTCGGCCTCTGCAAAAAG-72 and enables a Cas9-mediated double-strand break at position 69 was found to be superior. To synthesise sgRNA-*PrnP*−69, two oligonucleotides (fw-5′-CACCGATGTCGGCCTCTGCAAAAAG-3′ and rev-5′-AAACCTTTTTGCAGAGGCCGACATC−3′), flanked by the NGG PAM sequence with overhangs (underlined) were annealed and cloned into the BpI-digested vector pX459 using T4-polynucleotide kinase ligase (New England Biolabs, Hitchin, UK) according to the specification of the manufacturer. To generate *Prnp* knockout cells, pX459/gRNA-*PrnP*−69 was transfected into N2a cells using Lipofectamine LTX/Plus (Thermo Fisher Scientific) as recommended by the manufacturer and cells were selected with 2 μg puromycin per ml of medium.

### Cloning of Myc-tagged PrP variants

Double stranded DNA molecules of the full-length mouse *Prnp* gene (Genbank NP_001265185.1) with Myc tag (aa's EQKLISEEDL), inserted at positions p.Gly45, p.Gly70 and p.Glu90 were synthesised by GeneArt (Thermo Fisher Scientific, UK). Corresponding *Prnp* fragments (10 ng) were Gibson assembled by overlap extension using GeneArt Gibson Assembly HiFi MM (Thermo Fisher Scientific, Regensburg, Germany) into the linearised pLNXC2 vector (25 ng) (Addgene, Teddington, UK). Vector and inserts at a 1:1 molar ratio were incubated for 1 h on a thermocycler (PCRmax, East Sussex, UK), transformed at 37 °C for 1 h in One Shot TOP10 chemically competent E.Coli cells (Thermo Fisher Scientific, UK) and then plated onto LB agar plates, containing 50 μg/ml ampicillin (Sigma, UK). After overnight culture, positive plasmids were digested with restriction enzymes and analysed by gel electrophoresis to confirm that the insert is in the correct position prior to Sanger sequencing (Eurofins Genomics, Ebersberg, Germany). Mouse *Prnp*, tagged at Gly30 with myc, containing Not1 and Xho1 restriction sites was obtained from GeneArt Life Technology. Cloning into the linearised pLNXC2 vector was done as described above. Transformation of competent JS4 cells was carried out in presence of ampicillin overnight. Colonies were screened by PCR to identify those with inserts. Plasmid DNA was extracted and digested and the sequence of Gly30-myc *Prnp* was confirmed by Sanger sequencing.

For retroviral production, HEK-293 derivative Phoenix cells were seeded at a concentration of $1 \times 10^6$ cells per 10 cm plate in producer cell growth media (DMEM, 10% FCS, 1% Penicillin-Streptomycin, 1% Glutamine), 24 h before transfection with 3 μg *Prnp* Myc positive constructs and 12 μl Fugene HD (Promega, UK) as per manufacturer's instructions. After 24 h the medium was replaced with fresh growth medium and supernatants were collected at 2 days after transfection. After filtration of supernatants through 0.45 μm filters (Acrodisc Syringe filters, Cat# 4219, PALL Europe, Portsmouth, UK) and supplementation of polybrene (8 μg/ml), PK1-10 Si8 double-knockdown cells[13] were transduced and incubated at 37 °C for 4−5 h before replacing with complete media (Optimem, 10% FCS, 1% Penicillin-Streptomycin). In the next split the cells were cultured under subsequent selection markers G418 (200 μg/ml, ThermoFisher, UK) and Puromycin (2 μg/ml, Invitrogen, UK) and passaged at least 5 times in presence of antibiotics. A control with empty pLNXC2 vector was set up in parallel experiments to monitor the efficacy of antibiotic selection.

### Assessing prion susceptibility of myc-tagged *Prnp*

To investigate the permissiveness of myc-*Prnp*-expressing cell lines at position p.Gly30, p.Gly45, p.Gly70 and p.Glu90 to mouse-adapted prion strains, cells were plated out in 96-well culture plates at a concentration of $1 \times 10^5$ cells/ml OFCS. The next day, the culture medium was replaced with 300 μl OFCS, containing $10^{-4}$ dilutions of RML (10%, w/v) and CD1 (10%, w/v) for mock infections and left until cells were sub-confluent to undergo a split. At passages 3 and 4, respectively, 100 μl of a 1:10 dilution of cells, resuspended in PBS were transferred onto wells of 96-well Elispot plates (PerkinElmer, UK) to determine the number of PrP$^{Sc}$-infected cells using Scrapie Cell Assay, essentially as described previously[71].

### Infection and labelling of myc PrP-expressing cells

For immunofluorescence-based detection of myc-tagged PrP$^d$ aggregates, 500 μl aliquots of $5 \times 10^4$ cells per ml OFCS were seeded into 8-well chamber slides (Thermo Fisher Scientific, UK). After 16 h, cells were infected with prion-containing media, i.e. RML-prion and uninfected CD1 homogenates or exosome-enriched cell supernatants from persistently infected or uninfected S7 cells. Twenty four hours following cell plating, the culture medium was replaced with 500 μl OFCS, supplemented with a $1 \times 10^{-4}$ dilution of 10% brain homogenates (RML and CD1 mock) or 30 μl exosomes per ml medium. To examine de novo PrP conversion following prion infection, myc PrP-expressing cells were grown for 3 days prior to infection with brain homogenates or exosome-enriched supernatants as above. Prion-containing solutions were removed by washing with PBS after various time points as specified in Figs. 6 and 7 and cells were fixed with 4% formalin for 15 min at room temperature. Cells were then incubated with ice cold acetone for 1 min, followed by denaturation with 3.5 M GTC for 10 min at room temperature. The cells were incubated with anti-PrP mAb 6D11 (1:10,000 BioLegend) and directly conjugated mAb AF488 anti-myc 9E10 (1:1000 Millipore, UK) in Superblock/PBS antibody diluent containing 5% FBS and left overnight at 4 °C. AffiniPure Rhodamine Red goat anti-mouse IgG2a subclass secondary antibody was used at a 1:1,000 dilution overnight at 4 °C. All images were captured on a Zeiss LSM 700 confocal microscope, equipped with a 63 × oil objective at the same acquisition settings.

### Identification of de novo PrP conversion sites

To investigate the cellular sites of de novo PrP conversion in G70 PrP-expressing cells, threshold levels for the detection of myc-tagged PrP$^c$ were first determined using mock-infected cells and PMT gains were adjusted to exclude background fluorescence. The same settings were then applied to examine the sites of de novo PrP conversion in freshly prion-infected cells at the time points specified in Fig. 6C, D. All cells were co-labelled with the discriminatory mAb 6D11 to increase the confidence of scoring de novo infected cells. Only double-labelled cells were scored as shown in representative images (Fig. 6C). All experiments were scored independently by three investigators and at least 50 myc-PrP$^d$ positive cells were analysed for each condition.

### Quantifying infected myc *Prnp* cells

To determine the proportion of infected G70 PrP-expressing cells at early time points after prion infection, three separate automated scans of 9 × 9 tiles with approximately 40 cells per tile were set up for every condition with a 5% overlap among tiles. Background levels were adjusted as specified above. For cell phenotypic analysis, cells

double-positive for anti-myc and 6D11 were manually scored based on fluorescence intensity using Zen black 2.3SP1. The total number of cells was determined by scoring DAPI-positive nuclei using Volocity software. To determine the percentage of 5B2-positive cells (Fig. 7B), tile scans of 9 × 9 tiles were acquired and cells were scored based on the detection of anti-5B2-positive $PrP^d$ at the plasma membrane.

## Transcriptional gene silencing

Transient knockdown of gene expression was conducted by reverse transfection of cells using siRNA pools (siTOOLs Biotech GmbH, Martinsried, Germany) which contain up to 30 siRNAs to achieve maximal transcript coverage and highly efficient gene knockdown. Briefly, siRNA pools were reconstituted with nuclease-free water to a stock concentration of 10 μM. Two μl siPool and 6 μl RNAimax transfection reagent were added to 42 μl FBS-free OPTIMEM and incubated to form a complex. After 5 min, 950 μl OFCS were added to the RNA complex and 1 ml of $1.33 \times 10^5$ cells (reverse transfection) was added for a final concentration of $6.7 \times 10^5$ cells/ml.

## Quantifying gene knockdown levels

We used real-time quantitative PCR (RT-qPCR) to determine the level of gene expression in cells, following transcriptional silencing. Briefly, total RNA was isolated 72 hrs after transfection of cells with siPools and a non-targeting control using Direct-zol RNA Mini-prep kit (Cambridge Bioscience, Cambridge, UK). Genomic DNA was removed during RNA isolation as per manufacturer's instructions. RNA purity and concentration was measured spectrophotometrically (Nanodrop 2000, Thermo Fisher Scientific, USA). Total RNA (500 ng) was used to synthesise the first strand complementary DNA (cDNA) using QuantiTect Reverse Transcription kit (Qiagen, Manchester, UK). All real-time qPCR reactions were carried out with QuantStudio 12 K Flex (Applied Biosystems, Cheshire, UK) with the following cycling parameters: 50 °C, 2 mins; 94 °C, 15 min; 40 cycles at 94 °C, 15 s; 60 °C, 1 min. Samples were set up in triplicates in 10 μl reactions containing QuantiTect SYBR Green (Qiagen), 1x QuantiTect customised Primer Assay (lyophilised in TE, pH 8.0), 25 ng of cDNA made to a final volume with ddH20. Gapdh and Actb were used as endogenous controls for normalising target gene expression levels. Data acquisition of the fluorescent signal was performed at the end of the run to assess the expression of mRNA by evaluating threshold cycle (CT) values. Double delta CT calculations were measured as logarithm and then converted to fold change of 100, after untreated control levels were subtracted from target levels.

## Cell lysis and proteinase K digestion

Uninfected and prion-infected S7 cells were grown to confluency in 6-well plates and treated with 0.5 mM BafA1 or DMSO (0.05%) for 1 h. Cells were washed with OFCS and further incubated for 16 h. Cells were then lysed in RIPA buffer (50 mM Tris, 150 mM sodium chloride, 0.5% sodium deoxycholate, 1% Triton (v/v), pH 7.4) on ice for 15 min. Lysates (100 μg total protein) were incubated with 10 μg/ml proteinase K (PK, Merck, 539480) for 30 min at 37 °C. Protein digestion was stopped with 10 mM 4-(2-aminoethyl) benzenesulfonyl fluoride hydrochloride (AEBSF) for 10 min on ice. Lysates were mixed with NuPAGE LDS sample buffer (4X) (Thermo Fisher Scientific, NP0007) and boiled for 10 min.

## Western blotting

Fifty micrograms of total protein or protein marker SeeBlue2 (Invitrogen) were separated on 12% Bis-tris gels (NuPage, Thermo Fisher Scientific, UK) at 200 V for 90 min. Proteins were then transferred onto methanol-activated PVDF membranes (Immobilon-P, Millipore) at 15 V overnight. Blots were incubated in blocking buffer (5% skimmed milk powder (Sigma Aldrich), 0.05% Tween-20, PBS) for 1 h at room temperature. Blots were then incubated with 0.15 μg anti-PrP antibody ICSM35 (D-Gen Ltd, MRC Prion Unit at UCL, London, UK) per ml

blocking buffer for 1 h. After washing, blots were incubated with AP-conjugated anti-mouse IgG (Fab) antibody (A2179, Sigma Aldrich) for 1 h at room temperature. After at least five washes in PBST, blots were incubated twice in activation buffer (20 mM Tris, 1 mM MgCl2, pH 9.8) for 5 min. Blots were incubated with Tropix CDP-Star chemiluminescent substrate (CDP-Star, Thermo Fisher Scientific, T2147) and revealed on Amersham Hyperfilm ECL (Cytiva, 28906836). To control for equal loading, blots were stripped with Restore Plus Western blot stripping buffer (Thermo Fisher Scientific, 46430) and reprobed with anti-mouse actin antibody (Sigma Aldrich, A5441).

## Immunoprecipitation

Prion-infected S7 cells were grown to confluency on 15 cm dishes for 5 days with half medium changes on days three and four. Cells were washed with PBS, 10 ml sterile distilled water was added and cells were incubated for 15 min at room temperature, followed by thorough resuspension under microscopic control. The cells were discarded and 1.5 ml ice-cold RIPA buffer was added to the decellularised ECM, followed by vigorous resuspension. The lysate was collected and centrifuged at $1000 \times g$ for 5 min at 4 °C. In parallel experiments confluent dishes of prion-infected S7 cells were washed with PBS and lysed with 1.5 ml of RIPA buffer for 15 min. All lysates were centrifuged for 5 min at $1000 \times g$ at 4 °C. For immunoprecipitation, 1 μg anti-PrP antibody ICM18 (D-Gen Ltd, MRC Prion Unit at UCL, London, UK) and 50 μl μMACS Protein G Microbeads (Miltenyi Biotech, 130-071-101) were added to 300 μl lysates. Following incubation on ice for 30 min, immune complexes were separated magnetically on μ columns essentially as specified by the manufacturer Miltenyi Biotech. The immunoprecipitates were eluted from μ columns with boiling 1 x LDS sample buffer. Western blots were conducted as described in "Western blot". Following blocking in 5% milk for 1 h, membranes were briefly washed twice in PBST and incubated with biotin-conjugated ICM35 (D-Gen). After washing, membranes were incubated with avidin/biotin ABC complex (Vectastain ABC-HRP, Vector Laboratories, Peterborough, UK). All antibodies were incubated in 3% BSA/PBST for 1 h and revealed with ECL substrate SuperSignal West Pico PLUS Chemiluminescent Substrate (Thermo Fisher Scientific, 34579).

## Lysosome isolation

Lysosomes were isolated by Lamp1 immunoprecipitation according to Stahl-Meyer et al.[75] with minor modifications. Briefly, uninfected and prion-infected S7 cells were grown to confluency in 15 cm dishes and treated with 0.5 mM BafA1 or DMSO (0.05%) for 1 h. Cells were washed with OFCS and further incubated for 16 h. Cells were then washed with cold PBS, resuspended in 10 ml PBS and centrifuged at 500 x g for 5 min at 4 °C. Cell pellets were resuspended in isolation buffer (10 mM HEPES, 70 mM sucrose, 210 mM mannitol, pH 7.5) and centrifuged at 1000 x g for 5 min at 4 °C. This wash step was repeated once more. Cells were then lysed in 700 μl isolation buffer, supplemented with 2.5 U Benzonase/ml, 0.5 mM Tris(2-carboxyethyl)phosphine hydrochloride and 7 μl Halt Protease and phosphatase inhibitor (Cat# 78440, Thermo Fisher Scientific) by syringing the cells 15 times through a 25-gauge needle and centrifuged at 1500 x g for 10 min at 4 °C. The supernatant was carefully collected and centrifuged again at 1500 x g for 10 min at 4 °C. The protein concentration of post-nuclear supernatants was determined by Pierce BCA assay (Cat#: 23225, Thermo Fisher Scientific). Equal amounts of protein were incubated with 3 μl polyclonal Lamp1 antibody (ab24170, Abcam), pre-bound on 50 μl Miltenyi μMACS protein G beads (Cat# 130-071-101, Miltenyi Biotech) and shaken overnight at 4 °C. MACS columns were rinsed with isolation buffer and mounted onto a magnetic separator. The immunoprecipitate was captured on MACS columns, washed with isolation buffer and eluted with pre-heated SDS sample buffer. For on-column PK digestion, enriched lysosomes were incubated with 10 μg PK/ml

PBS. MACS columns were sealed with parafilm and digested for 30 min at 37 °C. Protein was eluted with pre-heated SDS sample buffer.

## Statistics

Data was analysed using SPSS, Graphpad Instat 3.10 and Microsoft Excel. The Kolmogorov-Smirnov test was applied to test whether experimental data were normally distributed. Normally distributed data was analysed by parametric tests, including Student's t-test and Analysis of variance (ANOVA) followed by post-hoc Bonferroni test. Unless otherwise stated, statistical tests were two-sided. In case of small sample sizes ( < 20) or data that did not fit the normal distribution, Kruskal–Wallis and Mann–Whitney $U$ test were employed, followed by the Dunn's multiple range test. Unless otherwise stated, a p-value < 0.01 was considered statistically significant. Where technical replicates were pooled across independent experiments, data was normalised within each data set. Boxplot elements in Figs. 2B and 5C are defined as following: The line within the box represents the median, the bounds of the box represent the interquartile range, and the whiskers represent the minimum and maximum values no more than 1.5 times the interquartile range. Where the minima and maxima lie outside the whiskers they are represented as dots.

## Reporting summary

Further information on research design is available in the Nature Portfolio Reporting Summary linked to this article.

## Data availability

Data supporting the findings of this study are presented within the article and supplementary information and are available from the corresponding author upon request. The data underlying Figs. 1G, 2B, 3D, F, G, 4G, J–K, 5A–C, E, G–J, 6B and 7A–C and Supplementary Figs. 5C, F, H–O and 7A–B are provided in the Source Data file. The composition of siPools and primer sequences of qPCR assay is proprietary information by siTOOLs Biotech GmbH and Qiagen, respectively. Source data are provided with this paper.

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

## Acknowledgements

This study was funded by the UK Medical Research Council (MRC) and by the UK Biotechnology and Biological Sciences Research Council (BBSRC); S. A. T. was supported by The Francis Crick Institute which receives its core funding from Cancer Research UK (FC001187), the UK Medical Research Council (FC001187). This research was funded in whole, or in part, by the Wellcome Trust (FC001187). For the purpose of Open Access, the author has applied a CC BY public copyright licence to any Author Accepted Manuscript version arising from this submission. We are grateful to Ms Parineeta Arora and Prof Parmjit Jat for providing PK1-10 Si8 double-knockdown cells and a Gly30-myc *Prnp* expression clone. We thank Christian Schmidt, Parvin Ahmed and George Thirlway for technical assistance with Automated Scrapie Cell Assays and the staff of the MRC Prion Unit at UCL Biological Services Facility for animal care and technical assistance. We are grateful to Prof John Collinge for helpful discussion.

## Author contributions

Conceptualisation, P.C.K. and S.A.T.; Methodology, J.M.R., M.P., H.H., A.B. and P.C.K.; Validation, J.M.R., M.P., H.H. and P.C.K.; Investigation, J.M.R, M.P., H.H. and P.C.K; Resources, J.M.R., M.P., P.C.K.; Writing – original draft, P.C.K.; Writing – review & editing, S.A.T., J.M.R., M.P. and P.C.K.; Visualisation, P.C.K.; Project administration, P.C.K.; supervision, P.C.K.; Funding acquisition, P.C.K.

## Competing interests

The authors declare no competing interests.
