## [Peer review file · Nature Communications]

nature portfolio

Peer Review FileReviewers' comments:

Reviewer #1 (Remarks to the Author):

In this study, the authors used monospecific anti-PrP monoclonal antibodies to distinguish between distinct types of PrPd found at different subcellular sites by confocal microscopy and SIM. As an experimental model, they use in most experiments a prion-infected neuronal cell line derived from N2a neuroblastoma cells infected with RML scrapie prions, and confirm some findings in primary astrocyte culture and an additional mouse-adapted scrapie prion strain, 22L. They determined that PrPd exists as full length FL-PrPd fibrils at the plasma membrane, and at intracellular sites as a truncated version, with the proteolytic processing occurring in acidic vesicles. The formation of FL-PrPd is slow and preceded by intracellular truncated PrPd. Truncated PrPd is cycled to the plasma membrane by vesicular transport. Using siRNA-mediate knock-down of pathway-specific proteins, they determined that prion propagation depends on dynamin and Cdc-42, implying a critical role for PrPd re-uptake in this process.

This is an interesting study providing additional information on the prion propagation process in neuronal cells. Parts of the study confirm previous work by Rouvinski et al., (2014), such as the occurrence of FL-PrPd strings at the plasma membrane, and the kinetics of formation with the need for extended culture times, and early work by Taraboulos et al (1992), describing N-terminal truncation of PrPSc in acidic vesicles, sensitive to pH-increasing compounds. Goold et al, (2011 and 2013) used a similar strategy of myc-tagged PrPC and found rapid conversion at the cell surface within 2 min of prion exposure and identified recycling pathways involved in PrPSc recycling to the cell surface.

Further comments:

- Fig 1H: in the results the authors refer to 'neuronal culture' but only show results for primary astrocytes – what was the result for neurons? It is interesting though that astrocytes show the ; same PrPd fibril-like aggregates, suggesting this is not typical for neurons; in this experiment, 22L was used for infection, for most other figures RML – what was the rationale, are strain differences expected in particular for the internalisation pathways– as Fehlinger et al (2017) described different routes of internalisation are used by RML and 22L to establish productive infection; it is suggested to include some discussion about the strain used and whether the authors expect similar results for all strains
- SAF32 recognizes cell surface and intracellular PrPd: the explanation of differences in the octarepeat region is not clear – is it not that 5B2 does not recognize intracellular PrPd because of N-terminal truncation – while possibly the SAF32 epitope is retained? Does SAF32 recognize N-terminally truncated PrPd eg by immunoblot?

- Fig.3E: from the significant reduction of 5B2 signal it is concluded that a block of intracellular PrPd to the plasma membrane inhibits the formation of FL-PrPd; however, first, cholesterol extraction would affect lipid raft integrity at the plasma membrane, know to reduce PrPSc formation – is lipid raft perturbation an alternative explanation for the lack of 5B2 signal? Second, cholesterol extraction can release GPI-anchored proteins. It is recommended to include a control for the release of GPI-anchored proteins from the cell surface, e.g. PrPC.

- Fig. 5: Arf1, Arf6 and Cav1 knock-down and effects on PrPd: it is commented in Table S5 that Cav1 is not expressed in the used cells; therefore, it should be omitted from the graph. The knock-down of Arf1 and Arf6 on the mRNA level is only around 40 – 50% efficient; the question arises whether this level of knock down is sufficient to affect PrPd, and how it translates into a reduction of protein levels, which could be even less efficient depending on the half life of the proteins. It is suggested to include analysis of protein levels for these proteins.

Reviewer #2 (Remarks to the Author):

The manuscript by Ribes et al characterizes specificity of various anti-PrP antibodies in imaging assays on fixed cultured cells, to identify those that can discriminate against native versus non-native PrPd conformations. With this knowledge, the authors seek to investigate de novo PrP conversion and

formation of fibril-like PrP aggregates in the periphery of cells as well as intracellularly. The study concludes that there are distinct pools of abnormal PrP that forms at the plasma membrane and also at perinuclear regions within minutes of infection with RML prions or with exosomes from prion cultures. Moreover, the authors conclude that there is differential N-terminal cleavage of PrPd and that the transport of intracellular PrPd occurs from inside the cell to the plasma membrane is active and not diffusive.

The characterization of the specificity of anti-PrP antibodies in the cell types investigated (N2a and S7 cells), using imaging is interesting and would provide a good resource for the prion field. Also, the data showing elongation of PrPd fibrils over time is convincing. However, there are major deficiencies in the manuscript. One of the main ones is the heavy or almost sole reliance on imaging data throughout the manuscript and lack of use of very well-established biochemical assays that are routinely used for studying prion conversion, etc. Also, major deficiencies are noted that relate primarily to insufficient or deficient or non-existent data that authors rely on (or not) to make the major conclusions made throughout the manuscript. They include all major conclusions of the study as outlined in their summary and discussion. Major points are noted below, but here is a summary: (1) PrPd is claimed to be in or associated with vesicles, data is colocalization analysis that is not apparent by visual inspection of the images provided and non-consistent with the Pearson coefficient provided for various comparisons; (2) PrPd is concluded to be proteolytically cleaved after treatment of prion-infected cells with BafA1, but no data is shown for proteolytic cleavage so the conclusions drawn about truncated PrP forms is totally speculative; (3) evidence that PrP conversion happens at two sites (plasma membrane and at perinuclear regions) is based on presence of anti-PrPd staining at these two sites 2 min after infection and beyond. But observation of PrPd at plasma membrane within this short period of time doesn't prove de novo conversion, but could be just initial binding of the infected prions to the cell surface. Moreover, the presence of PrPd at perinuclear regions could be explained by subsequent internalization of PrPd from the cell surface after initial infection, and not necessarily de novo conversion. These possibilities are not considered. Importantly and majorly, no assay is performed in this study showing conversion of PrP. I.e., no biochemical de novo conversion assays are used to show definitely de novo formation or conversion. However, assays for the study de novo conversion of PrP are well established in the prion field, and these would need to be done at different time points to discern de novo conversion versus persistent infection as in experiments in Fig. 6. These lacking does not provide confirmation that de novo infection is occurring at the time points suggested in the study. This is critical to do, as all the staining could be due to initial binding of externally-applied fibrils and accumulation of these fibrils over time. It is appreciated that this would be a high order proof, and thus it is agreed with the authors that this is the site of de novo PrP conversion has been so elusive to demonstrate and pin down. And still remains so; (4) conclusion claiming that PrPd is transported to the plasma membrane actively associated with synaptic and large-dense core vesicles is weak or not provided. First, colocalization of synaptic and dense core vesicle markers with PrPd is not convincing. Second, active transport is not shown. Live imaging of transport of these presumed vesicles would need to be done and it is a very feasible assay that would allow the rapid and fast visualization of moving PrPd vesicles. Merely using a static image showing presence of PrPd staining near the cell surface could be interpreted also as PrPd being localized there, but not transported there. There is no discussion as to how PrPd could get into the secretory pathway as claimed in the manuscript.

Moreover, there are several instances where conclusions reached on images that are not quantitated and where that does not show what authors indicate (e.g., Fig.3B on first instance in the manuscript, as noted in point 9 below). Furthermore, there are contractions in data, for example, reduction of endocytosis using dynasore and dynole or genetically by downregulation dynamin, is reported to lower PrPd steady state levels, but reducing clathrin does not. It is unclear and unexplored how some clathrin mediated endocytosis routes are affected but not others?

Considering all the points raised represent major issues, there would have to be substantial data collection to substantiate the conclusions reached in this manuscript.

Major points:

1. Imaging of staining of PrPd fibrils being made at the ECM (E.g., Fig 1C), using antibody against ECM-resident protein focal adhesion kinase (FAK) proteins is not provided. Because this is a key point to the study, this needs to be included.
2. Line 133: "In cultures from FVB wt mouse brains, fibril length continuously and significantly increased over three weeks, at mean fibril elongation rates of 190 nm per day, reaching lengths of up to 15 μ m, while no fibrils were detected at the periphery of astrocytes from brains of FVB Prnp^{-/-} mice (Fig. 2A, B)." It is unclear why fibril length is measured in the periphery of neurons in FVB wt mouse brains, but in the periphery of astrocytes in brains of FVB Prnp^{-/-} mice? It appears PrPd fibrils are only around astrocytes not primary neurons?
3. Fig.2C: in the co-labeling experiment, it is difficult to appreciate the degree of localization or not, as the figure is provided only as a merge of two channels. Separation of each channel in black/white (inverse could help), in addition to the merge would facilitate evaluation of the observations/ conclusions presented with these data.
4. Line 142: Data presented to suggest "that different aggregation states may co-exist during PrPd fibril formation" is based on light microscopy imaging. Biochemical characterization of these different PrPd aggregate states using antibodies used in these imaging studies would provide important orthogonal validation.
5. A very major deficiency in the manuscript with consequences to the rest of the manuscript starting with lines 165-185 is the following: The observation and conclusion that 5B2-positive aggregates in cells treated with BafA1 become visible due to blockage of proteolytic cleavage of full-length PrPd within less acidifying endolysosomes is not substantiated with data. First, there is no evidence provided that PrPd is inside or associated with endosomes (also see point 13 below). Second, evidence for proteolytic cleavage of PrPd is completely lacking so this cannot be concluded, therefore calling this fragment "truncated (TR) PrPd" is total speculation. Both light microscopy imaging as well as biochemical characterization of PrPd within endolysosomes and of proteolytic cleavage of PrPd in those endosomes (for example using endosome purification by sucrose gradient centrifugation) is required to substantiate these claims. BafA1 treatment alone does not indicate cleavage.
6. There are no experiments in the manuscript demonstrating that PrPd recognized by any of the antibodies tested are detecting "nascent" PrPd (versus non-nascent?). Thus, this claim as well as the one on point 5 above, i.e., lines 179-182 "These results suggest existence of two distinct aggregated PrPd species in prion-infected cells, nascent FL-PrPd which is associated with fibril elongation at the plasma membrane and a truncated (TR-) PrPd type at perinuclear sites." are not substantiated with data.
7. The claim of "crypticity" is also not substantiated with data to indicate that antibodies are or not binding to binding sites under particular conditions tested (\pm GTC), and on iS7 cells.
8. Lines 186 paragraph: No evidence is provided to show that PrPd as recognized by antibody 6C11 and that is close proximity to PrPd as recognized with antibody 5B2, reaches the plasma membrane from the inside of the cell or if it is on the outside of the cell. No plasma membrane marker staining is shown, so it is impossible to tell where 6C11 or 5B2 PrPd are.
9. The extent of colocalization of Synapsin 1 and PrPd on Fig. 3C is not clear from the image, which in fact by eye, show most instances of non-colocalization. There is no quantification provided or co-localization analysis done. Again, as noted in point 5 above, there is no evidence in this manuscript demonstrating that PrPd is associated within or with vesicles.

10. Fig.3D: the right Y-axis is labeled as "Relative toxicity". It is unclear why/how relative toxicity decreases with increasing concentration of m β CD, and also in manuscript it is stated that toxicity is lower below 1 mM m β CD, so this very confusing.

11. Lines 209-211: it is assumed (but not indicated in the manuscript) that the quantification of colocalization of intensities in Fig.3F is on the plasma membrane since both antibodies 5B2 and 6C11 label PrPd at the plasma membrane or just outside of it. Authors claim that fluorescence intensity of colocalization is low there in the presence of 0.4 mM m β CD, but this is not apparent in the images provided on Fig.3F, which show strong colocalization of PrPd in both 5B2 and 6C11 channels.

12. Lines 211-215: "In summary, the lowering of cellular cholesterol levels suggests (i) that intra- and extracellular pools of PrPd can be uncoupled and (ii) that blocking of PrPd transport to the plasma membrane inhibits the formation of FL-PrPd at the plasma membrane." Cholesterol does not block transport of vesicles to the plasma membrane, so this is incorrect.

13. Fig. 4A-C, and 4E-F does not convincingly show colocalization of PrPd with any of the vesicular markers shown. There is no quantitation nor colocalization analysis provided and by eye, colocalization is not at all clear. The lack of colocalization is also very clear on Fig.4F. Remarkably, Fig.3B is shown no less than 3X in the manuscript (also as Fig.4B, and Fig. S4G). Quantitation by Pearson coefficient is provided after manuscript description in Fig.4B and S4G, and yet the quantitation of colocalization by Pearson is not obvious as no colocalization that is apparent in the image.

14. Fig.4F: authors claim that Synapsin 1 is at the ECM. This is very confusing. How can a synapsin 1 vesicle be found extracellularly at the ECM?

15. Lines 240-249: it is unclear if authors are referring to lower steady state intracellular PrPd levels or total after treatment with CME inhibitors (dynasore and dynole). Treatment with these compounds would be expected to lower intracellular PrPd, as they block endocytosis. But applied PrPd should still be found at the cell surface (plasma membrane) as treatment with dynasore or dynole should not inhibit binding of PrPd⁻ to the cell surface. These observations contradict observations in the subsequent paragraph (lines 266-268) where authors report that downregulation of clathrin didn't lead to reductions in PrPd steady state levels (again, unclear whether intracellular or also at the cell surface).

16. Lines 330-333: "Notably, failure in detecting intracellular myc-tagged PrPd in persistently infected G30 and G45, but not in G70 PrP-expressing cells under denaturing conditions confirms that PrPd is N-terminally truncated at a site prior to G70 (Fig. S6C)." The observation of lack of detection of myc-tagged PrPd at G30 and G45 could indicate N-terminal truncation, but could also be explained by other scenarios, including by lack of binding of antibodies at those regions in the presence of the myc-tag. In other words, validation of proper binding of antibodies with myc-tagged PrPd constructs needs to be done.

17. Lines 339-349: The "de novo" conclusion for PrPd formation at plasma membrane and perinuclear is based on the presence of PrPd-positive signal at plasma membrane and perinuclear sites starting at 2 min post infection with RML or exosomes. However, the presence of 6C11 signal at perinuclear sites could be also the result of PrPd conversion at the plasma membrane and then followed by internalization of converted PrPd that then deposits at perinuclear sites. This possibility is not considered nor tested.

Minor points:

1. Throughout figures, it is important to provide name of cells being studied. For example, in Fig. 2D, are these cells also iS7? Because various cell lines are used including N2a, and because localization of

PrPd conformations might be different in cell lines, this is important to clarify.

2. Although in figure legend, it would be helpful and it is customary to label every panel in Fig. 6C with antibodies used.

3. In Fig. 6D clarify if percentage numbers inside is on cell numbers?

Response to reviewer's comments

Reviewer #1

We greatly appreciate the positive comments of this reviewer. Referring to the fibril-like PrP^d aggregates in astrocytes from primary neuronal cell cultures in **Fig. 1H**, this reviewer queries whether this phenotype is astrocyte-specific or whether neurons show the same fibril-like aggregates. We have now cultured primary hippocampal neuronal cultures from e17 embryonic mouse brains in the presence of 1 μ M cytarabine (AraC), a treatment that selectively removes glial cells and infected cultures with mouse RML prions. After 2-3 weeks of culture, fibril-like PrP^d aggregates were detected on Microtubule-associated protein 2 (Map2)-positive neurons with anti-PrP antibody 5B2 (**Fig. 1I**), in agreement with data on GFAP-positive astrocytes (**Fig. 1H**). This result demonstrates that fibril-like PrP^d aggregates are detected in prion-susceptible primary cells. To accommodate the additional **Fig. 1I**, images of astrocyte-bearing fibrils in **Figs. 1Hb** and **1Hc** were deleted. We unreservedly agree with this reviewer that strain-dependent differences in internalisation routes, as reported by Fehlinger et al (2017) is a highly relevant and interesting aspect, but argue that this question is beyond the scope of this publication. We would like to refer to our previous report on strain-dependent effects in astrocyte cultures instead¹, where we show that astrocytes in mixed neuronal cultures propagate the prion strains RML and 22L, but not Me7.

This reviewer requests clarification on the binding characteristics of Saf32, i.e. whether it recognises N-terminally truncated PrP on immunoblot and whether the Saf32 epitope is retained on truncated PrP. To clarify the binding characteristics of Saf32, we refer to the literature of this extensively characterised antibody in addition to our data. Saf32 has multiple binding sites in the octapeptide region² and recognises a truncated PrP fragment, C2, which is formed by beta cleavage³. In agreement with our study, Masujin et al. reported that Saf32 binds to abnormal PrP under native and denaturing conditions⁴. In the Masujin study abnormal PrP was isolated from BSE-infected mice. In agreement with results from Rouvinski et al.⁵, we show that Saf32 recognises plasma membrane bound PrP under native conditions. Our data using myc-Prnp chimera in **Fig. S6** show that intracellular PrP^d can be detected with an anti-myc antibody in G70 myc-Prnp expressing cells, but not in G30 myc Prnp and G45 myc-Prnp expressing cells, suggesting that PrP^d is cleaved prior to the third octapeptide repeat. Our data thus confirm that Saf32 is able to detect N-terminally truncated PrP^d species. Importantly though, Saf32 binds to extracellular PrP^d under native conditions, while its binding site on intracellular PrP^d is cryptic (**Fig. 2E**), suggesting protein conformational changes.

We thank this reviewer for the suggestion to test whether the GPI-anchored protein PrP^c is extracted from the plasma membrane under experimental conditions of **Fig. 3E**, i.e. with β -methyl cyclodextrin (m β CD) concentrations < 1 mM. We treated confluent monolayers of uninfected S7 cells with m β CD (0.5, 5 and 50 mM) for 30 minutes at 37°C in agreement with a m β CD extraction protocol⁶ and determined the loss of PrP^c fluorescence (**Fig. 3G**). While a significant reduction of PrP^c levels were determined at 50 mM m β CD, no significant changes in PrP^c levels were detected at 5 and 0.5 mM m β CD (**Fig. 3G**). We thus conclude that extraction of GPI-anchored PrP^c does not explain the significant reduction of PrP^d levels at the plasma membrane in **Fig. 3E, F**.

We agree with this reviewer that the perturbation of lipid rafts is an alternative explanation for a reduction of colocalisation at the plasma membrane, following m β CD incubation and we have added this in Results, lines 250-252.

This reviewer suggested that knockout efficacies of 40-50% for Arf1 and Arf6 may not suffice to exclude that prion steady state levels are affected by transcriptional silencing of said genes. We now used two novel custom-designed siPools with 30 distinct siRNAs against the 3'-UTR regions of Arf1 and Arf6 from siTools Biotech GmbH (Planegg, Germany); of note gene coding regions are the primary site for siRNAs at siTools Biotech GmbH. Knockdown efficacies of 80% (Arf1) and 90% (Arf6) were confirmed by qPCR (see Table S5). The average percent changes of prion steady state levels following gene silencing of Arf1 and Arf6 were 27% and 21%, respectively. We have updated these results in **Fig. 5E** and **Table S5**, including statistical analyses.

As suggested, we have also omitted caveolin 1 (Cav1) from bar chart in **Fig. 5E**.

While it is prudent to check knockdown efficacies on RNA as well as on protein level, we do not agree that this is a feasible strategy for a gene candidate approach due to the scarcity of validated antibodies. We nevertheless sourced commercial antibodies against Arf6 (3A-1, Santa Cruz), Cdc42 (B-8, Santa Cruz) and Arf1 (ab58578, Abcam), however, with disappointing results, since in immunocytochemistry high unspecific background staining was observed for all sourced antibodies. Given the moderate effect changes of Arf1 and Arf6 knockdown on prion steady state levels in **Fig. 5E**, we do not think that the reporting of knockdown levels on protein level adds value.

Reviewer #2

We thank this reviewer for acknowledging that our novel research tools are a “good source for the prion field”. We are furthermore grateful for this detailed review and the valuable suggestions. This reviewer started out with a few general comments, before addressing specifics of the paper point-by-point. In our response, we will first respond to the point-by-point comments, before addressing general comments.

Point 1: We thank this reviewer for suggesting to provide data on how distance to the extracellular matrix (ECM) was estimated in serial images in z. We now include an image of a representative z-stack of cells, colabelled with mAbs against PrP and ECM-resident Focal adhesion kinase (FAK, **Fig. S1F**). As further explained in Methods, in-focus detection of FAK coincides with in-focus detection of PrP^d fibrils at the ECM and was arbitrarily denoted “zero μm ”. This information was added to the figure legend in **Fig. S1F** and is further explained in Methods (lines 656-61).

Point 2: In regards to this reviewer’s question whether PrP^d fibrils are not observed in primary neurons, we refer to new data of PrP^d fibrils which are associated with Map2-positive neurons (**Fig. 1I**), added to this manuscript in response to comments from reviewer #1. This additional result confirms that Map2-positive neurons replicate prions and form fibril-like PrP^d aggregates that can be labelled with mAbs against the PrP N-terminus.

Point 3: We greatly appreciate the reviewer’s suggestion of displaying the two channels of the SIM image as greyscale in addition to the merged image to better appreciate the degree of colocalisation. We have amended this figure accordingly. To accommodate the additional space in **Fig. 2**, we deleted the second SIM image set in **Fig. 2Cb**.

Point 4: While we agree with this reviewer that biochemical characterisation of different PrP^d aggregation states using antibodies would provide further validation, all our attempts to isolate fibrillar PrP^d from persistently infected cells following cell lysis or by immunoprecipitation using anti-PrP mAbs failed. This result is in agreement with failure in detecting 5B2-positive fibrillar PrP^d on ELISA plates by Scrapie cell assay after cell trituration (data unpublished). After careful inspection, we found that fibrillar PrP^d strongly adheres to

the ECM and remains bound to plastic ware and glass slides following trituration of cells, as shown in the new supplementary figure, **Fig. S6B**. Cell lysis under stringent conditions did not change this outcome. To follow-up this outcome and to further investigate whether fibrillar PrP^d is lost after trituration of cells, we imaged resuspended cells in a time-dependent manner (**Fig. S6C-F**). Unexpectedly, we were unable to detect full-length PrP^d with the N-terminal antibody 5B2 before 3 days after resuspension of cells (**Fig. S6F**), suggesting that FL-PrP^d is fully degraded, presumably following rapid internalisation. That FL-PrP^d is rapidly degraded after resuspension of cells was indeed confirmed by blocking proteolysis (**Fig. S6G-J**). Pre-incubation of persistently infected cells with the V-ATPase inhibitor BafA1, a treatment that increases luminal pH levels, led to detection of intracellular FL-PrP^d (**Fig. S6H**). This result confirms that full-length and truncated PrP^d species greatly differ in their rate of formation, a result that corroborates our data on the fast kinetics of PrP conversion versus lagging formation of FL-PrP^d (see **Fig. 6F**). That BafA1 gives rise to intracellular FL-PrP^d substantiates our evidence of the pH-dependence of proteinases activation (see Point 5 below). To our knowledge this is the first evidence for chaperone-independent fibril fragmentation in mammalian cells. Fibril fragmentation was thus far only described in yeast cells. We report this experiment in lines 191-215, since it strongly corroborates the notion of intracellular processing of FL-PrP^d (**Fig. 2F**), which is relevant for Point 5 below.

Point 5: The reviewer here claims that the authors have not provided experimental evidence that intracellular PrP species are proteolytically truncated and suggests that “biochemical characterisation” of PrP^d in isolated endolysosomes will help to clarify this.

We strongly disagree with the reviewer’s opinion as elaborated below. In this study we provide three lines of evidence that intracellular PrP^d species are truncated: (1) Three mAbs against the N-terminus of PrP, AG4, 8B4 and 5B2 recognise PrP^d at the plasma membrane, but fail to detect intracellular PrP^d (**Fig. S4 and Fig. 2E**); that intracellular PrP^d is present under these conditions is demonstrated by double-labelling with 6D11, an antibody that binds to “core” PrP⁷ (epitope 93-109, **Fig. 1D,F,H and Fig. S4**). (2) An anti-myc mAb detects intracellular PrP^d only in cells where the myc tag has been cloned into the octapeptide region (G70 myc Prnp), but not in those where it has been cloned downstream of N-terminal codons 30 and 45 (**Fig. S7D**), suggesting that the site of proteolysis is prior to the third octapeptide repeat. (3) That lowering of the intracellular pH by lysosomotropic amines, including NH₄Cl, chloroquine and quinolone derivatives blocks proteolytic processing of PrP has been widely demonstrated⁸⁻¹³ using Western blotting and biochemical assays. By blocking proton shuttle across membranes, BafA1, a specific inhibitor of V-ATPases¹⁴ has the same net effect as lysosomotropic amines, i.e. an increase in the luminal pH. Our study provides the first evidence that full-length PrP^d can be detected intracellularly only after treatment of cells with NH₄Cl and BafA1 (**Fig. 2F** and in the new figure, **Fig. S6** (see Point 4)). This data unequivocally demonstrate that intracellular PrP^d is proteolytically processed. We thus strongly oppose the notion of this reviewer that truncated PrP^d is “total speculation”, since experimental proof in this study is evident using different experimental paradigms. With this in mind, we think the reviewer’s suggestion of isolating “endolysosomes” is “shifting the goalposts” of a study that provides novel insight into the cellular trafficking of truncated PrP^d on a whole cell context (see lines 429-434 for outcomes), a result that cannot be achieved with a *pars pro toto* approach as suggested by the reviewer. Finally, our new data confirming that PrP^d segregates into vesicular pathways (Point 9 and **Fig. 4H-J**) questions the value of isolating “endolysosomes”.

Point 6: By referring to lines 179-182, this reviewer notes that there is no experiment that justifies the term “nascent PrP^d” at the plasma membrane and we fully agree with this notion and are grateful to the reviewer for pointing this out. While the term “nascent PrP^{Sc}” has

been used by others to denote *de novo* formed PrP^d, like Rouvinski et al.⁵, our data show that *de novo* PrP^d is distinct from plasma membrane resident fibrillar PrP^d (**Fig. 6E,F**). We have therefore deleted the term “nascent” in this manuscript.

Point 7: This reviewer notes that the claim of “crypticity” is unsubstantiated and we disagree with this comment. As shown in **Fig. S4** and **Fig. 2E**, only mAbs against the PrP N-terminus (5B2, 8B4, AG4 and Saf32) detect abnormal PrP (PrP^d) under native conditions, whereas all other mAbs against “core PrP”⁷ do not. Hence, epitopes of these mAbs are cryptic under native conditions. Note that first evidence of the crypticity of antibody binding sites in abnormal PrP was reported as early as 1990¹⁵ and the authors suggested that denaturing conditions are required for antibody binding. We show that incubation of fixed cells with the denaturing agent GTC exposes epitopes that were cryptic under native conditions (**Fig. 2E**). In summary, the phenomenon of crypticity is fully characterised for all mAbs used in this study. Antibody epitopes that are cryptic on abnormal PrP are listed in **Fig. 2E**. Our model for cryptic binding sites (**Fig. 2G**) is in agreement with Rouvinski et al.⁵.

Point 8: This reviewer suggests that *“no evidence is provided to show that PrP^d as recognized by antibody 6D11 [corrected from “6C11”] and that is close proximity to PrP^d as recognized with antibody 5B2, reaches the plasma membrane from the inside of the cell or if it is on the outside of the cell. No “plasma membrane marker” staining is shown, so it is impossible to tell where 6D11[corrected from “6C11”] or 5B2 PrP^d are.”*

In response, we would like to refer to **Fig. 3C**, where we use an antibody against the “plasma membrane marker” integrin $\beta 1$ to show that 6D11-positive PrP^d is detected proximal to and at the level of the plasma membrane. However, we agree that this does not clarify whether 6D11-positive PrP^d is transported from the cytosolic side to the plasma membrane. We now, however, provide unequivocal evidence for the directional transport of intracellular PrP^d to the plasma membrane via synaptic vesicles, following potassium chloride-evoked depolarisation. This experiment is described in full in Point 9 below.

Point 9: This reviewer notes that inspection of representative images of Synapsin 1 with PrP^d in **Fig. 3C** does not provide evidence of colocalisation and concludes that evidence that PrP^d is associated with synaptic vesicles is not provided.

Firstly, we would like to note that **Fig. 3C** is a representative image and Pearson correlation coefficients, a widely acknowledged and used metric for assessing the colocalisation of proteins are shown in **Fig. 4G** and represent background-corrected mean values of 20-40 images. We thus suggested that this provides evidence that PrP^d reaches the plasma membrane by vesicular transport (Line 200).

We now provide further evidence that PrP^d reaches the plasma membrane by vesicular transport. We subjected persistently prion-infected cells with KCl concentrations of up to 100 mM, a treatment that leads to fast depolarisation, followed by calcium influx¹⁶. Under these conditions Syn1-positive vesicles rapidly fuse with the plasma membrane (**Fig. 4H**), while 6D11/5B2 co-labelling at the plasma membrane significantly increases (**Fig. 4I, J**). This provides unambiguous evidence that PrP^d segregates into the vesicular pathway and reaches the plasma membrane by means of regulated exocytosis. To accommodate for additional **Fig. 4H-J**, we deleted what was formerly **Fig. 4F**. This experiment is described in lines 269-74 in the manuscript.

Point 10: We thank this reviewer for noting that the Y axis in **Fig. 3D** reads “relative toxicity” and we apologise for this error. We now changed the axis to “relative viability”.

Point 11: We thank this reviewer for pointing out that more technical detail would benefit **Fig. 3F**. For clarity, we now added to the figure legend that “total fluorescence above threshold” was determined. In addition, we have replaced the term “colocalisation” in the legend of **Fig. 3D** with “intersection”, the technical term for colabelling used in image quantification software and specified in the Methods paragraph *Quantifying relative PrP^d fluorescence intensities*. Intersection thus denotes colabelled areas where pixels are detected in more than one fluorescence channel” to distinguish the term “intersection” from “colocalisation” which is based on a distinct algorithm as explained in said Methods section. We do not agree with the reviewer that there are inconsistencies in fluorescence levels in **Fig. 3E, F**. Notably, the drop in 5B2 fluorescence and 5B2/6D11 colabelling shown in **Fig. 3F** is in agreement with reduced colocalisation in m β CD-treated prion-infected cells, compared to mock-treated cells (**Fig. 3E**). For clarity, we added to **Fig. 3F** legend “data normalised to vehicle control (1.0)”.

Point 12: We here refer to amendments of the manuscript in response to Reviewer #1 in paragraph 3 above where we fully agree that the perturbation of lipid rafts is an alternative explanation for our results in **Fig. 3E, F**. This was added in results, lines 250-252.

Point 13: This reviewer comments that there is no convincing evidence for colocalisation in **Fig. 4** and that “by eye, colocalisation is not at all clear”. Analysis of colocalisation by Pearson’s correlation coefficients has already been addressed in Point 9 and we reiterate that we disagree with the reviewer’s opinion and would like to point out that Pearson’s correlation coefficients in this study are based on average values of 20-40 images, while the images displayed in **Fig. 4** are single representative images. We therefore consider the argument of “by eye inspection” misleading as statistically evaluated data is presented (see **Fig. 4G**).

Point 14: Here the reviewer wonders how Syn1 can be found extracellularly at the ECM. For clarification, we refer to recent publications^{17, 18}. Please note that this image has been removed to accommodate for additional **Fig. 4H-K**.

Point 15: The reviewer notes that “*it is unclear if authors are referring to lower steady state intracellular PrP^d levels or total after treatment with CME inhibitors (dynasore and dynole). Treatment with these compounds would be expected to lower intracellular PrP^d, as they block endocytosis.*” The points that this reviewer raises here are incorrect, since prion steady state levels in this manuscript were determined by “Scrapie cell assay” which monitors the infectious titre of a cell population at steady state, but does not provide information about intracellular or plasma membrane PrP^d levels. Incubation of persistently infected cells with Dynasore led to a 90% reduction of prion titres within 3 days (see Methods for details).

The reviewer furthermore points out that the inhibitory effects of the “CME inhibitors” dynasore and dynole contradict that clathrin inhibition has no effect on prion levels. We find this conclusion concerning, given unequivocal evidence of clathrin-independent endocytosis (CIE) pathways during the past two decades (for recent reviews see¹⁹⁻²²). In fact, the strong inhibitory effect of the dynamin inhibitor dynasore (**Fig. 5A**) and the failure of Pitstop2, an inhibitor of clathrin suggest a role in CIE and this is further supported in **Fig. 5E**, where expression of dynamins and clathrin are directly targeted by transcriptional silencing. We here note that knowledge of CIE is highly relevant to understand the conclusions drawn in this manuscript in regards to gene perturbation effects on prion steady state levels (**Fig. 5E**) and inhibition of prion infection (**Fig. 6G**), respectively.

Point 16: In reference to **Fig. S6D** and lines 377-379 of the manuscript: “Notably, failure in detecting...”, this reviewer notes that “*lack of detection of myc-tagged PrP^d at G30 and G45 could indicate N-terminal truncation, but could also be explained by other scenarios, including by lack of binding of antibodies at those regions in the presence of the myc-tag. In*

other words, validation of proper binding of antibodies with myc-tagged PrP^d constructs needs to be done.” We are grateful to the reviewer for pointing this out. We now colabelled G30 and G45 myc *Prnp* expressing cells with N-terminal anti-PrP mAbs. For compatibility with the position of the tag, we colabelled persistently prion-infected G45 myc *Prnp* expressing cells with anti-myc and 5B2 (48-50) and that of persistently infected G30 myc *Prnp* expressing cells with anti-myc and AG4 (32-52, **Fig. S8A, B**). As shown in anti-myc/anti-PrP double-labelled cells (**Fig. S8A**), as well as in serial sections of said cell types (**Fig. S8B**), this antibody pair detected, under denaturing conditions, PrP^d aggregates at the level of the plasma membrane with no intracellular label detected. This corroborates that G30 myc PrP^d and G45 myc PrP^d are N-terminally truncated.

Point 17: Here reviewer #2 notes that intracellular PrP^d at 2 minutes after infection “*could be the result of PrP^d conversion at the plasma membrane and then followed by internalization of converted PrP^d that then deposits at perinuclear sites. This possibility is not considered nor tested.*” In response to this notion, it is important to point out that the frequency of PrP conversion is invariably low (see **Fig. 6E**), i.e. only one in a thousand cells shows evidence of PrP conversion, a finding that is consistent with the low proportion of infected cells following prion exposure for short time periods (**Fig. 5G**). This undoubtedly limits possibilities in exploring ultrafast internalisation processes of myc-tagged PrP chimera by image analysis. We therefore took advantage of infectivity testing (SCA) to check whether internalisation of infectious seeds can be blocked by knockdown of dynamins and Cdc42, two mediators of endocytosis that greatly affect prion steady state levels following gene loss-of-function (**Fig. 5A, E**). In fact, we now show that knockdown of Cdc42, double-knockdown of Dnm1/Dnm2 and triple-knockdown of Cdc42/Dnm1/Dnm2, respectively, markedly blocked prion infection of G70 myc *Prnp* expressing cells by as much as 80% (**Fig. 6G**). This strongly suggests that internalisation of the infectious seed is critical to establish an infection, thus ruling out the plasma membrane as the sole site of PrP conversion. In response to point 17, we report this new data in lines 412-419 and point out the experimental limitations of investigating PrP conversion using myc-tagged PrP chimera, given the low frequency of events.

Minor points: As requested, cell types are now explicitly specified throughout the manuscript (point 1), mAbs used in **Fig. 6C** has been specified for each figure panel (point 2) and the frequency of occurrence in Venn diagrams has been specified in the legend of **Fig. 6D** (point 3).

General comments

In paragraph 2, this reviewer notes “*there are major deficiencies in the manuscript. One of the main ones is the heavy or almost sole reliance on imaging data throughout the manuscript and lack of use of very well established biochemical assays that are routinely used for studying prion conversion.*”

We strongly disagree with this reviewer’s claim of “*heavy or sole reliance on imaging data throughout the manuscript*” and would like to point out that this study is based on correlative imaging and prion titre output methodology. In fact, this is the first study that addresses *de novo* PrP conversion in neuronal cells by monitoring prion levels, rather than relying on surrogate markers of prions, like proteinase K-resistant PrP (PrP^{Sc}) bands on Western blot. Following the Goold et al study²³, we thus provide the

first independent evidence for fast PrP conversion rates using the Scrapie Cell Assay (SCA)^{24, 25} (**Fig. 5FG-I**). Moreover, the critical finding that prion steady state levels are dependent on functional dynamins and Cdc42 (**Fig. 5A, E**) is based on SCA data. Furthermore, that prion infection of neuronal cells is highly dependent on clathrin-independent endocytosis (**Fig. S6F**) is based on monitoring *bona fide* prion infectivity levels. I hope this clarifies that pivotal imaging outcomes, generated with tagged *Prnp* chimera in this study are validated by prion based methodology.

In line with the response above, we strongly disagree with the reviewer's notion "*lack of use of very well established biochemical assays that are routinely used for studying prion conversion.*" We strongly believe that studies that address the molecular underpinning of prion propagation should be conducted with *bona fide* infectivity readout methodology, such as mouse bioassays or cell-based prion titre determination, like the SCA and not with surrogate markers for prions, like PrP^{Sc} bands on Western blots. Secondly, Goold et al.²³ provided first evidence that PrP converts within minutes using myc-tagged *Prnp* chimera. This work has not been followed up or repeated since submission of our data. Hence the notion "*established biochemical assays that are routinely used for studying prion conversion*" is highly misleading and incorrect.

This reviewer further notes that "*PrP^d is claimed to be in or associated with vesicles*". In response, we now provide unequivocal evidence that PrP^d segregates into regulated exocytosis pathways by triggering depolarisation (**Fig. 4H-J**), a treatment that significantly increased plasma membrane PrP^d levels, thus corroborating the finding that PrP^d colocalises with markers of vesicular transport (**Fig. 4A,B,C,G**).

We strongly disagree with the reviewer's further remarks that "*data in colocalization analysis that is not apparent by visual inspection of the images provided and non-consistent with the Pearson coefficient provided for various comparisons*" and here refer to Points 9 and 13.

This reviewer further notes that "*(2) PrP^d is concluded to be proteolytically cleaved after treatment of prion-infected cells with BafA1, but no data is shown for proteolytic cleavage so the conclusions drawn about truncated PrP forms is totally speculative*". Firstly, to clarify, BafA1 is an inhibitor and not a stimulator of V-ATPases. In fact, in **Fig. 2F** we show that incubation with BafA1 blocks PrP^d truncation rather than "*proteolytically cleaving*" PrP^d as falsely stated. Irrespective of this misunderstanding, we would like to point out that evidence for the activation of proteases at low luminal pH is broadly acknowledged²⁶⁻²⁸ while the V-ATPase inhibitor BafA1 counters auto-activation of proteinases²⁹. Our new data in **Fig. S6** confirm that proteolytic cleavage of FL-PrP^d following cell dissociation can be prevented by BafA1 inhibition (**Fig. S6G-J**) thus corroborating our initial finding in **Fig. 2F**. We hence strongly disagree with the notion that conclusions drawn on distinct proteolytically processed PrP^d species are "totally speculative" and further refer to Point 5 for our response.

This reviewer further notes that "*(3) evidence that PrP conversion happens at two sites...is based on presence of anti-PrP^d staining at these two sites 2 min after infection and beyond. But observation of PrP^d at plasma membrane within this short period of time doesn't prove de novo conversion, but could be just initial binding of the infected prions to the cell surface.*"

To clarify a principal misunderstanding, our experimental strategy of using myc-tagged *Prnp* chimera is designed to avoid use of anti-PrP antibodies in favour of antibodies against the myc tag. This enables us to unequivocally distinguish *de novo* PrP conversion from anti-PrP immunopositive inoculum. The risk of false-positive results due to "*initial binding of the infected prions to the cell surface*" is thus excluded by use of anti-myc antibodies. In regards to the "internalisation of *de novo* converted PrP", we now present further evidence that the

initial infection of cells is dependent on the internalisation of infectious seeds (**Fig. S6F**) which provides experimental evidence for intracellular PrP conversion.

In reference to the reviewer's notion of the possibility of "*internalisation of PrP^d from the cell surface after initial infection*", we refer to our response on additional data presented in Point 17.

The reviewer claims that "*Importantly and majorly, no assay is performed in this study showing conversion of PrP. I.e., no biochemical de novo conversion assays are used to show definitely de novo formation or conversion.*" As explained above, the use of myc-tagged *Prnp* expressing cells, a principal strategy to detect PrP conversion in presence of inoculum which was first used by Goold et al.²³, is a versatile method to investigate *de novo* PrP conversion (see **Fig. 6**).

In line with what is said above in regards to advantages of myc-tagging strategies to distinguish *de novo* PrP conversion against inoculum, we reject the reviewer's notion: "*This is critical to do, as all the staining could be due to initial binding of externally-applied fibrils and accumulation of these fibrils over time.*"

The reviewer's notion "(4) [the] conclusion claiming that PrP^d is transported to the plasma membrane actively associated with synaptic and large-dense core vesicles is weak or not provided" has already been addressed above; we refer to Point 9 for further detail.

In reference to the reviewer's comment that "*first, colocalization of synaptic and dense core vesicle markers with PrP^d is not convincing. Second, colocalisation of synaptic and dense core vesicle markers with PrP^d is not convincing*", we refer to additional data corroborating that PrP^d is secreted by regulated exocytosis (see Point 9).

This reviewer is mistaken about the feasibility of "*live imaging of transport of these presumed vesicles ... is a very feasible assay that would allow the rapid and fast visualization of moving PrP^d vesicles*". Following our thorough investigation of cryptic PrP binding sites (**Fig. 2D-G and Fig. S6E**), it should be apparent that detection of intracellular PrP^d requires denaturing conditions, an experimental condition that is incompatible with live imaging.

Referring to the reviewer's notion "*There is no discussion as to how PrP^d could get into the secretory pathway as claimed in the manuscript*" we would like to note that this is beyond the scope of this project and will be investigated in a BBSRC-funded study (project reference BB/V001310/1).

This reviewer adds that "*Furthermore, there are contradictions in data, for example, reduction of endocytosis using dynasore and dynole or genetically by downregulation dynamin, is reported to lower PrP^d steady state levels, but reducing clathrin does not. It is unclear and unexplored how some clathrin mediated endocytosis routes are affected but not others?*" To reiterate our response of Point 15, we reject the reviewer's claim that our data on clathrin and dynamin inhibition are contradictory and would like to refer to a large body of evidence on clathrin-independent endocytosis (CIE). For our full response to the reviewer's response see Point 15.

All other points in the general comments of this author have been addressed above or in the point-by-point comments.

References

1. Piliastides A, Ribes JM, Yip DC, Schmidt C, Benilova I, Klöhn PC. A New Cell Model for Investigating Prion Strain Selection and Adaptation. LID - 10.3390/v11100888 [doi] LID - 888.
2. Yin S, *et al.* Prion proteins with insertion mutations have altered N-terminal conformation, increased ligand-binding activity and are more susceptible to oxidative attack. *J Biol Chem*, (2006).
3. Lewis V, *et al.* Increased proportions of C1 truncated prion protein protect against cellular M1000 prion infection. *J Neuropathol Exp Neurol* **68**, 1125-1135 (2009).
4. Masujin K, *et al.* The N-Terminal Sequence of Prion Protein Consists an Epitope Specific to the Abnormal Isoform of Prion Protein (PrP(Sc)). *PLoS ONE* **8**, e58013 (2013).
5. Rouvinski A, *et al.* Live imaging of prions reveals nascent PrP^{Sc} in cell-surface, raft-associated amyloid strings and webs. *J Cell Biol* **204**, 423-441 (2014).
6. Ilangumaran S, Hoessli DC. Effects of cholesterol depletion by cyclodextrin on the sphingolipid microdomains of the plasma membrane.
7. Prusiner SB. Novel proteinaceous infectious particles cause scrapie. *Science* **216**, 136-144 (1982).
8. Caughey B, Raymond GJ, Ernst D, Race RE. N-terminal truncation of the scrapie-associated form of PrP by lysosomal protease(s): implications regarding the site of conversion of PrP to the protease-resistant state. *J Virol* **65 No 12**, 6597-6603 (1991).
9. Taraboulos A, Raeber A, Borchelt DR, Serban D, Prusiner SB. Synthesis and trafficking of prion proteins in cultured cells. *Mol Biol of the Cell* **3**, 851-863 (1992).
10. Shyng S-L, Huber MT, Harris DA. A prion protein cycles between the cell surface and an endocytic compartment in cultured neuroblastoma cells. *J Biol Chem* **268 (21)**, 15922-15928 (1993).
11. Supattapone S, Nguyen HOB, Cohen FE, Prusiner SB, Scott MR. Elimination of prions by branched polyamines and implications for therapeutics. *Proceedings of the National Academy of Sciences of the United States of America* **96**, 14529-14534 (1999).
12. Doh-ura K, Iwaki T, Caughey B. Lysosomotropic agents and cysteine protease inhibitors inhibit scrapie-associated prion protein accumulation. *J Virol* **74**, 4894-4897 (2000).
13. Klingenstein R, Melnyk P, Leliveld SR, Ryckebusch A, Korth C. Similar Structure-Activity Relationships of Quinoline Derivatives for Antiprion and Antimalarial Effects. *J Med Chem* **49**, 5300-5308 (2006).
14. Umata T, Moriyama Y, Futai M, Mekada E. The cytotoxic action of diphtheria toxin and its degradation in intact Vero cells are inhibited by bafilomycin A1, a specific inhibitor of vacuolar-type H(+)-ATPase. *J Biol Chem* **265**, 21940-21945 (1990).
15. Taraboulos A, Serban D, Prusiner SB. Scrapie prion proteins accumulate in the cytoplasm of persistently infected cultured cells. *J Cell Biol* **110**, 2117-2132 (1990).
16. Gärtner A, Staiger V. Neurotrophin secretion from hippocampal neurons evoked by long-term-potential-inducing electrical stimulation patterns.

17. Schiera G, Di Liegro CM, Di L, I. Cell-to-Cell Communication in Learning and Memory: From Neuro- and Glio-Transmission to Information Exchange Mediated by Extracellular Vesicles. *Int J Mol Sci* **21**, (2019).
18. Xia X, Wang Y, Qin Y, Zhao S, Zheng JC. Exosome: A novel neurotransmission modulator or non-canonical neurotransmitter? *Ageing Res Rev* **74**, 101558 (2022).
19. Mayor S, Pagano RE. Pathways of clathrin-independent endocytosis. *Nat Rev Mol Cell Biol* **8**, 603-612 (2007).
20. Shafaq-Zadah M, Dransart E, Johannes L. Clathrin-independent endocytosis, retrograde trafficking, and cell polarity. *Curr Opin Cell Biol* **65**, 112-121 (2020).
21. Sandvig K, Kavaliauskiene S, Skotland T. Clathrin-independent endocytosis: an increasing degree of complexity. *Histochem Cell Biol* **150**, 107-118 (2018).
22. Mayor S, Parton RG, Donaldson JG. Clathrin-independent pathways of endocytosis. *Cold Spring Harb Perspect Biol* **6**, (2014).
23. Goold R, *et al.* Rapid cell-surface prion protein conversion revealed using a novel cell system. *Nat Commun* **2**, 281 (2011).
24. Klohn P, Stoltze L, Flechsig E, Enari M, Weissmann C. A quantitative, highly sensitive cell-based infectivity assay for mouse scrapie prions. *Proc Natl Acad Sci USA* **100**, 11666-11671 (2003).
25. Schmidt C, *et al.* A systematic investigation of production of synthetic prions from recombinant prion protein. *Open Biol* **5**, (2015).
26. Johe P, Jaenicke E, Neuweiler H, Schirmeister T, Kersten C, Hellmich UA. Structure, interdomain dynamics, and pH-dependent autoactivation of pro-rhodesain, the main lysosomal cysteine protease from African trypanosomes. *J Biol Chem* **296**, 100565 (2021).
27. Wang JK, *et al.* Matriptase autoactivation is tightly regulated by the cellular chemical environments. *Plos One* **9**, e93899 (2014).
28. Li DN, Matthews SP, Antoniou AN, Mazzeo D, Watts C. Multistep autoactivation of asparaginyl endopeptidase in vitro and in vivo. *J Biol Chem* **278**, 38980-38990 (2003).
29. Ishidoh K, Kominami E. Processing and activation of lysosomal proteinases. *Biol Chem* **383**, 1827-1831 (2002).

REVIEWER COMMENTS

Reviewer #3 (Remarks to the Author):

Evaluating the authors' response to the original reviewers' critique

1. Authors' rebuttal to Reviewer #1 critique

Point 1

Reviewer #1: - Fig 1H: in the results the authors refer to 'neuronal culture' but only show results for primary astrocytes – what was the result for neurons? It is interesting though that astrocytes show the same PrP^d fibril-like aggregates, suggesting this is not typical for neurons; in this experiment, 22L was used for infection, for most other figures RML – what was the rationale, are strain differences expected in particular for the internalisation pathways– as Fehlinger et al (2017) described different routes of internalisation are used by RML and 22L to establish productive infection; it is suggested to include some discussion about the strain used and whether the authors expect similar results for all strains

Authors' response: Referring to the fibril-like PrP^d aggregates in astrocytes from primary neuronal cell cultures in **Fig. 1H**, this reviewer queries whether this phenotype is astrocyte-specific or whether neurons show the same fibril-like aggregates. We have now cultured primary hippocampal neuronal cultures from e17 embryonic mouse brains in the presence of 1µM cytarabine (AraC), a treatment that selectively removes glial cells and infected cultures with mouse RML prions. After 2-3 weeks of culture, fibril-like PrP^d aggregates were detected on Microtubule-associated protein 2 (Map2)-positive neurons with anti-PrP antibody 5B2 (**Fig. 1I**), in agreement with data on GFAP-positive astrocytes (**Fig. 1H**). This result demonstrates that fibril-like PrP^d aggregates are detected in prion-susceptible primary cells. To accommodate the additional **Fig. 1I**, images of astrocyte-bearing fibrils in **Figs. 1Hb** and **1Hc** were deleted.

We unreservedly agree with this reviewer that strain-dependent differences in internalisation routes, as reported by Fehlinger et al (2017) is a highly relevant and interesting aspect, but argue that this question is beyond the scope of this publication. We would like to refer to our previous report on strain-dependent effects in astrocyte cultures instead¹, where we show that astrocytes in mixed neuronal cultures propagate the prion strains RML and 22L, but not Me7.

Reviewer #3:

- *Authors responded well by adding neuronal cultures.*
- As to the strain dependent internalization routes, the reviewer suggested that this point to be discussed in the paper. *As far as I can see this was not followed up.*

Point 2

Reviewer #1: - SAF32 recognizes cell surface and intracellular PrP^d: the explanation of differences in the octarepeat region is not clear – is it not that 5B2 does not recognize intracellular PrP^d because of N-terminal truncation – while possibly the SAF32 epitope is

retained? Does SAF32 recognize N-terminally truncated PrPd eg by immunoblot?

Authors' response: This reviewer requests clarification on the binding characteristics of Saf32, i.e. whether it recognises N-terminally truncated PrP on immunoblot and whether the Saf32 epitope is retained on truncated PrP. To clarify the binding characteristics of Saf32, we refer to the literature of this extensively characterised antibody in addition to our data. Saf32 has multiple binding sites in the octapeptide region² and recognises a truncated PrP fragment, C2, which is formed by beta cleavage³. In agreement with our study, Masujin et al. reported that Saf32 binds to abnormal PrP under native and denaturing conditions⁴. In the Masujin study abnormal PrP was isolated from BSE infected mice. In agreement with results from Rouvinski et al.⁵, we show that Saf32 recognises plasma membrane bound PrP under native conditions. Our data using myc-Prnp chimera in **Fig. S6** show that intracellular PrP_d can be detected with an anti-myc antibody in G70 myc-Prnp expressing cells, but not in G30 myc Prnp and G45 myc-Prnp expressing cells, suggesting that PrP_d is cleaved prior to the third octapeptide repeat. Our data thus confirm that Saf32 is able to detect N-terminally truncated PrP_d species. Importantly though, Saf32 binds to extracellular PrP_d under native conditions, while its binding site on intracellular PrP_d is cryptic (**Fig. 2E**), suggesting protein conformational changes.

Reviewer #3:

Saf32 stands out among other antibodies in that it reacts natively with cell surface PrPd but requires denaturation to stain intracellular PrPd.

This was construed by the authors as a proof of a conformational difference between cell surface and intracellular PrPd. ("L160: Saf32 showed notable propensities, in that it detected extracellular fibril-like PrPd under native conditions, like N-terminal mAbs and intracellular PrPd under denaturing conditions, unlike N-terminal mAbs, suggesting that extracellular and intracellular PrPd species are conformationally distinct at the octapeptide repeat region (Fig. 2E and Fig. S4)." Also L182)

Reviewer #1 raised 3 points:

a. He/she questioned the authors' interpretation (about conformational distinction) and suggested that the contrasting reactivities may just reflect the presence of the Saf32 epitope in full length PrPd on the cell surface vs its removal by proteolysis in perinuclear PrPd.

The authors responded by a description of the Saf32 octapeptide repeat epitopes, some of which may remain under the so-called beta cleavage of PrP. *I find this response satisfying*, and suggest that this description should be included in the paper, including a reference for the beta cleavage (eg Mange et al. 2004).

b. Reviewer #1 asks "does Saf32 recognize N-terminally truncated PrPd eg by immunoblot?". I believe that additional immunoblots would probably not add much to the current ms, mainly because they would include both cell surface and intracellular PrPd, which differ in the extent of their N-terminus. *It's OK that the authors' didn't directly address that.*

c. *I disagree with the authors' assessment* that the fact that "Saf32 binds to extracellular PrP_d under native conditions, while its binding site on intracellular PrP_d is cryptic (**Fig. 2E**)" suggests "protein conformational changes". That is because SAF32 staining of cell surface PrPd can occur via its N-proximal epitopes without indicating the extent to which the C-proximal octapeptide is cryptic, or not, in cell surface PrPd. *This was not raised by the original review and is a new point.* I suggest that this point be at least discussed in the ms.

Point 3

Reviewer #1: - Fig.3E: from the significant reduction of 5B2 signal it is concluded that a block of intracellular PrP^d to the plasma membrane inhibits the formation of FL-PrP^d; however, first, cholesterol extraction would affect lipid raft integrity at the plasma membrane, know to reduce PrP^{Sc} formation – is lipid raft perturbation an alternative explanation for the lack of 5B2 signal? Second, cholesterol extraction can release GPI-anchored proteins. It is recommended to include a control for the release of GPI-anchored proteins from the cell surface, e.g. PrP^C.

Authors' response: We thank this reviewer for the suggestion to test whether the GPI-anchored protein PrP^C is extracted from the plasma membrane under experimental conditions of **Fig. 3E**, i.e. with β -methyl cyclodextrin (m β CD) concentrations < 1 mM. We treated confluent monolayers of uninfected S7 cells with m β CD (0.5, 5 and 50 mM) for 30 minutes at 37°C in agreement with a m β CD extraction protocol⁶ and determined the loss of PrP^C fluorescence (**Fig. 3G**). While a significant reduction of PrP^C levels were determined at 50 mM m β CD, no significant changes in PrP^C levels were detected at 5 and 0.5 mM m β CD (**Fig. 3G**). We thus conclude that extraction of GPI-anchored PrP^C does not explain the significant reduction of PrP^d levels at the plasma membrane in **Fig. 3E, F**.

We agree with this reviewer that the perturbation of lipid rafts is an alternative explanation for a reduction of colocalisation at the plasma membrane, following m β CD incubation and we have added this in Results, lines 250-252.

Reviewer #3:

- The authors now refer to a possible involvement of raft perturbation in the m β CD results: L243 “Alternatively to inhibition of vesicle transport, a perturbation of lipid rafts by m β CD could account for the reduced 5B2/6D11 colocalisation at the plasma membrane.” *This is satisfactory. However, I suggest that they include an explanation as to how this is at all relevant.*

- As to the PrP^C extraction: The authors have now treated uninfected cells to monitor whether PrP^C is extracted: L237 “To exclude the possibility that m β CD treatment leads to an extraction of PrP^C under the experimental conditions of Figure 3EF, we treated uninfected cells with m β CD concentrations of up to 50 mM (Figure 4G), confirming that PrP^C levels remain unchanged at m β CD concentrations below 5 mM.” Since they mentioned “... the experimental conditions of Fig. 3EF,... (is that a typo?), this reader assumed that treatment length was also 16h (thus a proper control). However, in their response above the authors claim a 30min extraction. Please specify in the ms the time frame of this experiment.

If this is indeed true, then this control is not acceptable.

Point 4

Reviewer #1: - Fig. 5: Arf1, Arf6 and Cav1 knock-down and effects on PrP^d: it is commented in Table S5 that Cav1 is not expressed in the used cells; therefore, it should be omitted from the graph. The knock-down of Arf1 and Arf6 on the mRNA level is only around 40 – 50% efficient; the question arises whether this level of knock down is sufficient to affect PrP^d, and

how it translates into a reduction of protein levels, which could be even less efficient depending on the half life of the proteins. It is suggested to include analysis of protein levels for these proteins.

Authors' response: This reviewer suggested that knockout efficacies of 40-50% for Arf1 and Arf6 may not suffice to exclude that prion steady state levels are affected by transcriptional silencing of said genes. We now used two novel custom-designed siPools with 30 distinct siRNAs against the 3'-UTR regions of Arf1 and Arf6 from siTools Biotech GmbH (Planegg, Germany); of note gene coding regions are the primary site for siRNAs at siTools Biotech GmbH. Knockdown efficacies of 80% (Arf1) and 90% (Arf6) were confirmed by qPCR (see Table S5). The average percent changes of prion steady state levels following gene silencing of Arf1 and Arf6 were 27% and 21%, respectively. We have updated these results in **Fig. 5E** and **Table S5**, including statistical analyses.

As suggested, we have also omitted caveolin 1 (Cav1) from bar chart in **Fig. 5E**. While it is prudent to check knockdown efficacies on RNA as well as on protein level, we do not agree that this is a feasible strategy for a gene candidate approach due to the scarcity of validated antibodies. We nevertheless sourced commercial antibodies against Arf6 (3A-1, Santa Cruz), Cdc42 (B-8, Santa Cruz) and Arf1 (ab58578, Abcam), however, with disappointing results, since in immunocytochemistry high unspecific background staining was observed for all sourced antibodies. Given the moderate effect changes of Arf1 and Arf6 knockdown on prion steady state levels in **Fig. 5E**, we do not think that the reporting of knockdown levels on protein level adds value.

Reviewer #3: They have opted to remove the data altogether, which is disappointing since they could have run western blots to assess the knockdown. *However, I consider this response satisfactory.*

2. Authors' rebuttal to Reviewer #2 critique

Response to "Major points"

Point 1

Reviewer #2: 1. Imaging of staining of PrPd fibrils being made at the ECM (E.g., Fig 1C), using antibody against ECM-resident protein focal adhesion kinase (FAK) proteins is not provided. Because this is a key point to the study, this needs to be included.

Authors' response: We thank this reviewer for suggesting to provide data on how distance to the extracellular matrix (ECM) was estimated in serial images in z. We now include an image of a representative z-stack of cells, colabelled with mAbs against PrP and ECM-resident Focal adhesion kinase (FAK, Fig. S1F). As further explained in Methods, in-focus detection of FAK coincides with in-focus detection of PrPd fibrils at the ECM and was arbitrarily denoted "zero μm ". This information was added to the figure legend in Fig. S1F and is further explained in Methods (lines 656-61).

Reviewer #3: In keeping with Reviewer #2 recommendation, the authors now provide a z-stack of PrP/FAK co-stained infected cells (Fig. S1F). *This is nominally a satisfactory response but the picture is low quality.* It is supposed to show that PrP^d fibrils appear at the same z depths as the FAK spots, but I am not seeing this. Please provide enlarged insets. Also, in the legend to this panel (L1187) you send the reader to Methods. Where in the Methods? What are we supposed to look for there?

Point 2

Reviewer #2: Line 133: “In cultures from FVB wt mouse brains, fibril length continuously and significantly increased over three weeks, at mean fibril elongation rates of 190 nm per day, reaching lengths of up to 15 μ m, while no fibrils were detected at the periphery of astrocytes from brains of FVB Prnp^{-/-} mice (Fig. 2A, B).” It is unclear why fibril length is measured in the periphery of neurons in FVB wt mouse brains, but in the periphery of astrocytes in brains of FVB Prnp^{-/-} mice? It appears PrP^d fibrils are only around astrocytes not primary neurons?

Author’s response: In regards to this reviewer’s question whether PrP^d fibrils are not observed in primary neurons, we refer to new data of PrP^d fibrils which are associated with Map2- positive neurons (**Fig. 11**), added to this manuscript in response to comments from reviewer #1. This additional result confirms that Map2-positive neurons replicate prions and form fibrillike PrP^d aggregates that can be labelled with mAbs against the PrP N-terminus.

Reviewer #3: *This is satisfactory.* However, *the new panel I is not explained in the text.* Re the text and the legend: when was the labeling done post infection? At what point in the establishment of the culture were the cells infected? Why did you choose RML here vs 22L for the astrocytes?

Point 3

Reviewer #2: Fig.2C: in the co-labeling experiment, it is difficult to appreciate the degree of localization or not, as the figure is provided only as a merge of two channels. Separation of each channel in black/white (inverse could help), in addition to the merge would facilitate evaluation of the observations/conclusions presented with these data.

Response: Point 3: We greatly appreciate the reviewer’s suggestion of displaying the two channels of the SIM image as greyscale in addition to the merged image to better appreciate the degree of co-localisation. We have amended this figure accordingly. To accommodate the additional space in Fig. 2, we deleted the second SIM image set in Fig. 2Cb.

Reviewer #3: *Satisfactory*

Point 4

Reviewer #2: 4. Line 142: Data presented to suggest “that different aggregation states may co-exist during PrPd fibril formation” is based on light microscopy imaging. Biochemical characterization of these different PrPd aggregate states using antibodies used in these imaging studies would provide important orthogonal validation.

Response: Point 4: While we agree with this reviewer that biochemical characterisation of different PrPd aggregation states using antibodies would provide further validation, all our attempts to isolate fibrillar PrPd from persistently infected cells following cell lysis or by immunoprecipitation using anti-PrP mAbs failed. This result is in agreement with failure in detecting 5B2-positive fibrillar PrPd on ELISA plates by Scrapie cell assay after cell trituration (data unpublished). After careful inspection, we found that fibrillar PrPd strongly adheres to the ECM and remains bound to plastic ware and glass slides following trituration of cells, as shown in the new supplementary figure, **Fig. S6B**. Cell lysis under stringent conditions did not change this outcome. To follow-up this outcome and to further investigate whether fibrillar PrPd is lost after triturating cells, we imaged resuspended cells in a time-dependent manner (**Fig. S6C-F**). Unexpectedly, we were unable to detect full-length PrPd with the N terminal antibody 5B2 before 3 days after resuspension of cells (**Fig. S6F**), suggesting that FL-PrPd is fully degraded, presumably following rapid internalisation. That FL-PrPd is rapidly degraded after resuspension of cells was indeed confirmed by blocking proteolysis (**Fig. S6G-J**). Pre-incubation of persistently infected cells with the V-ATPase inhibitor BafA1, a treatment that increases luminal pH levels, led to detection of intracellular FL-PrPd (**Fig. S6H**). This result confirms that full-length and truncated PrPd species greatly differ in their rate of formation, a result that corroborates our data on the fast kinetics of PrP conversion versus lagging formation of FL-PrPd (see **Fig. 6F**). That BafA1 gives rise to intracellular FLPrPd substantiates our evidence of the pH-dependence of proteinases activation (see Point 5 below). To our knowledge this is the first evidence for chaperone-independent fibril fragmentation in mammalian cells. Fibril fragmentation was thus far only described in yeast cells. We report this experiment in lines 191-215, since it strongly corroborates the notion of intracellular processing of FL-PrPd (**Fig. 2F**), which is relevant for Point 5 below.

Reviewer #3: The authors explain that they were unable to isolate the fibrillar structures at the ECM (ECM presumably meaning the contacts with the plastic dish). This is because the fibrils stay attached to the substrate following “trituration”. However, they do not explain what are the experimental conditions of the trituration, or of the “Cell lysis under stringent conditions” that “did not change this outcome”, so that I cannot relate to the experiment. What happened to the cell lysate in these experiments? Did it not contain any full length PrPd? I agree that images of fibrillar structures attached to the plastic substrate (Fig. S6B) are very interesting, but the Reviewer #2 expressly asked for biochemical characterizations that should have been at least attempted using cell lysates.

In my opinion, the authors’ response to this point is not satisfactory.

I also disagree with the last paragraph of their response, since it is based on the assumption that the cell surface fibrillar assemblies are just plain amyloid fibrils (or rods as they are called elsewhere in the paper). As far as I could see, they do not provide any proof for that and this interpretation seems to be at odd with previous data on “prion

strings" (ref 14) (that seem to have been renamed here to "fibrillar structures".) However, this remark was not included in the original Review.

Point 5

Reviewer #2: A very major deficiency in the manuscript with consequences to the rest of the manuscript starting with lines 165-185 is the following: The observation and conclusion that 5B2-positive aggregates in cells treated with BafA1 become visible due to blockage of proteolytic cleavage of full-length PrPd within less acidifying endolysosomes is not substantiated with data. First, there is no evidence provided that PrPd is inside or associated with endosomes (also see point 13 below). Second, evidence for proteolytic cleavage of PrPd is completely lacking so this cannot be concluded, therefore calling this fragment "truncated (TR) PrPd" is total speculation. Both light microscopy imaging as well as biochemical characterization of PrPd within endolysosomes and of proteolytic cleavage of PrPd in those endosomes (for example using endosome purification by sucrose gradient centrifugation) is required to substantiate these claims. BafA1 treatment alone does not indicate cleavage.

Response: The reviewer here claims that the authors have not provided experimental evidence that intracellular PrP species are proteolytically truncated and suggests that "biochemical characterisation" of PrPd in isolated endolysosomes will help to clarify this.

We strongly disagree with the reviewer's opinion as elaborated below. In this study we provide three lines of evidence that intracellular PrPd species are truncated: (1) Three mAbs against the N-terminus of PrP, AG4, 8B4 and 5B2 recognise PrPd at the plasma membrane, but fail to detect intracellular PrPd (**Fig. S4 and Fig. 2E**); that intracellular PrPd is present under these conditions is demonstrated by double-labelling with 6D11, an antibody that binds to "core" PrP7 (epitope 93-109, **Fig. 1D,F,H and Fig. S4**). (2) An anti-myc mAb detects intracellular PrPd only in cells where the myc tag has been cloned into the octapeptide region (G70 myc Prnp), but not in those where it has been cloned downstream of N-terminal codons 30 and 45 (**Fig. S7D**), suggesting that the site of proteolysis is prior to the third octapeptide repeat. (3) That lowering of the intracellular pH by lysosomotropic amines, including NH₄Cl, chloroquine and quinolone derivatives blocks proteolytic processing of PrP has been widely demonstrated⁸⁻¹³ using Western blotting and biochemical assays. By blocking proton shuttle across membranes, BafA1, a specific inhibitor of V-ATPases¹⁴ has the same net effect as lysosomotropic amines, i.e. an increase in the luminal pH. Our study provides the first evidence that full-length PrPd can be detected intracellularly only after treatment of cells with NH₄Cl and BafA1 (**Fig. 2F and in the new figure, Fig. S6** (see Point 4)). This data unequivocally demonstrate that intracellular PrPd is proteolytically processed. We thus strongly oppose the notion of this reviewer that truncated PrPd is "total speculation", since experimental proof in this study is evident using different experimental paradigms. With this in mind, we think the reviewer's suggestion of isolating "endolysosomes" is "shifting the goalposts" of a study that provides novel insight into the cellular trafficking of truncated PrPd on a whole cell context (see lines 429-434 for outcomes), a result that cannot be achieved with a *pars pro toto* approach as suggested by the reviewer. Finally, our new

data confirming that PrPd segregates into vesicular pathways (Point 9 and **Fig. 4H-J**) questions the value of isolating “endolysosomes”.

Reviewer #3: Reviewer #2 asked that “Both light microscopy imaging as well as biochemical characterization of PrPd within endolysosomes and of proteolytic cleavage of PrPd in those endosomes.” The authors strongly opposed that view, arguing that their immunofluorescence data with a variety of antibodies, together with the data with bafilomycin, NH₄Cl, etc sufficiently indicate that intracellular PrPd is usually truncated, unless acidic proteases are inhibited.

Reviewer #2 also asked for an independent assessment that PrPd in endolysosomes is indeed truncated, but becomes full length when acidification is prevented. Here, the authors argued “we think the reviewer’s suggestion of isolating “endolysosomes” is “shifting the goalposts”...”, especially considering their data “confirming that PrPd segregates into vesicular pathways (Point 9 and **Fig. 4H-J**)”, which “questions the value of isolating “endolysosomes”.

Even though it is probable that the authors’ conclusions (that PrPd is truncated by acidic proteases in endolysosomes) may turn out to be true (as hinted by studies from the 1990s), I stand by Reviewer #2 demands here.

Simple immunoblots with relevant antibodies would go a long way to prove that: a. Intracellular PrPd is usually truncated, but becomes full length with acidification inhibitors, and b. that the PrPd species involved is indeed aggregated PrPd (which is presumably the same as PrPSc?), and not PrPC, for instance. Admittedly, they would have to use established methods in the field such as sedimentation of detergent lysates, etc, but this would help to shore up their conclusions.

In my opinion, this issue illustrates *Reviewer #2’s critique that “major deficiencies are noted that relate primarily to insufficient or deficient or non-existent data that authors rely on (or not) to make the major conclusions made throughout the manuscript.”*

The same is true for Reviewer #2’s other request, that endolysosomes be isolated using established procedures and their PrP content analyzed. In my opinion, that is just a proper control that has nothing to do with “shifting the goalposts”, but rather should have been included in their original experiments.

I also don’t agree with their claim that the fact that PrPd was detected elsewhere in the “vesicular pathway” (is that the endomembrane system?) “questions the value of isolating “endolysosomes”. Quite the opposite, this would help ascertain that the truncation takes place in acidic and hydrolytic compartments, to the exclusion of other vesicles that they have detected in this respect.

Therefore, I don’t feel that their rebuttal arguments are solid. In my opinion, their answer to this point are completely unsatisfactory.

Point 6

Reviewer #2: There are no experiments in the manuscript demonstrating that PrPd recognized by any of the antibodies tested are detecting “nascent” PrPd (versus non-nascent?). Thus, this claim as well as the one on point 5 above, i.e., lines 179-182 “These results suggest existence of two distinct aggregated PrPd species in prion-infected cells, nascent FL-PrPd which is associated with fibril elongation at the plasma membrane and a truncated (TR-) PrPd type at perinuclear sites.” are not substantiated with data.

Authors’ response: By referring to lines 179-182, this reviewer notes that there is no experiment that justifies the term “nascent PrPd” at the plasma membrane and we fully agree with this notion and are grateful to the reviewer for pointing this out. While the term “nascent PrPSc” has been used by others to denote *de novo* formed PrPd, like Rouvinski et al.5, our data show that *de novo* PrPd is distinct from plasma membrane resident fibrillar PrPd (**Fig. 6E,F**). We have therefore deleted the term “nascent” in this manuscript.

Reviewer #3: *This is satisfactory.*

Point 7

Reviewer #2: The claim of “crypticity” is also not substantiated with data to indicate that antibodies are or not binding to binding sites under particular conditions tested (\pm GTC), and on iS7 cells.

Authors’ response: This reviewer notes that the claim of “crypticity” is unsubstantiated and we disagree with this comment. As shown in **Fig. S4** and **Fig. 2E**, only mAbs against the PrP N-terminus (5B2, 8B4, AG4 and Saf32) detect abnormal PrP (PrPd) under native conditions, whereas all other mAbs against “core PrP”⁷ do not. Hence, epitopes of these mAbs are cryptic under native conditions. Note that first evidence of the crypticity of antibody binding sites in abnormal PrP was reported as early as 1990¹⁵ and the authors suggested that denaturing conditions are required for antibody binding. We show that incubation of fixed cells with the denaturing agent GTC exposes epitopes that were cryptic under native conditions (**Fig. 2E**). In summary, the phenomenon of crypticity is fully characterised for all mAbs used in this study. Antibody epitopes that are cryptic on abnormal PrP are listed in **Fig. 2E**. Our model for cryptic binding sites (**Fig. 2G**) is in agreement with Rouvinski et al.5.

Reviewer #3: *OK*

Point 8

Reviewer #2: Lines 186 paragraph: No evidence is provided to show that PrPd as recognized by antibody 6C11 and that is close proximity to PrPd as recognized with antibody 5B2, reaches the plasma membrane from the inside of the cell or if it is on the outside of the cell. No plasma membrane marker staining is shown, so it is impossible to tell where 6C11 or 5B2 PrPd are.

Authors' response: This reviewer suggests that “no evidence is provided to show that PrPd as recognized by antibody 6D11 [corrected from “6C11”] and that is close proximity to PrPd as recognized with antibody 5B2, reaches the plasma membrane from the inside of the cell or if it is on the outside of the cell. No “plasma membrane marker” staining is shown, so it is impossible to tell where 6D11[corrected from “6C11”] or 5B2 PrPd are.” In response, we would like to refer to **Fig. 3C**, where we use an antibody against the “plasma membrane marker” integrin $\beta 1$ to show that 6D11-positive PrPd is detected proximal to and at the level of the plasma membrane. However, we agree that this does not clarify whether 6D11-positive PrPd is transported from the cytosolic side to the plasma membrane. We now, however, provide unequivocal evidence for the directional transport of intracellular PrPd to the plasma membrane via synaptic vesicles, following potassium chloride-evoked depolarisation. This experiment is described in full in Point 9 below.

Reviewer #3: (Re paragraph starting L211).
OK.

Point 9

Reviewer #2: The extent of colocalization of Synapsin 1 and PrPd on Fig. 3C is not clear from the image, which in fact by eye, show most instances of non-colocalization. There is no quantification provided or co-localization analysis done. Again, as noted in point 5 above, there is no evidence in this manuscript demonstrating that PrPd is associated within or with vesicles.

Authors' response: This reviewer notes that inspection of representative images of Synapsin 1 with PrPd in **Fig. 3C** does not provide evidence of colocalisation and concludes that evidence that PrPd is associated with synaptic vesicles is not provided. Firstly, we would like to note that **Fig. 3C** is a representative image and Pearson correlation coefficients, a widely acknowledged and used metric for assessing the colocalisation of proteins are shown in **Fig. 4G** and represent background-corrected mean values of 20-40 images. We thus suggested that this provides evidence that PrPd reaches the plasma membrane by vesicular transport (Line 200). We now provide further evidence that PrPd reaches the plasma membrane by vesicular transport. We subjected persistently prion-infected cells with KCl concentrations of up to 100 mM, a treatment that leads to fast depolarisation, followed by calcium influx¹⁶. Under these conditions Syn1-positive vesicles rapidly fuse with the plasma membrane (**Fig. 4H**), while 6D11/5B2 co-labelling at the plasma membrane significantly increases (**Fig. 4I, J**). This provides unambiguous evidence that PrPd segregates into the vesicular pathway and reaches the plasma membrane by means of regulated exocytosis. To accommodate for additional **Fig. 4H-J**, we deleted what was formerly **Fig. 4F**. This experiment is described in lines 269-74 in the manuscript.

Reviewer #3: I agree with the authors that the statistical analysis sides with PrPd being associated with synapsin 1 vesicles. The KCl experiments indeed add value to this contention. However, I agree with Reviewer #2 that PrPd could still be attached to the cytosolic side of the vesicles “there is no evidence in this manuscript demonstrating that PrPd is associated within or with vesicles”. It seems to me that this possibility could at least be could be mentioned in the text.

However, it is also possible that Reviewer #2 directed to another co-localization problem in the experiments summarized in Fig. 4G: can one derive from your data (here or elsewhere in the paper) what is the relative amount of PrPd in each of these vesicular locations? Such data is crucial to evaluate the power of the trafficking scheme in panel F. In other words, what is the fraction of the total PrPd found in each of these vesicles? How much in synaptic vesicles? In endolysosomes? Assuming that this was the original critique, then it is crucial that the authors address that at least in the text.

However, the KCl results (Fig. 4I) raise a major problem in my mind. As far as I can see, following this treatment there is a huge amount of 5B2-reactive PrP that now appears on the cell surface. And yet, the authors' interpretation is that intracellular PrPd is externalized on the cell surface (this is a test to see if "intracellular TR-PrPd segregates into regulated exocytosis" L263). How can this happen? Is it possible that this is PrPC that is quantitatively externalized by the KCl treatment? Have the authors run controls with uninfected cells? It is imperative that the authors address respond to this question. Of note, Reviewer #2 could not have raised this question, since these are new experiments that were not available in the original manuscript (as far as I know).

Point 10

Reviewer #2: Fig.3D: the right Y-axis is labeled as "Relative toxicity". It is unclear why/how relative toxicity decreases with increasing concentration of m β CD, and also in manuscript it is stated that toxicity is lower below 1 mM m β CD, so this very confusing.

Authors' response: We thank this reviewer for noting that the Y axis in **Fig. 3D** reads "relative toxicity" and we apologise for this error. We now changed the axis to "relative viability".

Reviewer #3: OK

Point 11

Reviewer #2: Lines 209-211: it is assumed (but not indicated in the manuscript) that the quantification of colocalization of intensities in Fig.3F is on the plasma membrane since both antibodies 5B2 and 6C11 label PrPd at the plasma membrane or just outside of it. Authors claim that fluorescence intensity of colocalization is low there in the presence of 0.4 mM m β CD, but this is not apparent in the images provided on Fig.3F, which show strong colocalization of PrPd in both 5B2 and 6C11 channels.

Author's response: We thank this reviewer for pointing out that more technical detail would benefit **Fig. 3F**. For clarity, we now added to the figure legend that "total fluorescence above threshold" was determined. In addition, we have replaced the term "colocalisation" in the legend of **Fig. 3D** with "intersection", the technical term for colabelling used in image quantification software and specified in the Methods paragraph *Quantifying relative PrPd fluorescence intensities*. Intersection thus denotes colabelled areas where pixels are detected in more than one fluorescence channel" to distinguish the term "intersection" from "colocalisation" which is based on a distinct algorithm as explained in said Methods section. We do not agree with the reviewer that

there are inconsistencies in fluorescence levels in **Fig. 3E, F**. Notably, the drop in 5B2 fluorescence and 5B2/6D11 colabelling shown in **Fig. 3F** is in agreement with reduced colocalisation in m β CD-treated prion-infected cells, compared to mock-treated cells (**Fig. 3E**). For clarity, we added to **Fig. 3F** legend “data normalised to vehicle control (1.0)”.

Reviewer #3: OK

Point 12

Reviewer #2: Lines 211-215: “In summary, the lowering of cellular cholesterol levels suggests (i) that intra- and extracellular pools of PrPd can be uncoupled and (ii) that blocking of PrPd transport to the plasma membrane inhibits the formation of FL-PrPd at the plasma membrane.” Cholesterol does not block transport of vesicles to the plasma membrane, so this is incorrect.

Authors’ response: We here refer to amendments of the manuscript in response to Reviewer #1 in paragraph 3 above where we fully agree that the perturbation of lipid rafts is an alternative explanation for our results in **Fig. 3E, F**. This was added in results, lines 250-252.

Reviewer #3: The authors do not address Reviewer #2 remark: Cholesterol does not block transport of vesicles to the plasma membrane, so this is incorrect.” In fact, how can we accept such a conclusion if m β CD is so pleiotropic and non-specific, especially if used for protracted periods of time such as 16h? There are plenty of references about cholesterol depletion decreasing endocytosis and many other processes. Furthermore, have the authors examined the distribution of synapsin 1 under these conditions? *In my opinion, these caveats must be at least mentioned and discussed, in response to Reviewer #2’s question.*

Point 13

Reviewer #2: Fig. 4A-C, and 4E-F does not convincingly show colocalization of PrPd with any of the vesicular markers shown. There is no quantitation nor colocalization analysis provided and by eye, colocalization is not at all clear. The lack of colocalization is also very clear on Fig.4F. Remarkably, Fig.3B is shown no less than 3X in the manuscript (also as Fig.4B, and Fig. S4G). Quantitation by Pearson coefficient is provided after manuscript description in Fig.4B and S4G, and yet the quantitation of colocalization by Pearson is not obvious as no colocalization that is apparent in the image.

Authors’ response: This reviewer comments that there is no convincing evidence for colocalisation in **Fig. 4** and that “by eye, colocalisation is not at all clear”. Analysis of colocalisation by Pearson’s correlation coefficients has already been addressed in Point 9 and we reiterate that we disagree with the reviewer’s opinion and would like to point out that Pearson’s correlation coefficients in this study are based on average values of 20-40 images, while the images displayed in **Fig. 4** are single representative images. We therefore consider the argument of “by eye inspection” misleading as statistically evaluated data is presented (see **Fig. 4G**).

Reviewer #3: Assuming that there are no mistakes in how the statistics were computed, then I would tend to accept the authors explanation.

However, by looking at Fig. 4 and Fig. S5, it seems that the PrPd distribution changes in each image to fit the distribution of the particular vesicular marker. For example, syn 1 and lamp1. Is this because the focal plane was chosen to get a clear image of the chosen marker? This would make sense, but the question is then: when calculating correlation coefficients, did you include whole z-stacks? From Methods L635 paragraph, this does not appear to be the case.

I am thus not convinced that the Reviewer's question was well addressed. In my opinion, then, these questions should be addressed in the manuscript and the method used well explained.

Point 14

Reviewer #2: Fig.4F: authors claim that Synapsin 1 is at the ECM. This is very confusing. How can a synapsin 1 vesicle be found extracellularly at the ECM?

Authors' response: Here the reviewer wonders how Syn1 can be found extracellularly at the ECM. For clarification, we refer to recent publications 17, 18. Please note that this image has been removed to accommodate for additional Fig. 4H-K.

Reviewer #3: *I am puzzled by the "brush off" response of the authors here.* Publication 17 describes an improved method to extract PrPSc using detergents and proteases, and ref 18 deals with the structure of PrP fibrils obtained using this method. They do not have anything with the suggested presence of Syn1 vesicles or PrPd in the ECM of living cells, which was what the Reviewer wondered about. The authors basically sends the reviewer to these publication, and then removes the disputed panel and its legend, without dealing with the question at hand. In my opinion, we are not merely dealing with a manuscript, but also with the Science. At the very least, the Authors should answer Reviewer #2 his/her legitimate question.

Furthermore, Fig. 4F, which was supposedly removed in the new version, is still quoted in L255: "Notably, at the ECM, 5B2-positive FL-PrPd fibrils are decorated with Syn1 (Fig. 4F)." I wonder what the authors' position is with the ECM/Syn1 vesicle issue.

Can the authors thus please state if they believe that there are Syn1 vesicles in the ECM, and how do "recent publication 17, 18" show that?

Point 15

Reviewer #2: Lines 240-249: it is unclear if authors are referring to lower steady state intracellular PrPd levels or total after treatment with CME inhibitors (dynasore and dynole). Treatment with these compounds would be expected to lower intracellular PrPd, as they block endocytosis. But applied PrPd should still be found at the cell surface (plasma membrane) as treatment with dynasore or dynole should not inhibit binding of

PrPd to the cell surface. These observations contradict observations in the subsequent paragraph (lines 266-268) where authors report that downregulation of clathrin didn't lead to reductions in PrPd steady state levels (again, unclear whether intracellular or also at the cell surface).

Authors' response: The reviewer notes that "*it is unclear if authors are referring to lower steady state intracellular PrPd levels or total after treatment with CME inhibitors (dynasore and dynole). Treatment with these compounds would be expected to lower intracellular PrPd, as they block endocytosis.*" The points that this reviewer raises here are incorrect, since prion steady state levels in this manuscript were determined by "Scrapie cell assay" which monitors the infectious titre of a cell population at steady state, but does not provide information about intracellular or plasma membrane PrPd levels. Incubation of persistently infected cells with Dynasore led to a 90% reduction of prion titres within 3 days (see Methods for details).

The reviewer furthermore points out that the inhibitory effects of the "CME inhibitors" dynasore and dynole contradict that clathrin inhibition has no effect on prion levels. We find this conclusion concerning, given unequivocal evidence of clathrin-independent endocytosis (CIE) pathways during the past two decades (for recent reviews see 19-22). In fact, the strong inhibitory effect of the dynamin inhibitor dynasore (**Fig. 5A**) and the failure of Pitstop2, an inhibitor of clathrin suggest a role in CIE and this is further supported in **Fig. 5E**, where expression of dynamins and clathrin are directly targeted by transcriptional silencing. We here note that knowledge of CIE is highly relevant to understand the conclusions drawn in this manuscript in regards to gene perturbation effects on prion steady state levels (**Fig. 5E**) and inhibition of prion infection (**Fig. 6G**), respectively.

Reviewer #3: Authors appear to scorn the Reviewer "We find this conclusion concerning, given unequivocal evidence of clathrin-independent endocytosis (CIE) pathways during the past two decades", but is that warranted?

I would like to suggest that the authors explain to the reader, in the paper, what are these CIE pathways, and at least speculate which of them (there are many) can potentially fit raft associated fibrils, etc.

General comments

In paragraph 2, this reviewer notes "*there are major deficiencies in the manuscript. One of the main ones is the heavy or almost sole reliance on imaging data throughout the manuscript and lack of use of very well established biochemical assays that are routinely used for studying prion conversion.*"

We strongly disagree with this reviewer's claim of "*heavy or sole reliance on imaging data throughout the manuscript*" and would like to point out that this study is based on correlative imaging and prion titre output methodology. In fact, this is the first study that addresses *de novo* PrP conversion in neuronal cells by monitoring prion levels, rather than relying on surrogate markers of prions, like proteinase K-resistant PrP (PrP^{Sc}) bands on Western blot.

Reviewer #3: about the paragraph above.

It is true that this study uses both immunofluorescence and SCA-based bioassays. In fact, in my opinion, the authors should be lauded for using bioassays in cell biological studies, which have traditionally utilized primarily microscopy and biochemical assays.

However, I agree with Reviewer #2 that in some cases biochemical studies are still absolutely needed to reinforce conclusions inferred from the immunomicroscopy studies. This is particularly evident in studies of Prd trafficking. This was explained in the point-by-point critique.

Additional comments from the reviewer:

One essential point

1. I am puzzled by the connection made throughout the paper between the cell surface fibrillar structures, amyloid fibrils, and prion “rods”.

Question 1: Do the authors actually equate the 5B2 positive fibrillar structures on the cell surface with bona fide amyloid fibrils? If so, on what grounds, and what is the exact type of amyloid fibrils that they have in mind? Are these the well-known prion rods, for instance?

If they indeed conceive the cell surface fibrillar aggregates as amyloid fibrils, then this should be stated very clearly and explicitly in the Introduction, and convincing arguments should be brought forward.

Is that the concept behind the title “... precedes fibrillisation”?

In the absence of a clear statement, we are left with hints in the Intro:

- L52ff: “... early studies of prion-infected mice showed evidence of “prion-amyloid filaments” in the extracellular space beneath the ependyma¹⁶ and recent studies characterised ex vivo isolated prion rods^{17, 18”.}

- L55: “A recent study reported the detection of lipid raft-associated amyloid strings and webs of PrP...^{14”.}

However, these references do not prove that the 5B2 fibrils are “amyloidic fibrils” or rods. To the contrary, the rods isolated in refs 17, 18 are classical rods that were formed after proteolysis (so no 5B2) and detergent extraction.

As to ref 14, they do not claim that their cell surface “strings” (that you have renamed fibrils) are amyloid fibrils, but perhaps alignments of 2D PrPSc structures that do stain with Thioflavin. If their model is right, then the current conclusion (I438) that “... proteolytic processing of FL-PrPd following reuptake provides a chaperon-independent cellular mechanism for fibril fragmentation, a cellular process that is considered critical to maintain prion replication.” is unwarranted.

As to rods: In methods, Line 515, “... under these conditions, the build-up of rod-like PrPd aggregates at the plasma membrane is observed ...”. Why do you label these aggregates “rod-like”? In the prion field,

“rods” is applied to a specific structure that can be studied by EM, etc (see your own refs 17, 18.) Have you determined that these aggregates are “rod-like” in terms of tinctorial properties, Ab labeling, EM structure? If not, then, in my opinion, you must refrain from such claims as they are very confusing.

L643: “To image PrPd rods at the plasma membrane and ECM, cells were grown for extended culture times...”

L1257, “5B2- positive rods”

Question 2: By contrast, do the authors assume that perinuclear PrPd is NOT amyloidic?

For instance, L429: “... N-terminally truncated TR-PrPd at perinuclear sites fails to form amyloidogenic aggregates... Owing to the lack of amyloidogenic properties, TR-PrPd may represent amorphous aggregates or oligomers.” I was unable to follow the logic here. Is it assumed that because we do not see fibrils by light microscopy, there is no amyloid? Rods can be as short as 80nm in length (ref 18 Fig. 2f). Thus can't perinuclear PrPd be amyloids? Has anyone shown that directly by thioflavin staining etc?

Perhaps the anti-amyloid oligomers antibody (ab126892) from Abcam can be used to examine that in future studies.

By the way, what do the authors mean by “amyloidogenic fibrils”? Merriam Webster: “producing or tending to produce amyloid deposits”. As opposed to amyloidic (established amyloid, perhaps with tinctorial properties or some EM or structural proof).

This reader was confused and perplexed by this issue.

Minor points and typos:

L158: Citation 22: wrong reference, before the discovery of PrP.

L238: “ ... under the experimental conditions of Figure 3EF, we... “. Is it Fig. E and F?

Also, by “experimental conditions” do you mean m β CD for 16h? Please state this. It's not in the Fig. legend either.

L245: “PrPd is segregated into vesicles of the regulated secretory pathway and colocalises with markers of the endosomal pathway”. Same title for Fig. 4 legend L1079. This title is confusing. Do you mean “in part” for both localizations, which are supposedly mutually exclusive?

L259: “secretory vesicles (Syp, Vamp4, Syn1, Scg2, Chga, Col4) and endosomal/lysosomal 259 markers (Eea1, Lamp1) (Fig. 4G, H).” No, panel H is a KCl experiment now.

L263: “we triggered membrane depolarisation with potassium chloride (KCl).” How long the treatment with KCl? No indication in Fig. 4 legend either, nor in Methods. Please specify.

L266: “...confirming that FT-PrPd reaches the plasma...” What is FT-PrPd? Is that supposed to be FL-PrPd, or TR-PrPd?

L1081: “(A-D) Prion-infected S7 cells were co-labelled with 6D11 and vesicular markers”. This is just an example, out of many throughout the ms, of how the reader cannot determine what type of cells were used: persistently infected? Acutely infected? In the latter case, what is the timing of infection?

L608 and following: From the description of the method it is not clear if, and how, the cell lysates were normalized for cell #, or protein concentration, etc, prior to cholesterol determination using the Amplex red kit. Please detail the procedure.

Thus, In Fig. 3D, the “relative cholesterol content” – is that normalized to cellular protein content? Or otherwise normalized?

L 1074: “ ...PrPc extraction from the plasma membrane as specified in Methods.” Please specify which paragraph in “Methods”. This reviewer, for one, was unable to ascertain this detail.

L612: Fig. X?

L531: What is “ribolysed”? Is this term used to describe the use of a Ribolyzer grinding instrument? Please mention that. How is it performed? In what buffer? This is crucial as it pertains to how cells are infected in each experiment. In L539, what is “clarified” brain homogenate? Please specify if all infections were performed using “ribolyzed” brains, or clarified homogenates, or whether there were exceptions?

L625: Is there really 10% Pen/Strep in the blocking step?

Fig. S1C: inset is misplaced

L122 and L1025, 1028, and more: The use of the terms “neuronal” and “neural” is incorrect. See for instance the Thermofisher explanation of these terms: “There are several types of cells in the nervous system including neurons, astrocytes, and microglia. The term "neural" refers to any type of nerve cell, including a mixture of brain cells, whereas "neuronal" is specifically related to neurons. The term "neural" is used in context of both primary cultures and neural stem cells.”

(<https://www.thermofisher.com/il/en/home/life-science/cell-culture/primary-cell-culture/neuronal-cell-culture.html>).

Fig. 2F: Arrows not explained/mentioned in the text or Fig. legend.

Fig. 3A: the arrows are not mentioned in the text nor in the legend. The arrow in the original magnification is not followed in the inserts a, b. Do the right panels show another cell? Arrows in original magnification do not point at anything I can see.

Can you please mention the estimated lateral resolution with the SIM attachment?

L1061: re panel B. “A schematic for additive colour mixing was added next to the merged image to depict colocalisation patterns.” I don’t see the schematic in the figure. Also, scale bars in B?

Why do we need this panel here? There is almost no co-localization data to be gleaned from it. Is it SIM as well? Can you show a proper magnification (you did that anyway in Fig. 4).

L1027 specify “at 3 wks post infection”

L126 Panel I is not at all explained in the text. In the text and in the legend, how long after the infection?

Fig. S5: If the text is followed, then “Asynchronous changes in intra- and extracellular PrPd pools after cell dissociation” should be Fig. S5, not S6.

L 1083: E: all these cells were triple labeled, it’s just that in b, c, you just show two colours.

L1087: Levels of colocalization . specify what: 6D11 vs vesicular markers?

L1087: Pearson

L276: “Two pharmacological dynamin inhibitors, dynasore and dynole showed a strong dose-dependent decrease of cellular prion steady state levels by as much as 90% over three days (Fig. 5A). Perhaps specify: ... decrease of infectious cell associated prion ... as determined by SCA”?

Fig. 5 legend (L1100) and tables S3 and S4: please include the time length of treatment with the inhibitors.

L1356: Table S3: Here and elsewhere, can you please indicate which of the two medium change regimes (Methods, L510 ff) was used?

Reviewer #4 (Remarks to the Author):

This paper addresses an important and insufficiently explored question in prion biology: where and when are prions generated in the cell? Answering this question is experimentally challenging, even in model cell lines, because of the lack of availability of antibodies or other probes that can distinguish PrPC from PrPSc, making it difficult to track the conversion process kinetically and at subcellular resolution.

This paper attempts to overcome this difficulty using existing PrP antibodies with selective reactivities toward certain conformational and sequence-specific epitopes, as well as an epitope-tagged form of PrPC. A key strategy relies on the use of antibodies against the N-terminal region of PrPC, an approach that builds on the observations of Rouvinski et al. (2014), who first described PrPSc strings at the plasma membrane of infected cells. The use of myc epitope-tagged PrP derives from earlier work by Goold et al. (2011). By applying these strategies in immunolabeling experiments, the authors identify truncated and full-length forms of PrPSc, and characterize their cellular localization and growth kinetics. They find that the two pools are connected by exocytic transport of truncated PrPSc to the surface, and clathrin-independent endocytosis of the full length PrPSc into the cell.

The work is very rigorous, thorough, and well controlled. The authors have made a serious attempt to address the individual points raised by the previous reviewers, most notably reviewer #2 who had the most extensive criticisms. The authors have undertaken several new experiments, although the manuscript revisions have been relatively modest in scope.

Although it has major strengths, this study also suffers from a significant limitation, which is the exclusive reliance on immunofluorescence detection with a set of antibodies whose selective recognition of PrP^{Sc} is conditional, depending on their relative reactivities under native vs. denaturing conditions, and on their staining of PrP^{Sc} molecules before and after proteolytic truncation of the N-terminus. Thus, the conclusions are, of necessity, somewhat indirect. The addition of another methodology to complement the immunofluorescence approach would help strengthen the arguments the authors are trying to make. For example, isolation and characterization of different forms of cell-associated PrP^{Sc} in conjunction with subcellular fractionation would provide confirmatory evidence, as would pulse-chase labeling to follow the kinetic transformations of the different forms and their trafficking through the cell. Even if some fibrillar PrP^{Sc} remains bound to the substrate (Fig. S6), it should be possible to isolate cell-associated pools of PrP^{Sc} associated with particular organelles. Of course, it would also help to derive new antibodies with more selective reactivities, or to tag the PrP molecules with fluorescent probes that would allow their recognition by real-time imaging.

Other points:

1. In addition to relying on N-terminal antibodies to follow PrP^{Sc} formation and processing, why don't the authors try monitoring myc-tagged PrP not only at early times after infection, but also at later times, after the cells grow fibrils. This will also allow the authors to unambiguously determine if the fibrillar PrP^{Sc} derives from endogenous PrP^C, or if it results from residual PrP^{Sc} in the inoculum.
2. The paper leaves open the identity of the perinuclear region where truncated PrP^{Sc} accumulates. Co-staining with additional marker proteins would help here.
3. As written, the paper will be a bit opaque to the non-prion biologist, since it does not clearly convey the big picture of the results, and fails to engage the reader with an easily comprehensible story line. The use of multiple antibodies makes this especially important.

Minor Points:

4. Figure 1H – The primary astrocytes are infected with 22L and not RML. There is no discussion on strain differences in the paper so it would be best to substitute this for an image of RML infected primary astrocytes to keep this consistent throughout.

5. Line 519 - Although the paper specifies use of neuronal cultures, the images presented in Figures 1 and 2 are astrocytes (except Figure 1H). Are the astrocytes part of a coculture system or is it a glial culture? This should be clarified to prevent confusion and keep the image legends consistent.

6. The paper bounces around in the naming of the cell cultures as infected S7, iS7, and PK1. It would be best to change all references to S7 or infected S7, instead of using the iS7 designation, and to rename the cells in Figure S6A and S6G as S7 cells, if they are indeed the same thing.

7. References vary in terms of the number of authors mentioned vs. et al.

Response to reviewers' comments NCOMMS-22-17638A-Z

For clarity, all comments by reviewers #3 and #4 were answered in this document to provide context to comments and responses from the previous review. Notably, comments by reviewers #3 and #4 submitted by email by the editor were copied into this document as well and addressed. For clarity, all new responses by the authors in this document are in font color blue. We chronologically responded to comments and “additional comments” from reviewer #3, followed by the response to reviewer #4. New amendments to the manuscript are in font color blue to distinguish those from previous changes.

We are grateful to both reviewers for stepping in to review the manuscript in the absence of the previous reviewers. Notwithstanding the resulting disruption, we have worked diligently to address the questions raised by all reviewers.

1. Response to comments from reviewer #3

Evaluating the authors' response to the original reviewers' critique

1. Authors' rebuttal to Reviewer #1 critique

Point 1

Reviewer #1: - Fig 1H: in the results the authors refer to 'neuronal culture' but only show results for primary astrocytes – what was the result for neurons? It is interesting though that astrocytes show the ; same PrP^d fibril-like aggregates, suggesting this is not typical for neurons; in this experiment, 22L was used for infection, for most other figures RML – what was the rationale, are strain differences expected in particular for the internalisation pathways– as Fehlinger et al (2017) described different routes of internalisation are used by RML and 22L to establish productive infection; it is suggested to include some discussion about the strain used and whether the authors expect similar results for all strains

Authors' response: Referring to the fibril-like PrP^d aggregates in astrocytes from primary neuronal cell cultures in **Fig. 1H**, this reviewer queries whether this phenotype is astrocyte-specific or whether neurons show the same fibril-like aggregates. We have now cultured primary hippocampal neuronal cultures from e17 embryonic mouse brains in the presence of 1µM cytarabine (AraC), a treatment that selectively removes glial cells and infected cultures with mouse RML prions. After 2-3 weeks of culture, fibril-like PrP^d aggregates were detected on Microtubule-associated protein 2 (Map2)-positive neurons with anti-PrP antibody 5B2 (**Fig. 1I**), in agreement with data on GFAP-positive astrocytes (**Fig. 1H**). This result demonstrates that fibril-like PrP^d aggregates are detected in prion-susceptible primary cells. To accommodate the additional **Fig. 1I**, images of astrocyte-bearing fibrils in **Figs. 1Hb** and **1Hc** were deleted.

We unreservedly agree with this reviewer that strain-dependent differences in internalisation routes, as reported by Fehlinger et al (2017) is a highly relevant and interesting aspect, but argue that this question is beyond the scope of this publication. We would like to refer to our previous report on strain-dependent effects in astrocyte cultures instead¹, where we show that astrocytes in mixed neuronal cultures propagate the prion strains RML and 22L, but not Me7.

Reviewer #3:

- *Authors responded well by adding neuronal cultures.*

- As to the strain dependent internalization routes, the reviewer suggested that this point

to be discussed in the paper. *As far as I can see this was not followed up.*

Response to Reviewer #3:

We thank this reviewer for acknowledging that Point 1 of reviewer #1 was fully addressed by providing additional data on the formation of fibril-like PrP^d aggregates in hippocampal neurons, following infection with mouse RML prions (Fig. 1I). Regarding the second comment by reviewer #3, we argued in our response to reviewer #1 and remain adamant that strain-dependent internalisation routes are not subject to and beyond the scope of this manuscript.

Point 2

Reviewer #1: - SAF32 recognizes cell surface and intracellular PrP^d: the explanation of differences in the octarepeat region is not clear – is it not that 5B2 does not recognize intracellular PrP^d because of N-terminal truncation – while possibly the SAF32 epitope is retained? Does SAF32 recognize N-terminally truncated PrP^d eg by immunoblot?

Authors' response: This reviewer requests clarification on the binding characteristics of Saf32, i.e. whether it recognises N-terminally truncated PrP on immunoblot and whether the Saf32 epitope is retained on truncated PrP. To clarify the binding characteristics of Saf32, we refer to the literature of this extensively characterised antibody in addition to our data. Saf32 has multiple binding sites in the octapeptide region² and recognises a truncated PrP fragment, C2, which is formed by beta cleavage³. In agreement with our study, Masujin et al. reported that Saf32 binds to abnormal PrP under native and denaturing conditions⁴. In the Masujin study abnormal PrP was isolated from BSE infected mice. In agreement with results from Rouvinski et al.⁵, we show that Saf32 recognises plasma membrane bound PrP under native conditions. Our data using myc-Prnp chimera in **Fig. S6** show that intracellular PrP^d can be detected with an anti-myc antibody in G70 myc-Prnp expressing cells, but not in G30 myc Prnp and G45 myc-Prnp expressing cells, suggesting that PrP^d is cleaved prior to the third octapeptide repeat. Our data thus confirm that Saf32 is able to detect N-terminally truncated PrP^d species. Importantly though, Saf32 binds to extracellular PrP^d under native conditions, while its binding site on intracellular PrP^d is cryptic (**Fig. 2E**), suggesting protein conformational changes.

Reviewer #3:

Saf32 stands out among other antibodies in that it reacts natively with cell surface PrP^d but requires denaturation to stain intracellular PrP^d.

This was construed by the authors as a proof of a conformational difference between cell surface and intracellular PrP^d. (“L160: Saf32 showed notable propensities, in that it detected extracellular fibril-like PrP^d under native conditions, like N-terminal mAbs and intracellular PrP^d under denaturing conditions, unlike N-terminal mAbs, suggesting that extracellular and intracellular PrP^d species are conformationally distinct at the octapeptide repeat region (Fig. 2E and Fig. S4).” Also L182)

Reviewer #1 raised 3 points:

a. He/she questioned the authors' interpretation (about conformational distinction) and suggested that the contrasting reactivities may just reflect the presence of the Saf32 epitope in full length PrP^d on the cell surface vs its removal by proteolysis in perinuclear PrP^d.

The authors responded by a description of the Saf32 octapeptide repeat epitopes, some of which may remain under the so-called beta cleavage of PrP. *I find this response satisfying*, and suggest that this description should be included in the paper, including a reference for the beta cleavage (eg Mange et al. 2004).

b. Reviewer #1 asks “does Saf32 recognize N-terminally truncated PrPd eg by immunoblot?”. I believe that additional immunoblots would probably not add much to the current ms, mainly because they would include both cell surface and intracellular PrPd, which differ in the extent of their N-terminus. *It's OK that the authors' didn't directly address that.*

c. *I disagree with the authors' assessment* that the fact that “Saf32 binds to extracellular PrPd_e under native conditions, while its binding site on intracellular PrPd_i is cryptic (Fig. 2E)” suggests “protein conformational changes”. That is because SAF32 staining of cell surface PrPd can occur via its N-proximal epitopes without indicating the extent to which the C-proximal octapeptide is cryptic, or not, in cell surface PrPd. *This was not raised by the original review and is a new point.* I suggest that this point be at least discussed in the ms.

Response to Reviewer #3:

This reviewer makes an excellent point here that has not been considered. Given that Saf32 has multiple binding sites along the octapeptide region (OPR) and is cleaved within the OPR in line with our data from myc-Prnp chimera (Fig. S7D), this reviewer argues that Saf32 binding to cell surface PrPd “*can occur via its N-proximal epitopes without indicating the extent to which the C-proximal octapeptide is cryptic, or not, in cell surface PrPd*”. In other words, there could be inherent conformational differences between N-proximal and C-proximal octapeptide repeats which may be invariant to limited proteolysis. We agree with this suggestion and have incorporated this possibility into the main body of the manuscript.

Changes to manuscript (added content is underlined):

Line 164: Saf32, an antibody that has multiple binding sites at the octapeptide repeat region (Yin, 2006 #10737), showed notable propensities,...

Line 167:extracellular and intracellular PrPd_i species are conformationally distinct at the octapeptide repeat region (Fig. 2E and Fig. S4). Alternatively, inherent conformational differences may exist between N-proximal and C-proximal octapeptide repeats that render C-proximal octapeptide repeats cryptic and invariant to limited proteolysis.

Notably, a considerable body of evidence shows that PrP is cleaved at the level of the OPR, giving rise to the C2 fragment (Mange, 2004 #8572; Lewis, 2009 #15458).

Discussion, Line 399: Notably, failure in detecting intracellular myc-tagged PrPd in persistently infected G30 and G45, but not in G70 PrP-expressing cells under denaturing conditions confirms that PrPd is N-terminally truncated at a site prior to G70 prior to the third octapeptide repeat (Figs. S7C and S8A, B).

Point 3

Reviewer #1: - Fig.3E: from the significant reduction of 5B2 signal it is concluded that a block of intracellular PrPd to the plasma membrane inhibits the formation of FL-PrPd; however, first, cholesterol extraction would affect lipid raft integrity at the plasma membrane, know to reduce PrPSc formation – is lipid raft perturbation an alternative explanation for the lack of 5B2 signal? Second, cholesterol extraction can release GPI-anchored proteins. It is recommended to include a control for the release of GPI-anchored proteins from the cell

surface, e.g. PrP^c.

Authors' response: We thank this reviewer for the suggestion to test whether the GPI-anchored protein PrP^c is extracted from the plasma membrane under experimental conditions of **Fig. 3E**, i.e. with β -methyl cyclodextrin (m β CD) concentrations < 1 mM. We treated confluent monolayers of uninfected S7 cells with m β CD (0.5, 5 and 50 mM) for 30 minutes at 37°C in agreement with a m β CD extraction protocols and determined the loss of PrP^c fluorescence (**Fig. 3G**). While a significant reduction of PrP^c levels were determined at 50 mM m β CD, no significant changes in PrP^c levels were detected at 5 and 0.5 mM m β CD (**Fig. 3G**). We thus conclude that extraction of GPI-anchored PrP^c does not explain the significant reduction of PrP^d levels at the plasma membrane in **Fig. 3E, F**.

We agree with this reviewer that the perturbation of lipid rafts is an alternative explanation for a reduction of colocalisation at the plasma membrane, following m β CD incubation and we have added this in Results, lines 250-252.

Reviewer #3:

- The authors now refer to a possible involvement of raft perturbation in the m β CD results: L243 "Alternatively to inhibition of vesicle transport, a perturbation of lipid rafts by m β CD could account for the reduced 5B2/6D11 colocalisation at the plasma membrane." *This is satisfactory. However, I suggest that they include an explanation as to how this is at all relevant.*

We thank this reviewer for confirming that our remark on the perturbation of lipid rafts is satisfactory. In response to this reviewer's suggestion, we have added a short phrase on the relevance of lipid rafts.

Changes to manuscript (added content is underlined):

Line 271: Lipid rafts are thought to play an important role in the formation and propagation of prions (Kobayashi, 2018 #24306; Lewis, 2011 #16407; Marshall, 2016 #22696).

- As to the PrP^c extraction: The authors have now treated uninfected cells to monitor whether PrP^c is extracted: L237 "To exclude the possibility that m β CD treatment leads to an extraction of PrP^c under the experimental conditions of Figure 3EF, we treated uninfected cells with m β CD concentrations of up to 50 mM (Figure 4G), confirming that PrP^c levels remain unchanged at m β CD concentrations below 5 mM." Since they mentioned "... the experimental conditions of Fig. 3EF,..." (is that a typo?), this reader assumed that treatment length was also 16h (thus a proper control). However, in their response above the authors claim a 30min extraction. Please specify in the ms the time frame of this experiment.

If this is indeed true, then this control is not acceptable.

We agree with this reviewer that the incubation time for this control experiment is critical. We therefore repeated this experiment and incubated cells for 16 hours with non-toxic m β CD concentrations of up to 2 mM and determined the level of surface PrP^c. As shown in Fig. 3G, none of the m β CD concentrations tested provides evidence of reduced PrP^c surface expression. At the lowest m β CD concentration, PrP^c levels were instead slightly elevated which might be due to an increased surface retention of PrP^c under mild cholesterol depletion. We have added this possibility to the main body in line 260 (see below). However, our result clearly counters the argument that the decrease in co-labelling upon cholesterol lowering with m β CD (Fig. 3E, F) can be attributed to an extraction of

GPI-anchored PrP^c.

Changes to manuscript (added content is underlined):

Line 264: A slight increase in PrP^c surface levels between 0.4 - 1 mM mβCD may be due to a higher cell surface retention of PrP^c.

Line 1287, figure legend Fig. 3G: (G) Dose-dependent effects of mβCD on PrPc levels extraction from at the plasma membrane during a 16h-incubation as specified in Methods. Mean values and SEM of three independent experiments are shown. Statistical significance was assessed by Student's T-test ANOVA with Bonferroni multiple comparison test with at least 148 images per condition analysed.

Point 4

Reviewer #1: - Fig. 5: Arf1, Arf6 and Cav1 knock-down and effects on PrPd: it is commented in Table S5 that Cav1 is not expressed in the used cells; therefore, it should be omitted from the graph. The knock-down of Arf1 and Arf6 on the mRNA level is only around 40 – 50% efficient; the question arises whether this level of knock down is sufficient to affect PrPd, and how it translates into a reduction of protein levels, which could be even less efficient depending on the half life of the proteins. It is suggested to include analysis of protein levels for these proteins.

Authors' response: This reviewer suggested that knockout efficacies of 40-50% for Arf1 and Arf6 may not suffice to exclude that prion steady state levels are affected by transcriptional silencing of said genes. We now used two novel custom-designed siPools with 30 distinct siRNAs against the 3'-UTR regions of Arf1 and Arf6 from siTools Biotech GmbH (Planegg, Germany); of note gene coding regions are the primary site for siRNAs at siTools Biotech GmbH. Knockdown efficacies of 80% (Arf1) and 90% (Arf6) were confirmed by qPCR (see Table S5). The average percent changes of prion steady state levels following gene silencing of Arf1 and Arf6 were 27% and 21%, respectively. We have updated these results in **Fig. 5E** and **Table S5**, including statistical analyses.

As suggested, we have also omitted caveolin 1 (Cav1) from bar chart in **Fig. 5E**. While it is prudent to check knockdown efficacies on RNA as well as on protein level, we do not agree that this is a feasible strategy for a gene candidate approach due to the scarcity of validated antibodies. We nevertheless sourced commercial antibodies against Arf6 (3A-1, Santa Cruz), Cdc42 (B-8, Santa Cruz) and Arf1 (ab58578, Abcam), however, with disappointing results, since in immunocytochemistry high unspecific background staining was observed for all sourced antibodies. Given the moderate effect changes of Arf1 and Arf6 knockdown on prion steady state levels in **Fig. 5E**, we do not think that the reporting of knockdown levels on protein level adds value.

Reviewer #3: They have opted to remove the data altogether, which is disappointing since they could have run western blots to assess the knockdown. However, I consider this response satisfactory.

In response to additional data provided in Fig. 5E and Table S5, showing highly effective knockdown of Arf1 and Arf6 with new siPools, this reviewer surprisingly notes that “they have opted to remove the data altogether, which is disappointing”. This is incorrect and may be a misunderstanding. Briefly, reviewer #1 correctly argued that knockdown

efficacies of 40-50% were too low to draw conclusions. We thus tested additional siPools and obtained knockdown efficacies of 80% (Arf1) and 90% (Arf6) (Table S5). We then repeated experiments to check effects of Arf1 and Arf6 knockdown on prion steady state levels and updated the data in Fig. 5E. We further omitted Cav1 from Fig. 5E, since reviewer #1 correctly noted that Cav1 was not expressed in the cells as mentioned in Table S5. In summary, no data were removed; data were instead updated for valid reasons. We thank this reviewer for considering the response to comments from reviewer #1 satisfactory.

2. Authors' rebuttal to Reviewer #2 critique

Response to "Major points"

Point 1

Reviewer #2: 1. Imaging of staining of PrP^d fibrils being made at the ECM (E.g., Fig 1C), using antibody against ECM-resident protein focal adhesion kinase (FAK) proteins is not provided. Because this is a key point to the study, this needs to be included.

Authors' response: We thank this reviewer for suggesting to provide data on how distance to the extracellular matrix (ECM) was estimated in serial images in z. We now include an image of a representative z-stack of cells, colabelled with mAbs against PrP and ECM-resident Focal adhesion kinase (FAK, Fig. S1F). As further explained in Methods, in-focus detection of FAK coincides with in-focus detection of PrP^d fibrils at the ECM and was arbitrarily denoted "zero μm ". This information was added to the figure legend in Fig. S1F and is further explained in Methods (lines 656-61).

Reviewer #3: In keeping with Reviewer #2 recommendation, the authors now provide a z-stack of PrP/FAK co-stained infected cells (Fig. S1F). *This is nominally a satisfactory response but the picture is low quality.* It is supposed to show that PrP^d fibrils appear at the same z depths as the FAK spots, but I am not seeing this. Please provide enlarged insets. Also, in the legend to this panel (L1187) you send the reader to Methods. Where in the Methods? What are we supposed to look for there?

In point 1, reviewer #2 requested to show, in supplementary data, how double-labelling of aggregated PrP^d and focal adhesion kinase (FAK) can be used to classify extracellular PrP^d. The authors added a representative z-stack of prion-infected cells that was double-labelled with FAK/PrP (Fig. S1F). However, reviewer #3 noted that it is not clear from the images that PrP^d and FAK spots are detected at the same z depths. We therefore reordered a FAK antibody (anti-FAK-Tyr925, Cell Signaling Technology, Cat#: 3284) and repeated double-labeling of FAK and PrP^d at the level of the ECM. We thus replaced Fig. S1F with a new image set of labelled cells showing the transition from blurred to in-focus and back to out-of-focus detection of either protein in Fig. S1F. Due to supply chain issues for sourcing the anti-FAK antibody pY397, we purchased an anti-FAK antibody from Cell Signaling Technology and corrected the specification of the antibody in lines 517 and 684 correspondingly. In the Fig. 1F legend, we furthermore specified the relevant Methods section "Image acquisition using laser-scanning microscopy" (Line 1412-3).

Point 2

Reviewer #2: Line 133: “In cultures from FVB wt mouse brains, fibril length continuously and significantly increased over three weeks, at mean fibril elongation rates of 190 nm per day, reaching lengths of up to 15 µm, while no fibrils were detected at the periphery of astrocytes from brains of FVB Prnp^{-/-} mice (Fig. 2A, B).” It is unclear why fibril length is measured in the periphery of neurons in FVB wt mouse brains, but in the periphery of astrocytes in brains of FVB Prnp^{-/-} mice? It appears PrP^d fibrils are only around astrocytes not primary neurons?

Author’s response: In regards to this reviewer’s question whether PrP^d fibrils are not observed in primary neurons, we refer to new data of PrP^d fibrils which are associated with Map2- positive neurons (**Fig. 1I**), added to this manuscript in response to comments from reviewer #1. This additional result confirms that Map2-positive neurons replicate prions and form fibrillike PrP^d aggregates that can be labelled with mAbs against the PrP N-terminus.

Reviewer #3: *This is satisfactory.* However, *the new panel I is not explained in the text.* Re the text and the legend: when was the labeling done post infection? At what point in the establishment of the culture were the cells infected? Why did you choose RML here vs 22L for the astrocytes?

We apologise for omitting to explain the new panel in the text; this is now added.

Changes to manuscript (added content is underlined):

Line 126: When glial proliferation in primary neuronal cultures was inhibited with 1 µM cytosine arabinoside (AraC), followed by infection with RML or uninfected CD1 control homogenates, Map2-positive neurons with fibril-like PrP^d aggregates were visible upon RML infection (Fig. 1I), demonstrating that this fibrillar PrP^d phenotype is prevalent in neurons as well.

Figure legends, Line 1236: (I) Primary hippocampal neuronal cultures from embryonic e17 mouse brains were isolated, cultured for 6 days and incubated with 1µM AraC. The following day, cells were and infected with a 10⁻⁵ dilution of RML (10%, w/v) and uninfected CD1 (10%, w/v, mock-infected) brain homogenate, respectively. Three weeks after infection, cells were ~~Fixed cells were~~ and double-labelled with 5B2 and anti-Map2.

Reviewer #3 further queries why neurons were infected with RML, whereas astrocytes were infected with 22L. Notably, a further question regarding prion strains was raised by reviewer #4 in “minor points” on page 32 below and this reviewer suggested using RML consistently for both neural cell types. We have followed this suggestion, repeated the infection of primary neuronal cultures with RML in absence of AraC and report changes to the manuscript under minor points 4 below (page 33).

Point 3

Reviewer #2: Fig.2C: in the co-labeling experiment, it is difficult to appreciate the degree of localization or not, as the figure is provided only as a merge of two channels. Separation of each channel in black/white (inverse could help), in addition to the merge would facilitate evaluation of the observations/conclusions presented with these data.

Response: Point 3: We greatly appreciate the reviewer’s suggestion of displaying the two channels of the SIM image as greyscale in addition to the merged image to better

appreciate the degree of co-localisation. We have amended this figure accordingly. To accommodate the additional space in Fig. 2, we deleted the second SIM image set in Fig. 2Cb.

Reviewer #3: *Satisfactory*

We thank this reviewer for acknowledging that our response to point 3 is satisfactory.

Point 4

Reviewer #2: 4. Line 142: Data presented to suggest “that different aggregation states may co-exist during PrP^d fibril formation” is based on light microscopy imaging. Biochemical characterization of these different PrP^d aggregate states using antibodies used in these imaging studies would provide important orthogonal validation.

Response: Point 4: While we agree with this reviewer that biochemical characterisation of different PrP^d aggregation states using antibodies would provide further validation, all our attempts to isolate fibrillar PrP^d from persistently infected cells following cell lysis or by immunoprecipitation using anti-PrP mAbs failed. This result is in agreement with failure in detecting 5B2-positive fibrillar PrP^d on ELISA plates by Scrapie cell assay after cell trituration (data unpublished). After careful inspection, we found that fibrillar PrP^d strongly adheres to the ECM and remains bound to plastic ware and glass slides following trituration of cells, as shown in the new supplementary figure, **Fig. S6B**. Cell lysis under stringent conditions did not change this outcome. To follow-up this outcome and to further investigate whether fibrillar PrP^d is lost after triturating cells, we imaged resuspended cells in a time-dependent manner (**Fig. S6C-F**). Unexpectedly, we were unable to detect full-length PrP^d with the N terminal antibody 5B2 before 3 days after resuspension of cells (**Fig. S6F**), suggesting that FL-PrP^d is fully degraded, presumably following rapid internalisation. That FL-PrP^d is rapidly degraded after resuspension of cells was indeed confirmed by blocking proteolysis (**Fig. S6G-J**). Pre-incubation of persistently infected cells with the V-ATPase inhibitor BafA1, a treatment that increases luminal pH levels, led to detection of intracellular FL-PrP^d (**Fig. S6H**). This result confirms that full-length and truncated PrP^d species greatly differ in their rate of formation, a result that corroborates our data on the fast kinetics of PrP conversion versus lagging formation of FL-PrP^d (see **Fig. 6F**). That BafA1 gives rise to intracellular FLPrP^d substantiates our evidence of the pH-dependence of proteinases activation (see Point 5 below). To our knowledge this is the first evidence for chaperone-independent fibril fragmentation in mammalian cells. Fibril fragmentation was thus far only described in yeast cells. We report this experiment in lines 191-215, since it strongly corroborates the notion of intracellular processing of FL-PrP^d (**Fig. 2F**), which is relevant for Point 5 below.

Reviewer #3: The authors explain that they were unable to isolate the fibrillar structures at the ECM (ECM presumably meaning the contacts with the plastic dish). This is because the fibrils stay attached to the substrate following “trituration”. However, they do not explain what are the experimental conditions of the trituration, or of the “Cell lysis under stringent conditions” that “did not change this outcome”, so that I cannot relate to the experiment. What happened to the cell lysate in these experiments? Did it not contain any full length PrP^d? I agree that images of fibrillar structures attached to the plastic substrate (Fig. S6B) are very interesting, but the Reviewer #2 expressly asked for biochemical characterizations that should have been at least attempted using cell

lysates.

In my opinion, the authors' response to this point is not satisfactory.

I also disagree with the last paragraph of their response, since it is based on the assumption that the cell surface fibrillar assemblies are just plain amyloid fibrils (or rods as they are called elsewhere in the paper). As far as I could see, they do not provide any proof for that and this interpretation seems to be at odd with previous data on “prion strings” (ref 14) (that seem to have been renamed here to “fibrillar structures”.) *However, this remark was not included in the original Review.*

In Point 4, reviewer #2 suggested “*biochemical characterization of these distinct PrP^d aggregates*”. This point refers to our identification of phenotypically distinct PrP^d aggregates in cells, identified by monospecific discriminatory antibodies. In brief, punctate PrP^d aggregates were identified at intracellular sites and fibril-like PrP^d aggregates are tethered to the plasma membrane (see Figs. 1 and 2). In response to reviewer #2, we reported that we tried, but failed to isolate fibrillary PrP^d which “*strongly adheres to the ECM and remains bound to plastic ware and glass slides following trituration of cells (Fig. S5B)*”.

Reviewer #3 requested further experimental detail regarding our pursuit of isolating PrP^d fibrils. We have now included additional experimental detail to Fig. S5, Methods and the main body of the manuscript as detailed below.

To investigate whether PrP^d fibrils, embedded into the ECM of infected cells can be isolated following decellularisation, we used detergent lysis followed by immuno-isolation with the anti-PrP antibody ICSM18 (epitope 143–153). Briefly, persistently infected S7 cells were grown to confluency in 10 cm dishes. While cells were lysed with high-stringency buffer (RIPA) in one set of plates to provide control cell lysates, cells were resuspended in distilled water in a second set of plates and cells were discarded. The remaining decellularised ECM (dECM) was then incubated with RIPA buffer for 15 min, followed by resuspension. Lysates from both sets were immuno-precipitated with ICSM18 using Miltenyi μ MACS Protein G magnetic beads, washed and eluted in sample buffer. While ICSM18 effectively immunoprecipitated PrP-specific bands from cell lysates, we failed to reveal PrP-specific bands from dECM lysates (Fig. S5C). As a control experiment, persistently infected cells were grown in 8-well chamber slides. After cells reached confluency, cells were resuspended under microscopic control to ensure complete decellularisation. One set of chamber slides was incubated with PBS, a second set of chamber slides was incubated with RIPA buffer. As depicted in Fig. S5D, this procedure failed to solubilise PrP^d aggregates from the glass. In summary, our additional data shows that stringent detergent lysis of dECM failed to isolate measurable quantities of immuno-isolated PrP^d on Western blot. Importantly, in response to the request for “biochemical characterization of PrP^d aggregates” from reviewer #2, failure of isolating fibrillar PrP^d has prompted us to further investigate the turnover of fibrillary PrP^d after resuspension of cells and report “asynchronous changes in intra- and extracellular PrP^d pools after cell dissociation” in lines 194 f and Fig. S5 E-H.

Changes to manuscript (added content is underlined):

Line 201: Attempts to immuno-isolate PrP^d fibrils from decellularised ECM (dECM) following stringent lysis with RIPA buffer failed (Fig. S5C). Under these conditions, 5B2-positive fibrils were still visible at the dECM (Fig. S5D). Unexpectedly, when cells were resuspended and replated according to Fig. S5A, 5B2-positive PrP^d aggregates were absent in cells within.....

Lines 1465-71, Fig S5C/D legend: (C) Following decellularisation of infected S7 cells, the remaining ECM was lysed with RIPA buffer (dECM) for 15 min as specified in Methods. As control, confluent layers of infected S7 cells were lysed in RIPA buffer (cells). Lysates were incubated with primary antibody ICSM18 and isolated with μ MACS Protein G microbeads (Miltenyi). Western blots were labelled with biotinylated ICSM35, followed by avidin/biotin ABC complex (see Methods).

(D) Following cell trituration of infected S7 cells in 8-well chamber slides, dECM was resuspended in cold RIPA buffer and PBS (control), followed by labelling with 5B2.

Lines 865-83, Methods, Immunoprecipitation: Prion-infected S7 cells were grown to confluency on 15 cm dishes for 5 days with half medium changes on days three and four.....(see Methods).

In the second paragraph of his/her response, reviewer #3 addresses the reported turnover of PrP^d fibrils and our suggestion that “*to our knowledge this is the first evidence for chaperone-independent fibril fragmentation in mammalian cells.*” We here refer to additional comments of reviewer #3 on page 25 where we address detailed questions of reviewer #3 regarding the phenotype of fibrils and amyloid and where we correct our suggestion of a “chaperone-independent fibril fragmentation” (see page 25-27, Q1-5 and changes to manuscript in line 473).

Point 5

Reviewer #2: A very major deficiency in the manuscript with consequences to the rest of the manuscript starting with lines 165-185 is the following: The observation and conclusion that 5B2-positive aggregates in cells treated with BafA1 become visible due to blockage of proteolytic cleavage of full-length PrP^d within less acidifying endolysosomes is not substantiated with data. First, there is no evidence provided that PrP^d is inside or associated with endosomes (also see point 13 below). Second, evidence for proteolytic cleavage of PrP^d is completely lacking so this cannot be concluded, therefore calling this fragment “truncated (TR) PrP^d” is total speculation. Both light microscopy imaging as well as biochemical characterization of PrP^d within endolysosomes and of proteolytic cleavage of PrP^d in those endosomes (for example using endosome purification by sucrose gradient centrifugation) is required to substantiate these claims. BafA1 treatment alone does not indicate cleavage.

Response: The reviewer here claims that the authors have not provided experimental evidence that intracellular PrP species are proteolytically truncated and suggests that “biochemical characterisation” of PrP^d in isolated endolysosomes will help to clarify this.

We strongly disagree with the reviewer’s opinion as elaborated below. In this study we provide three lines of evidence that intracellular PrP^d species are truncated: (1) Three mAbs against the N-terminus of PrP, AG4, 8B4 and 5B2 recognise PrP^d at the plasma membrane, but fail to detect intracellular PrP^d (**Fig. S4 and Fig. 2E**); that intracellular PrP^d is present under these conditions is demonstrated by double-labelling with 6D11, an antibody that binds to “core” PrP⁷ (epitope 93-109, **Fig. 1D,F,H and Fig. S4**). (2) An anti-myc mAb detects intracellular PrP^d only in cells where the myc tag has been cloned into the octapeptide region (G70 myc Prnp), but not in those where it has been cloned downstream of N-terminal codons 30 and 45 (**Fig. S7D**), suggesting that the site of

proteolysis is prior to the third octapeptide repeat. (3) That lowering of the intracellular pH by lysosomotropic amines, including NH₄Cl, chloroquine and quinolone derivatives blocks proteolytic processing of PrP has been widely demonstrated⁸⁻¹³ using Western blotting and biochemical assays. By blocking proton shuttle across membranes, BafA1, a specific inhibitor of V-ATPases¹⁴ has the same net effect as lysosomotropic amines, i.e. an increase in the luminal pH. Our study provides the first evidence that full-length PrPd can be detected intracellularly only after treatment of cells with NH₄Cl and BafA1 (**Fig. 2F** and in the new figure, **Fig. S6** (see Point 4)). This data unequivocally demonstrate that intracellular PrPd is proteolytically processed. We thus strongly oppose the notion of this reviewer that truncated PrPd is “total speculation”, since experimental proof in this study is evident using different experimental paradigms. With this in mind, we think the reviewer’s suggestion of isolating “endolysosomes” is “shifting the goalposts” of a study that provides novel insight into the cellular trafficking of truncated PrPd on a whole cell context (see lines 429-434 for outcomes), a result that cannot be achieved with a *pars pro toto* approach as suggested by the reviewer. Finally, our new data confirming that PrPd segregates into vesicular pathways (Point 9 and **Fig. 4H-J**) questions the value of isolating “endolysosomes”.

Reviewer #3: Reviewer #2 asked that “Both light microscopy imaging as well as biochemical characterization of PrPd within endolysosomes and of proteolytic cleavage of PrPd in those endosomes.” The authors strongly opposed that view, arguing that their immunofluorescence data with a variety of antibodies, together with the data with bafilomycin, NH₄Cl, etc sufficiently indicate that intracellular PrPd is usually truncated, unless acidic proteases are inhibited.

Reviewer #2 also asked for an independent assessment that PrPd in endolysosomes is indeed truncated, but becomes full length when acidification is prevented. Here, the authors argued “we think the reviewer’s suggestion of isolating “endolysosomes” is “shifting the goalposts”...”, especially considering their data “confirming that PrPd segregates into vesicular pathways (Point 9 and **Fig. 4H-J**)”, which “questions the value of isolating “endolysosomes”.

Even though it is probable that the authors’ conclusions (that PrPd is truncated by acidic proteases in endolysosomes) may turn out to be true (as hinted by studies from the 1990s), I stand by Reviewer #2 demands here.

Simple immunoblots with relevant antibodies would go a long way to prove that: a. Intracellular PrPd is usually truncated, but becomes full length with acidification inhibitors, and b. that the PrPd species involved is indeed aggregated PrPd (which is presumably the same as PrPSc?), and not PrPC, for instance. Admittedly, they would have to use established methods in the field such as sedimentation of detergent lysates, etc, but this would help to shore up their conclusions.

In my opinion, this issue illustrates *Reviewer #2*’s critique that “major deficiencies are noted that relate primarily to insufficient or deficient or non-existent data that authors rely on (or not) to make the major conclusions made throughout the manuscript.”

The same is true for Reviewer #2’s other request, that endolysosomes be isolated using established procedures and their PrP content analyzed. In my opinion, that is just a proper control that has nothing to do with “shifting the goalposts”, but rather should have

been included in their original experiments.

I also don't agree with their claim that the fact that PrP^d was detected elsewhere in the "vesicular pathway" (is that the endomembrane system?) "questions the value of isolating "endolysosomes". Quite the opposite, this would help ascertain that the truncation takes place in acidic and hydrolytic compartments, to the exclusion of other vesicles that they have detected in this respect.

Therefore, I don't feel that their rebuttal arguments are solid. In my opinion, their answer to this point are completely unsatisfactory.

In response to Point 5, reviewer #3 suggests the authors provide immunoblots "to prove that a. intracellular PrP^d is usually truncated, but becomes full length with acidification inhibitors and b. that the PrP^d species involved is indeed aggregated PrP^d and not PrP^c."

We have now investigated changes in the levels of PrP^{Sc} upon BafA1 treatment on Western blots as elaborated below. However, we first would like to respectfully note that the assumption that immunoblots will "prove that intracellular PrP^d is usually truncated, but becomes full length with acidification inhibitors" is somewhat misleading. This is, because in absence of acidification inhibitors, persistently infected cells accrue considerable amounts of full-length PrP^d at the plasma membrane (See Fig. 1C, D and Fig. 2A, E) and thus cell lysates of said cells will contain truncated as well as full-length PrP^d. Inhibition of acidification will thus not result in a "band shift", as implied, but rather in a "band intensity shift", since following detergent lysis of cells, all PrP species are present in both, vehicle control- as well as inhibitor-treated cells.

To report our experimental approach, we first treated lysates of uninfected (S7) and infected S7 (iS7) cells with PNGase, an enzyme that quantitatively deglycosylates PrP. As shown in Fig. S5I, prion propagation in S7 cells gives rise to an additional low molecular weight band representing truncated PrP, denoted "TR" in Fig. S5I. For Western blots in Fig. S5, we used ICSM35, an anti-PrP antibody which reacts with PrP epitope 91-110¹. Furthermore, cell lysate from *Prnp*-knockout cells (KO) was used to identify an off-target band with a molecular weight of approximately 49 kDa.

We then treated confluent layers of S7 and iS7 cells with 0.5 mM BafA1 or vehicle control (DMSO) for 1 hour. Cells were washed and cultured for an additional 16 hours in a CO₂ incubator. Cells were then lysed with RIPA buffer (50 mM Tris, 150 mM sodium chloride, 0.5% sodium deoxycholate, 1% Triton (v/v), pH 7.4) at 4 °C. Where indicated, cell lysates were incubated in absence or presence of 10 µg Proteinase K (PK)/ml PBS to reveal PK-resistant PrP (PrP^{Sc}). As shown in Fig. S5J, left panel, BafA1 treatment resulted in a higher intensity of all full-length (FL-) PrP glycoforms in addition to unglycosylated PrP in cell lysates when PK digestion was omitted. When quantified over three independent experiments, BafA1 treatment changes the levels of FL-PrP to TR-PrP by a factor 3:1 (Fig. S5K). PK digestion of lysates from infected S7 cells (Fig. S5J, right panel), on the other hand led to a characteristic band shift to lower molecular weight bands due to PK-dependent proteolysis, while control lysates from uninfected S7 cells (band 5 in Fig. S5J) were fully digested. This result provides unequivocal evidence that PrP^d bearing cells contain PrP^{Sc} in both, DMSO (mock-) and BafA1-treated infected cells. As evident in PK digested cell lysates, treatment with BafA1 again led to an increase in PrP^{Sc}. This shift to higher PrP^{Sc} aggregation levels is specific to prion-infected cells, since BafA1 treatment of uninfected cells did not result in an increase of band intensities (Fig. S5L). These results

are in agreement with an increase in 5B2-immuoreactive PrP^d aggregates, observed by laser scanning microscopy after inhibition of luminal acidification by BafA1 (Fig. S5G, H). Of note, the quantitative changes in band intensities of FL-PrP and TR-PrP upon BafA1 in Fig. S5J,K are in agreement with a shift from TR-PrP to FL-PrP, even though the relative level of TR-PrP seems unchanged (i.e. 1). Importantly though, relative protein levels on WB do not inform on reverse transitions from FL-PrP back to TR-PrP which are dependent on relative FL-PrP levels. Thus, the pools of FL-PrP and TR-PrP are independent and affected by distinct proteolytic activities. Notably, while pH-dependent proteases, like serine proteases are blocked by BafA1 inhibition, the activation of metalloproteinases such as MMP-2 and MMP-9, various ADAM (A Disintegrin And Metalloproteinase) family members as well as Nephilysin are pH-independent.

Changes to manuscript (added content underlined):

Line 218ff: Prion propagation in S7 cells is associated with proteolytic processing of PrP, giving rise to truncated PrP conformers as evident in cell lysates after deglycosylation with PNGase (Fig. S5I); bands, representing full-length (FL-) and truncated (TR-) PrP are denoted. As evident on Western blots of cell lysates, BafA1 treatment of infected (Fig. S5J), but not of uninfected cells (Fig. S5L), led to higher levels of FL-PrP. Proteinase K (PK) digestion of cell lysates from infected cells gave rise to PK-resistant PrP (PrP^{Sc}), while PrP^c in cell lysates from uninfected cells was fully digested (band 5 in Fig. S5J). Notably, BafA1 treatment of infected cells resulted in higher levels of PrP^{Sc}, when compared to mock-treated cells.

Further, two method descriptions for cell lysis/PK digestion and Western blotting were added in Methods.

Lines 842-50, Methods: Cell lysis and proteinase K digestion

Lines 851-64, Methods: Western blotting

Lines 1493-1509, legend Fig. S5

(I) Lysates of confluent Prnp^{-/-} (KO), uninfected (S7) and infected S7 (iS7) cells were prepared and treated in presence and absence of PNGase F as specified in Methods. Lysates were separated on 12% Bis-Tris gels, blotted and labelled with ICSM35, followed by AP-conjugated anti-mouse antibody. Arrows depict full-length (FL) and truncated (TR) PrP.

(J) Cells (KO, S7 and iS7) were incubated with 0.5 mM BafA1 or DMSO (0.05 %) for 1 h, washed with complete medium and further incubated for 16 h. Cells were lysed in RIPA buffer and cell lysates treated in absence and presence of 10 µg/ml Proteinase K (PK) for 30 min at 37 °C. Protein digestion was stopped with AEBSF and protein lysates were separated and blotted as described above. Following development of blots with ICSM35, blots were stripped with Restore stripping buffer and reprobbed with anti-mouse actin as loading control.

(K) Quantitative analysis of relative band intensities after BafA1 treatment, compared to vehicle control in cell lysates without PK digestion. Combined band intensities for FL- and TR-PrP bands as denoted by the bracket in image K were determined. Mean values + Stdev of relative band intensities from three independent experiments are shown.

(L) Uninfected S7 cells were incubated with 0.5 mM BafA1 or DMSO (0.05 %) for 1 h, washed with complete medium and incubated for 16 h. Cells were further processed as described in J.

To address the isolation of endo-lysosomes of BafA1- and mock-treated cells, as suggested by Reviewers #2 and #3, we isolated lysosomes by immunoprecipitation using

an anti-Lamp1 antibody (ab24170, Abcam) as described previously by Stahl-Meyer et al². Notably, isolated lysosomes from density gradients are often considerably contaminated with mitochondria and peroxisomes and alternative methods, including immunoprecipitation or Lyso-IP are a preferred option³. Full detail of the isolation of lysosomes is given in Methods (Line 884-903). Our experimental approach yielded a 4-5 fold enrichment of Lamp1 (Fig. S5N). Briefly, uninfected and prion-infected S7 cells were grown to confluency in 15 cm dishes and treated with 0.5 mM BafA1 or DMSO (0.05%) for 1h. The cells were washed with medium and incubated for an additional 16 h, in analogy with the previous protocol (Fig. S5J-L). Cells were then resuspended in isolation buffer (10 mM HEPES, 70 mM sucrose, 210 mM mannitol, pH 7.5), sheared with 25-gauge needles and followed by immunoisolation of lysosomes using a polyclonal anti-Lamp1 antibody, essentially as described in Methods (“Lysosome isolation”, lines 884-903). Enriched lysosomes from vehicle-treated cells showed a dominant low-molecular weight band (band 3, Fig. S5M), representing TR-PrP, while enriched lysosomes from BafA1-treated cells presented two dominant glycosylated FL-PrP bands (band 4, Fig. S5M). Of note, under the experimental conditions used for lysosome isolation, BafA1 treatment led to a complete band shift to full-length PrP with no truncated PrP detected, a result that is apparent in the input (band 2) as well as in the IP (band 4). This has not been observed in RIPA-buffer containing cell lysates and is likely due to the detergent-free isolation buffer used for lysosome isolation. On-column PK digestion as described in Methods chapter “Lysosome isolation” digested PrP^c completely, while evidence of PrP^{Sc} in enriched lysosome fractions of persistently infected cells is apparent (Fig. S5O). Please note that conditions for PK digestion were kept constant throughout the manuscript (see “cell lysis and proteinase K digestion” in Methods) which resulted in over-digestion of enriched lysosomes on μ MACS columns. However, that BafA1 treatment leads to higher PrP^{Sc} levels is apparent in Fig. S5O.

Changes to manuscript (added content underlined):

Lines 226f: Since endolysosomes have been proposed as a potential site where abnormal PrP is N-proximally truncated⁸, we enriched lysosomes of BafA1 and vehicle-treated cells by immunoprecipitation (Fig. S5M). A band shift from TR-PrP to FL-PrP upon BafA1 treatment is apparent in Lamp1-enriched lysosomes (Fig S5M).

Lines 884-903, Methods, Lysosome isolation

Lysosome isolation

Lysosomes were isolated by Lamp1 immunoprecipitation according to Stahl-Meyer et al.⁷⁵ with minor modifications. Briefly, uninfected and prion-infected S7 cells were grown to confluency in 15 cm dishes and treated with 0.5 mM BafA1 or DMSO (0.05%) for 1 h. Cells were washed with OFCS and further incubated for 16 h. Cells were then washed with cold PBS, resuspended in 10 ml PBS and centrifuged at 500 x g for 5 min at 4 °C. Cell pellets were resuspended in isolation buffer (10 mM HEPES, 70 mM sucrose, 210 mM mannitol, pH 7.5) and centrifuged at 1,000 x g for 5 min at 4 °C. This wash step was repeated once more. Cells were then lysed in 700 μ l isolation buffer, supplemented with 2.5 U Benzonase/ml, 0.5 mM Tris(2-carboxyethyl)phosphine hydrochloride and 7 μ l Halt Protease and phosphatase inhibitor (Cat# 78440, Thermo Fisher Scientific) by syringing the cells 15 times through a 25-gauge needle and centrifuged at 1,500 x g for 10 min at 4 °C. The supernatant was carefully collected and centrifuged again at 1,500 x g for 10 min at 4 °C. The protein concentration of post-nuclear supernatants was determined by Pierce BCA assay (Cat#: 23225, Thermo Fisher Scientific). Equal amounts of protein were

incubated with 3 μ l polyclonal Lamp1 antibody (ab24170, Abcam), pre-bound on 50 μ l Miltenyi μ MACS protein G beads (Cat# 130-071-101, Miltenyi Biotech) and shaken overnight at 4 °C. MACS columns were rinsed with isolation buffer and mounted onto a magnetic separator. The immuno-precipitate was captured on MACS columns, washed with isolation buffer and eluted with pre-heated SDS sample buffer. For on-column PK digestion, enriched lysosomes were incubated with 10 μ g PK/ml PBS. MACS columns were sealed with parafilm and digested for 30 min at 37°C. Protein was eluted with pre-heated SDS sample buffer.

Lines 1510-18, Legend Fig. S5:

(M) Prion-infected S7 cells were incubated with 0.5 mM BafA1 or DMSO essentially as described in J and lysosomes were enriched by immunoprecipitation using anti-Lamp1 antibody ab24170 (Abcam) as specified in Methods. Following immunoblotting, PrP was revealed with biotinylated anti-PrP antibody ICSM35 as described in Methods.

(N) Relative enrichment level of Lamp1 following immunoprecipitation with anti-Lamp1 antibody ab24170 is depicted.

(O) On-column PK digestion of BafA1- or DMSO treated uninfected (S7) and prion-infected (iS7) cells. For technical details, see “Lysosome isolation” in Methods. Levels of PrP^{Sc} were revealed with biotinylated anti-PrP antibody ICSM35.

Point 6

Reviewer #2: There are no experiments in the manuscript demonstrating that PrPd recognized by any of the antibodies tested are detecting “nascent” PrPd (versus non-nascent?). Thus, this claim as well as the one on point 5 above, i.e., lines 179-182 “These results suggest existence of two distinct aggregated PrPd species in prion-infected cells, nascent FL-PrPd which is associated with fibril elongation at the plasma membrane and a truncated (TR-) PrPd type at perinuclear sites.” are not substantiated with data.

Authors’ response: By referring to lines 179-182, this reviewer notes that there is no experiment that justifies the term “nascent PrPd” at the plasma membrane and we fully agree with this notion and are grateful to the reviewer for pointing this out. While the term “nascent PrP^{Sc}” has been used by others to denote *de novo* formed PrPd, like Rouvinski et al.⁵, our data show that *de novo* PrPd is distinct from plasma membrane resident fibrillar PrPd (**Fig. 6E,F**). We have therefore deleted the term “nascent” in this manuscript.

Reviewer #3: *This is satisfactory.*

This reviewer acknowledges that our response to point 6 is satisfactory.

Point 7

Reviewer #2: The claim of “crypticity” is also not substantiated with data to indicate that antibodies are or not binding to binding sites under particular conditions tested (\pm GTC), and on iS7 cells.

Authors’ response: This reviewer notes that the claim of “crypticity” is unsubstantiated

and we disagree with this comment. As shown in **Fig. S4** and **Fig. 2E**, only mAbs against the PrP N-terminus (5B2, 8B4, AG4 and Saf32) detect abnormal PrP (PrPd) under native conditions, whereas all other mAbs against “core PrP”⁷ do not. Hence, epitopes of these mAbs are cryptic under native conditions. Note that first evidence of the crypticity of antibody binding sites in abnormal PrP was reported as early as 1990¹⁵ and the authors suggested that denaturing conditions are required for antibody binding. We show that incubation of fixed cells with the denaturing agent GTC exposes epitopes that were cryptic under native conditions (**Fig. 2E**). In summary, the phenomenon of crypticity is fully characterised for all mAbs used in this study. Antibody epitopes that are cryptic on abnormal PrP are listed in **Fig. 2E**. Our model for cryptic binding sites (**Fig. 2G**) is in agreement with Rouvinski et al.⁵.

Reviewer #3: OK

This reviewer is fine with the response to point 7.

Point 8

Reviewer #2: Lines 186 paragraph: No evidence is provided to show that PrPd as recognized by antibody 6C11 and that is close proximity to PrPd as recognized with antibody 5B2, reaches the plasma membrane from the inside of the cell or if it is on the outside of the cell. No plasma membrane marker staining is shown, so it is impossible to tell where 6C11 or 5B2 PrPd are.

Authors' response: This reviewer suggests that “*no evidence is provided to show that PrPd as recognized by antibody 6D11 [corrected from “6C11”] and that is close proximity to PrPd as recognized with antibody 5B2, reaches the plasma membrane from the inside of the cell or if it is on the outside of the cell. No “plasma membrane marker” staining is shown, so it is impossible to tell where 6D11[corrected from “6C11”] or 5B2 PrPd are.*” In response, we would like to refer to **Fig. 3C**, where we use an antibody against the “plasma membrane marker” integrin $\beta 1$ to show that 6D11-positive PrPd is detected proximal to and at the level of the plasma membrane. However, we agree that this does not clarify whether 6D11-positive PrPd is transported from the cytosolic side to the plasma membrane. We now, however, provide unequivocal evidence for the directional transport of intracellular PrPd to the plasma membrane via synaptic vesicles, following potassium chloride-evoked depolarisation. This experiment is described in full in Point 9 below.

Reviewer #3: (Re paragraph starting L211).
OK.

We are grateful to this reviewer for acknowledging that our response to point 8 is satisfactory.

Point 9

Reviewer #2: The extent of colocalization of Synapsin 1 and PrPd on Fig. 3C is not clear from the image, which in fact by eye, show most instances of non-colocalization. There is no quantification provided or co-localization analysis done. Again, as noted in point 5

above, there is no evidence in this manuscript demonstrating that PrPd is associated within or with vesicles.

Authors' response: This reviewer notes that inspection of representative images of Synapsin 1 with PrPd in **Fig. 3C** does not provide evidence of colocalisation and concludes that evidence that PrPd is associated with synaptic vesicles is not provided. Firstly, we would like to note that **Fig. 3C** is a representative image and Pearson correlation coefficients, a widely acknowledged and used metric for assessing the colocalisation of proteins are shown in **Fig. 4G** and represent background-corrected mean values of 20-40 images. We thus suggested that this provides evidence that PrPd reaches the plasma membrane by vesicular transport (Line 200). We now provide further evidence that PrPd reaches the plasma membrane by vesicular transport. We subjected persistently prion-infected cells with KCl concentrations of up to 100 mM, a treatment that leads to fast depolarisation, followed by calcium influx¹⁶. Under these conditions Syn1-positive vesicles rapidly fuse with the plasma membrane (**Fig. 4H**), while 6D11/5B2 co-labelling at the plasma membrane significantly increases (**Fig. 4I, J**). This provides unambiguous evidence that PrPd segregates into the vesicular pathway and reaches the plasma membrane by means of regulated exocytosis. To accommodate for additional **Fig. 4H-J**, we deleted what was formerly **Fig. 4F**. This experiment is described in lines 269-74 in the manuscript.

Reviewer #3: I agree with the authors that the statistical analysis sides with PrPd being associated with synapsin 1 vesicles. The KCl experiments indeed add value to this contention. However, I agree with Reviewer #2 that PrPd could still be attached to the cytosolic side of the vesicles “there is no evidence in this manuscript demonstrating that PrPd is associated within or with vesicles”. It seems to me that this possibility could at least be could be mentioned in the text.

However, it is also possible that Reviewer #2 directed to another co-localization problem in the experiments summarized in Fig. 4G: can one derive from your data (here or elsewhere in the paper) what is the relative amount of PrPd in each of these vesicular locations? Such data is crucial to evaluate the power of the trafficking scheme in panel F. In other words, what is the fraction of the total PrPd found in each of these vesicles? How much in synaptic vesicles? In endolysosomes? Assuming that this was the original critique, then it is crucial that the authors address that at least in the text.

However, the KCl results (Fig. 4I) raise a major problem in my mind. As far as I can see, following this treatment there is a huge amount of 5B2-reactive PrP that now appears on the cell surface. And yet, the authors' interpretation is that intracellular PrPd is externalized on the cell surface (this is a test to see if “intracellular TR-PrPd segregates into regulated exocytosis” L263). How can this happen? Is it possible that this is PrPC that is quantitatively externalized by the KCl treatment? Have the authors run controls with uninfected cells? It is imperative that the authors address respond to this question. Of note, Reviewer #2 could not have raised this question, since these are new experiments that were not available in the original manuscript (as fa as I know).

We thank this reviewer for supporting our conclusion that PrP^d is associated with synaptic vesicles, based on Pearson correlation coefficients (Fig. 4G) and on rapid changes in PrP^d levels at the plasma membrane following KCl-evoked depolarization (Fig. 4I,J). In this context, this reviewer raises a couple of important points that we address below. Firstly, this reviewer notes that he/she agrees with reviewer #2 that PrP^d could still be attached to the cytosolic side of the vesicles. We fully agree that our experimental data

does not inform on the orientation of PrP^d on vesicles, however, for clarity, we neither address nor specifically mention the orientation of PrP^d in vesicles in this manuscript. In addition, we used the more generic term “overlapping distributions” in Fig. S6: “Identification of overlapping distributions of PrP^d with organelle markers to map intracellular trafficking routes of PrP^d”.

Secondly, this reviewer queries whether colocalisation data of PrP^d with organelle markers provide information on the quantitative distribution of PrP^d between organelles. This is not the case, since Pearson correlation coefficients provide information on the degree of colocalisation between two labelled proteins exclusively rather than on relative protein amounts in a specific organelle, represented by an “organelle marker”. In other words, Pearson Correlation Coefficients inform on proximity of proteins, not on their quantities. Thirdly, reviewer #3 further addresses the KCl-evoked depolarisation experiment in Fig. 4H-J. Briefly, in line 288 of the manuscript we report that “depolarisation led to a fast depletion of Syn1-positive synaptic vesicles (Fig. 4H) and a significant increase in PrP^d aggregates at the plasma membrane (Fig. 4I,J), confirming that PrP^d reaches the plasma membrane by regulated exocytosis.” This reviewer suggests to control whether PrP^c is released under these conditions. We have now treated uninfected and infected S7 cells with 90 mM KCl for 5 minutes, a treatment that leads to rapid depolarisation and determined the amount of plasma membrane PrP^c using the anti-PrP antibody 8H4 following depolarisation (Fig. 4K). While no significant changes in fluorescence intensities were observed when uninfected S7 cells were subjected to KCl-evoked depolarization, the same treatment increased fluorescence in infected cells by 1.7 fold, compared to mock-treated (PBS) controls. This confirms that PrP^d is released by regulated exocytosis, while PrP^c is unlikely to reach the plasma membrane by the regulated exocytosis pathway.

Changes to manuscript (added content is underlined)

Line 294: “.....confirming that PrP^d reaches the plasma membrane by regulated exocytosis. In contrast, in uninfected cells, KCl-evoked depolarisation did not lead to an increase in PrP^c levels at the plasma membrane (Fig. 4K).”

Lines 1312-15, Figure legend Fig. 4: (K) Quantitative changes in surface PrP levels after KCl-evoked depolarisation in uninfected S7 versus prion-infected iS7 cells. Following fixation, cells were stained with anti-PrP antibody 8H4. At least 100 images were analysed per condition. Data represent mean values ± SEM. Statistical significance was evaluated by Student’s T-test (p < 0.01).

Point 10

Reviewer #2: Fig.3D: the right Y-axis is labeled as “Relative toxicity”. It is unclear why/how relative toxicity decreases with increasing concentration of mβCD, and also in manuscript it is stated that toxicity is lower below 1 mM mβCD, so this very confusing.

Authors’ response: We thank this reviewer for noting that the Y axis in **Fig. 3D** reads “relative toxicity” and we apologise for this error. We now changed the axis to “relative viability”.

Reviewer #3: OK

This reviewer acknowledges that our response to point 8 is satisfactory.

Point 11

Reviewer #2: Lines 209-211: it is assumed (but not indicated in the manuscript) that the quantification of colocalization of intensities in Fig.3F is on the plasma membrane since both antibodies 5B2 and 6C11 label PrP^d at the plasma membrane or just outside of it. Authors claim that fluorescence intensity of colocalization is low there in the presence of 0.4 mM m β CD, but this is not apparent in the images provided on Fig.3F, which show strong colocalization of PrP^d in both 5B2 and 6C11 channels.

Author's response: We thank this reviewer for pointing out that more technical detail would benefit Fig. 3F. For clarity, we now added to the figure legend that “total fluorescence above threshold” was determined. In addition, we have replaced the term “colocalisation” in the legend of Fig. 3D with “intersection”, the technical term for colabelling used in image quantification software and specified in the Methods paragraph *Quantifying relative PrP^d fluorescence intensities*. Intersection thus denotes colabelled areas where pixels are detected in more than one fluorescence channel” to distinguish the term “intersection” from “colocalisation” which is based on a distinct algorithm as explained in said Methods section. We do not agree with the reviewer that there are inconsistencies in fluorescence levels in Fig. 3E, F. Notably, the drop in 5B2 fluorescence and 5B2/6D11 colabelling shown in Fig. 3F is in agreement with reduced colocalisation in m β CD-treated prion-infected cells, compared to mock-treated cells (Fig. 3E). For clarity, we added to Fig. 3F legend “data normalised to vehicle control (1.0)”.

Reviewer #3: OK

We thank this reviewer for acknowledging that our response to point 11 is satisfactory.

Point 12

Reviewer #2: Lines 211-215: “In summary, the lowering of cellular cholesterol levels suggests (i) that intra- and extracellular pools of PrP^d can be uncoupled and (ii) that blocking of PrP^d transport to the plasma membrane inhibits the formation of FL-PrP^d at the plasma membrane.” Cholesterol does not block transport of vesicles to the plasma membrane, so this is incorrect.

Authors' response: We here refer to amendments of the manuscript in response to Reviewer #1 in paragraph 3 above where we fully agree that the perturbation of lipid rafts is an alternative explanation for our results in Fig. 3E, F. This was added in results, lines 250-252.

Reviewer #3: The authors do not address Reviewer #2 remark: Cholesterol does not block transport of vesicles to the plasma membrane, so this is incorrect.” In fact, how can we accept such a conclusion if m β CD is so pleiotropic and non-specific, especially if used for protracted periods of time such as 16h? There are plenty of references about cholesterol depletion decreasing endocytosis and many other processes. Furthermore, have the authors examined the distribution of synapsin 1 under these conditions? *In my opinion, these caveats must be at least mentioned and discussed, in response to Reviewer #2's question.*

We are grateful for the comments of reviewer #3 in regards to point 12 and have now corrected a semantic error in the authors' conclusion in line 262 (added content is underlined): “In summary, the drop of 6D11/5B2 colabelling following the lowering of cellular cholesterol levels suggests (i) that intra- and extracellular pools of PrP^d can be

uncoupled and (ii) that blocking of PrP^d transport to the plasma membrane inhibits the formation of FL-PrP^d at the plasma membrane.”

This reviewer further points to the “pleiotropic and non-specific” effects of mβCD. We would like to politely note that, in response to reviewers’ comments, we diligently examined plasma membrane PrP^c levels to exclude the possibility of PrP^c extraction which can be excluded (Fig. 4G). We further would like to point out that under our experimental conditions (mβCD concentrations < 1mM; 16h incubation) cholesterol was depleted by less than 20% (Fig. 4D, black squares), which is by definition “mild”. Briefly, “mild cholesterol depletion” is broadly defined as a 25-30% reduction of free cholesterol⁴⁻⁶. Notably, under mild cholesterol depletion, we observed a >95% block of 5B2/6D11 colabelling at the plasma membrane at a concentration as low as 0.4 mM mβCD, when compared to mock-treated cells. We therefore object to the reviewer’s suggestion that the observed effects are non-specific.

Reviewer #3 further suggests that the distribution of synapsin 1 following mβCD would be helpful in this context. We disagree with the reviewer on this point, since changes in the amount of synaptic vesicles are observed within seconds and not within the time frame of this experiment, i.e. 16 hours.

Point 13

Reviewer #2: Fig. 4A-C, and 4E-F does not convincingly show colocalization of PrP^d with any of the vesicular markers shown. There is no quantitation nor colocalization analysis provided and by eye, colocalization is not at all clear. The lack of colocalization is also very clear on Fig.4F. Remarkably, Fig.3B is shown no less than 3X in the manuscript (also as Fig.4B, and Fig. S4G). Quantitation by Pearson coefficient is provided after manuscript description in Fig.4B and S4G, and yet the quantitation of colocalization by Pearson is not obvious as no colocalization that is apparent in the image.

Authors’ response: This reviewer comments that there is no convincing evidence for colocalisation in **Fig. 4** and that “by eye, colocalisation is not at all clear”. Analysis of colocalisation by Pearson’s correlation coefficients has already been addressed in Point 9 and we reiterate that we disagree with the reviewer’s opinion and would like to point out that Pearson’s correlation coefficients in this study are based on average values of 20-40 images, while the images displayed in **Fig. 4** are single representative images. We therefore consider the argument of “by eye inspection” misleading as statistically evaluated data is presented (see **Fig. 4G**).

Reviewer #3: Assuming that there are no mistakes in how the statistics were computed, then I would tend to accept the authors explanation.

However, by looking at Fig. 4 and Fig. S5, it seems that the PrP^d distribution changes in each image to fit the distribution of the particular vesicular marker. For example, syn 1 and lamp1. Is this because the focal plane was chosen to get a clear image of the chosen marker? This would make sense, but the question is then: when calculating correlation coefficients, did you include whole z-stacks? From Methods L635 paragraph, this does not appear to be the case.

I am thus not convinced that the Reviewer’s question was well addressed. In my opinion, then, these questions should be addressed in the manuscript and the method used well

explained.

Firstly, we would like to note that the same imaging analysis protocol was used for all organelle markers investigated, irrespective of the location of the organelle in question. Briefly, stacks of 10-15 images in z-direction (0.5 μm per step) were analysed in respect to levels of colocalisation between PrP^d and organelle markers using imaging software Volocity. It is also important to note that the intracellular distribution of PrP^d is heterogeneous and can vary among cells of the same cell type. We have now explained the mode of analysis in Methods in more detail as specified below.

Changes to manuscript (added content is underlined)

Line 714, Methods: “To determine levels of colocalisation of 6D11-positive PrP^d with intracellular markers at perinuclear sites, stacks of 10-15 images in z-direction above the basement membrane were recorded with a step size of 0.5 μm and levels of colocalisation between PrP^d and organelle markers analysed using imaging software Volocity. Data of at least 20 single cells were analysed using the Volocity cropping function. Cells were double-labelled with primaries for 24 hours and washed with PBS.....”

Point 14

Reviewer #2: Fig.4F: authors claim that Synapsin 1 is at the ECM. This is very confusing. How can a synapsin 1 vesicle be found extracellularly at the ECM?

Authors' response: Here the reviewer wonders how Syn1 can be found extracellularly at the ECM. For clarification, we refer to recent publications 17, 18. Please note that this image has been removed to accommodate for additional Fig. 4H-K.

Reviewer #3: I am puzzled by the “brush off” response of the authors here.

Publication 17 describes an improved method to extract PrP^{Sc} using detergents and proteases, and ref 18 deals with the structure of PrP fibrils obtained using this method. They do not have anything with the suggested presence of Syn1 vesicles or PrP^d in the ECM of living cells, which was what the Reviewer wondered about. The authors basically sends the reviewer to these publication, and then removes the disputed panel and its legend, without dealing with the question at hand. In my opinion, we are not merely dealing with a manuscript, but also with the Science. At the very least, the Authors should answer Reviewer #2 his/her legitimate question.

Furthermore, Fig. 4F, which was supposedly removed in the new version, is still quoted in L255: “Notably, at the ECM, 5B2-positive FL-PrP^d fibrils are decorated with Syn1 (Fig. 4F).” I wonder what the authors' position is with the ECM/Syn1 vesicle issue.

Can the authors thus please state if they believe that there are Syn1 vesicles in the ECM, and how do “recent publication 17, 18” show that?

In Point 14, reviewer #2 noted that it is “very confusing” that synapsin 1 is found at the ECM. In response, the authors provided two representative references that provide evidence that synapsin 1 is released from cells into the ECM via exosomes. Here, a principal misunderstanding arose, when reviewer #3 noted that references 17 and 18 “do not have anything with the suggested presence of Syn1 vesicles or PrP^d in the ECM of living cells, which was what the reviewer wondered about.” We politely note that the cited references can be found in the references of the rebuttal document (**17:** Schiera et al., Int J Mol Sci 21, (2019); **18:** Xia et al., Ageing Res Rev 74, 101558 (2022)), not in the manuscript (please note that references from the original rebuttal document were not

copied into this document during review). We therefore hope that reviewer #3 agrees that we did not mean to “brush off” the comment of reviewer #2, but instead addressed this reviewer’s comment in an accommodative manner. Since we added a new panel of images including analysis in Fig. 4 to report the effects of KCl-evoked depolarisation, a result that is pivotal to understand how growing PrP^d fibril at the plasma membrane are associated with truncated intracellular PrP^d aggregates, we removed Syn1-decorated fibrils due to space restrictions.

We thank this reviewer for spotting that the content relating to Fig. 4F was not deleted in the main body of the manuscript. We thus deleted this phrase as specified below.

Changes in manuscript:

Lines 282-4 were removed: ~~Colocalisation of PrP^d with Syn1 is frequently observed at cell junctions as evident by SIM (Fig. 4E). Notably, at the ECM, 5B2-positive FL-PrP^d fibrils are decorated with Syn1 (Fig. 4F).~~

Point 15

Reviewer #2: Lines 240-249: it is unclear if authors are referring to lower steady state intracellular PrP^d levels or total after treatment with CME inhibitors (dynasore and dynole). Treatment with these compounds would be expected to lower intracellular PrP^d, as they block endocytosis. But applied PrP^d should still be found at the cell surface (plasma membrane) as treatment with dynasore or dynole should not inhibit binding of PrP^d to the cell surface. These observations contradict observations in the subsequent paragraph (lines 266-268) where authors report that downregulation of clathrin didn’t lead to reductions in PrP^d steady state levels (again, unclear whether intracellular or also at the cell surface).

Authors’ response: The reviewer notes that “*it is unclear if authors are referring to lower steady state intracellular PrP^d levels or total after treatment with CME inhibitors (dynasore and dynole). Treatment with these compounds would be expected to lower intracellular PrP^d, as they block endocytosis.*” The points that this reviewer raises here are incorrect, since prion steady state levels in this manuscript were determined by “Scrapie cell assay” which monitors the infectious titre of a cell population at steady state, but does not provide information about intracellular or plasma membrane PrP^d levels. Incubation of persistently infected cells with Dynasore led to a 90% reduction of prion titres within 3 days (see Methods for details).

The reviewer furthermore points out that the inhibitory effects of the “CME inhibitors” dynasore and dynole contradict that clathrin inhibition has no effect on prion levels. We find this conclusion concerning, given unequivocal evidence of clathrin-independent endocytosis (CIE) pathways during the past two decades (for recent reviews see 19-22). In fact, the strong inhibitory effect of the dynamin inhibitor dynasore (**Fig. 5A**) and the failure of Pitstop2, an inhibitor of clathrin suggest a role in CIE and this is further supported in **Fig. 5E**, where expression of dynamins and clathrin are directly targeted by transcriptional silencing. We here note that knowledge of CIE is highly relevant to understand the conclusions drawn in this manuscript in regards to gene perturbation effects on prion steady state levels (**Fig. 5E**) and inhibition of prion infection (**Fig. 6G**), respectively.

Reviewer #3: Authors appear to scorn the Reviewer “We find this conclusion concerning, given unequivocal evidence of clathrin-independent endocytosis (CIE)

pathways during the past two decades”, but is that warranted?

I would like to suggest that the authors explain to the reader, in the paper, what are these CIE pathways, and at least speculate which of them (there are many) can potentially fit raft associated fibrils, etc.

In point 15, reviewer #2 addresses the lowering of prion steady state levels following incubation of infected cells with two dynamin inhibitors, dynasore and dynole (Fig. 5A), incorrectly terms both inhibitors CME (clathrin-mediated endocytosis) inhibitors, while further concluding that these results “*contradict observations in the subsequent paragraph (lines 334-5) where authors report that downregulation of clathrin didn’t lead to reductions in PrP^d steady state levels*”. In our response, we fully explain the misinterpretation of these results by reviewer #2 and refer to CIE which was explicitly introduced in this manuscript: Paragraph 318-44, starting with “*To corroborate a role of CIE pathways in the maintenance of cellular prion steady state levels, we next targeted key regulators of endocytic pathways using RNA interference*”. Lines 334-5: “*Despite a 98% knockdown of gene expression, clathrin a (Clta) loss-of-function did not affect prion steady state levels (Fig. 5E)*”. Lines 338-40: “*A highly efficient knockdown of cell division cycle 42 (Cdc42) by 97% led to a 50% drop of cellular prion titres, suggested that Cdc42-mediated CIE plays an important role in maintaining prion steady state levels.*” Line 469-70: “*While PrP^c is believed to be internalised via CME³³, we provide evidence that PrP^d reuptake is mediated by CIE.*”

We thank reviewer #3 for suggesting to introduce CIE pathways in the manuscript and we have now added a short introduction and references as specified below (Lines 299-302). While we unanimously share this reviewer’s view of the importance of lipid rafts in this context, we prefer to refrain from his/her suggestion to “speculate which of them [CIE pathways] can potentially fit raft associated fibrils”, since more than 10 CIE pathways have been characterized in neuronal and non-neuronal cells to date.

Changes to manuscript (added content underlined)

Lines 303-6: While CME is considered the main endocytic pathway for “housekeeping functions” in cells, a growing number of fast non-canonical pathways, classified as CIE has been characterised and include Cdc42-, RhoA-, Arf6-regulated pathways^{36,37}.

General comments

In paragraph 2, this reviewer notes “*there are major deficiencies in the manuscript. One of the main ones is the heavy or almost sole reliance on imaging data throughout the manuscript and lack of use of very well established biochemical assays that are routinely used for studying prion conversion.*”

We strongly disagree with this reviewer’s claim of “*heavy or sole reliance on imaging data throughout the manuscript*” and would like to point out that this study is based on correlative imaging and prion titre output methodology. In fact, this is the first study that addresses *de novo* PrP conversion in neuronal cells by monitoring prion levels, rather than relying on surrogate markers of prions, like proteinase K-resistant PrP (PrP^{Sc}) bands on Western blot.

Reviewer #3: about the paragraph above.

It is true that this study uses both immunofluorescence and SCA-based bioassays. In fact, in my opinion, the authors should be lauded for using bioassays in cell biological studies, which have traditionally utilized primarily microscopy and biochemical assays.

However, I agree with Reviewer #2 that in some cases biochemical studies are still absolutely needed to reinforce conclusions inferred from the immunomicroscopy studies. This is particularly evident in studies of Prd trafficking. This was explained in the point-by-point critique.

We agree with this reviewer's comment that biochemical studies are necessary to corroborate conclusions drawn from imaging data. We refer to changes in Fig. S5, where we now examined changes in PrP^{Sc} upon acidification inhibition.

Additional comments from the reviewer #3:

One essential point

1. I am puzzled by the connection made throughout the paper between the cell surface fibrillar structures, amyloid fibrils, and prion "rods".

Question 1: Do the authors actually equate the 5B2 positive fibrillar structures on the cell surface with bona fide amyloid fibrils? If so, on what grounds, and what is the exact type of amyloid fibrils that they have in mind? Are these the well-known prion rods, for instance?

If they indeed conceive the cell surface fibrillar aggregates as amyloid fibrils, then this should be stated very clearly and explicitly in the Introduction, and convincing arguments should be brought forward.

Is that the concept behind the title "... precedes fibrillisation"?

In the absence of a clear statement, we are left with hints in the Intro:

- L52ff: "... early studies of prion-infected mice showed evidence of "prion-amyloid filaments" in the extracellular space beneath the ependyma¹⁶ and recent studies characterised ex vivo isolated prion rods^{17, 18}".

- L55: "A recent study reported the detection of lipid raft-associated amyloid strings and webs of PrP...¹⁴".

However, these references do not prove that the 5B2 fibrils are "amyloidic fibrils" or rods. To the contrary, the rods isolated in refs 17, 18 are classical rods that were formed after proteolysis (so no 5B2) and detergent extraction.

As to ref 14, they do not claim that their cell surface "strings" (that you have renamed fibrils) are amyloid fibrils, but perhaps alignments of 2D PrP^{Sc} structures that do stain with Thioflavin. If their model is right, then the current conclusion (L438) that "... proteolytic processing of FL-PrPd following reuptake provides a chaperon-independent cellular mechanism for fibril fragmentation, a cellular process that is considered critical to maintain prion replication." is unwarranted.

As to rods: In methods, Line 515, "... under these conditions, the build-up of rod-like PrPd aggregates at the plasma membrane is observed ...". Why do you label these aggregates "rod-like"? In the prion field, "rods" is applied to a specific structure that can be studied by EM, etc (see your own refs 17, 18.) Have you determined that these aggregates are "rod-like" in terms of tinctorial properties, Ab labeling, EM structure? If not, then, in my opinion, you must refrain from such claims as they are very confusing.

L643: "To image PrPd rods at the plasma membrane and ECM, cells were grown for extended culture times..."

L1257, "5B2- positive rods"

We thank this reviewer for addressing, in essential point 1 and 2 of additional comments, the phenomenology and terminology of fibrillar structures described in prion-infected cells

in this manuscript and by others. Here, this reviewer also flags inconsistencies in terminology in this manuscript and we gratefully welcome these comments for clarity. In response to the additional comments, we first address the terminology of fibril-like PrP^d aggregates, then elaborate on the phenomenology and finally address remaining questions.

In additional comments, reviewer #3 requests more information on fibril-like aggregates, reported in this manuscript on the context of previous work in the field. Notably, this reviewer's summary of the state of research on fibrils includes a multitude of terms which may – or may not – describe the same phenomenon, including “rods”, “strings”, “webs”, “fibrils”, “amyloid”, “classical rods”, “amyloid fibrils”, “rod-like” and “fibril-like” aggregates. Prior to drafting this manuscript, we discussed the phenomenology of prion-infected cells with experts in structural biology from the prion and Alzheimer's field and the term “fibril” or “fibril-like” was suggested unanimously to describe the PrP phenotype on the plasma membrane of prion-infected cells, while the term “rod” was considered less helpful or appropriate. The authors here underscore the importance of using language that is broadly used and understood across research fields in neurodegeneration, given the importance of prion-like phenomena. While following the recommendation of peers, we used the term “rod” only in reference to work by Wenborn et al. (2015, reference 17) and Terry et al. (2016, reference 18) (see line 54). As correctly pointed out by this reviewer, there are inconsistencies in the terminology in this manuscript (lines 55, 549, 693 and 1464) and we have thus replaced “rod” with “fibril” for four cases where the term “rod” was not associated with the above mentioned manuscripts. To address the important point of this reviewer on experimental data that justify the term “fibril-like PrP^d aggregates”, we elaborate on outcomes of our study below.

In this study, fibrils are biochemically defined by virtue of their resistance to denaturing agents

That most anti-PrP antibodies recognise PrP^d aggregates only under denaturing condition has been broadly reported and is confirmed in this study (Fig. 2D, E, G). However, we report the surprising finding that epitopes of the structured PrP domain in PrP^d fibrils remain cryptic upon denaturation. We show this at subdiffraction resolution in Fig. 2C and summarise the result in lines 140ff: *“Notably, extended segments of PrP^d fibrils did not show co-labelling under denaturing conditions. Instead, 5B2 immuno-positivity prevailed with 6D11-positive puncta along the lengths and at ends of fibrils. Where 6D11 puncta were detected in close proximity to 5B2-positive fibrils, areas of colocalisation were apparent (see inserts in Fig. 2C).”* Resistance to denaturation is therefore a distinctive biochemical feature of PrP^d fibrils which is traceable by imaging, but not by protein separation methods.

6D11-positive intracellular PrP^d are truncated and amorphous

For evidence of PrP^d truncation at intracellular sites, we here refer to our response to reviewer #2 (point 5) and additional experiments in response to reviewer #3 (Fig. S5). As summarized in the schematic of Fig. 2G, all *“mAbs that bind to the “core” PrP region (90-231) failed to detect PrP^d in prion-infected cells under native conditions”* (lines 162-3). However, PrP^d aggregates at intracellular sites become exposed under denaturing conditions. Leading on from our summary in both paragraphs above, it follows that the aggregation state of PrP^d aggregates at the plasma membrane and at intracellular sites is distinct. In discussion, we refer to data from other studies that postulate that fibril formation is dependent on the PrP N-terminus (for references, see Zahn, 2003; Frankenfield et al., 2005 and Trevitt et al., 2014 in manuscript). Line 462, discussion: *“Notably, N-terminally*

truncated TR-PrP^d at perinuclear sites fails to form amyloidogenic aggregates which suggests that the PrP N-terminus might be necessary for fibril formation. This is in agreement with the notion that the N-terminus mediates higher-order aggregation processes (Zahn, 2003 #8226; Frankenfield, 2005 #10242; Trevitt, 2014 #20341)." We therefore think that TR-PrP^d represents amorphous aggregates.

While the two paragraphs above may clarify most of the comments and questions raised by reviewer #3, additional points raised by this reviewer will be addressed below; questions are addressed in chronological order and labelled 1 to 5.

Q1: "Do the authors actually equate the 5B2 positive fibrillar structures on the cell surface with bona fide amyloid fibrils?" Yes, as explained in the paragraph above "In this study, fibrils are biochemically defined by virtue of their resistance to denaturing agents."

Q2: "If so, on what grounds, and what is the exact type of amyloid fibrils that they have in mind?" It goes without saying that ultrastructural detail is required to further assess the protofilament structure and this cannot be achieved by light microscopy. Other groups have made substantial progress in elucidating the structure of fibrils and strain-specific differences, but this is beyond the scope of this manuscript.

Q3: Is that the concept behind the title".....precedes fibrillisation?"

Yes. What is eluded by the title refers to a key finding of this study. While *de novo* PrP conversion occurs within minutes, PrP^d fibrils at the plasma membrane, detected by the discriminatory anti-PrP mAb 5B2 do not appear in freshly infected cells until 24 hours after infection.

See line 455: "proteolytically truncated TR-PrP^d aggregates reside at perinuclear sites and segregate into the regulated exocytotic pathway, where they dock and fuse with the plasma membrane, thus providing evidence for a directional transport of PrP^d seeds...."

This reviewer further notes that "if their model is right, then the current conclusion that "... proteolytic processing of FL-PrP^d following reuptake provides a chaperon-independent cellular mechanism for fibril fragmentation, a cellular process that is considered critical to maintain prion replication." is unwarranted." In agreement with the reviewers' comment in regards to the hypothetical nature of the conclusion in line 473, we deleted the term "provides a chaperon-independent cellular mechanism for" (see below). This leaves open the underlying mode by which PrP^d fibrils are processed following uptake (see Fig. S5E, F).

Changes to manuscript (added content is underlined):

Line 467-70: In keeping with the observation that FL-PrP^d is absent intracellularly, unless cells are treated with BafA1, proteolytic processing of FL-PrP^d following reuptake may result in ~~provides a chaperon-independent cellular mechanism for~~ fibril fragmentation, a cellular process that is considered critical to maintain prion replication.

Q4: "By contrast, do the authors assume that perinuclear PrP^d is NOT amyloidic?"

Yes. Given that PrP^d fibrils are full-length and are only detected at the plasma membrane, we assume that intracellular, truncated PrP^d aggregates are amorphous as explained in line 465: "Owing to the lack of amyloidogenic properties, TR-PrP^d may represent amorphous aggregates or oligomers."

It should be stressed in this context that amyloid fibrils and amorphous aggregates are not diametrically opposed phenomena. It is broadly assumed that during maturation, the transition from amorphous to fibrillar aggregates, protofibrils that are associated with toxicity and/or seeded aggregation are formed. It is important to note that the precise

mechanisms and intermediates involved in maturation have not been elucidated.

Q5: “Have you determined that these aggregates are “rod-like” in terms of tinctorial properties, Ab labeling, EM structure?”

While Thioflavin T (ThT) and Nile red are broadly used to detect protein fibrils under cell-free conditions, ECM-embedded PrP^d fibrils did not show a distinct label in our hands. Instead, ThT intensely labelled cells in an unspecific manner. The authors are not aware of studies that successfully used amyloid-binding dyes in the context of the cell. We refer to our response in Q2 in regards to the ultrastructure of PrP^d fibrils.

Minor points and typos:

L158 (now L162): Citation 22: wrong reference, before the discovery of PrP. Reference Prusiner (Science 216, 136-144 (1982)) replaced with Serban D, Taraboulos A, DeArmond SJ, Prusiner SB. Rapid detection of Creutzfeldt-Jakob disease and scrapie prion proteins. Neurology 40, 110-117 (1990) – see L162, reference 22.

L238: “... under the experimental conditions of Figure 3EF, we... “. Is it Fig. E and F? Also, by “experimental conditions” do you mean m β CD for 16h? Please state this. It’s not in the Fig. legend either. We thank the reviewer for pointing this out. This incubation time is now specified in the legend of Fig. 3G (Line 1288).

L245: “PrP^d is segregated into vesicles of the regulated secretory pathway and colocalises with markers of the endosomal pathway”. Same title for Fig. 4 legend L1299. This title is confusing. Do you mean “in part” for both localizations, which are supposedly mutually exclusive? We are grateful for this suggestion. We have now deleted “and colocalises with markers of the endocytic pathway” in lines 274 and 1264.

L259: “secretory vesicles (Syp, Vamp4, Syn1, Scg2, Chga, Col4) and endosomal/lysosomal 259 markers (Eea1, Lamp1) (Fig. 4G, H).” No, panel H is a KCl experiment now. Thank you. We have amended this now (Line 288).

L263: “we triggered membrane depolarisation with potassium chloride (KCl).” How long the treatment with KCl? No indication in Fig. 4 legend either, nor in Methods. Please specify. Thank you. We have now specified in the legend of Fig. 4H, I that cells were fixed at 5 minutes after KCl (Line 1309).

L266: “...confirming that FT-PrP^d reaches the plasma...” What is FT-PrP^d? Is that supposed to be FL-PrP^d, or TR-PrP^d? Thank you. FT is a typo and has now been deleted (see line 294).

L1081: “(A-D) Prion-infected S7 cells were co-labelled with 6D11 and vesicular markers”. This is just an example, out of many throughout the ms, of how the reader cannot determine what type of cells were used: persistently infected? Acutely infected? In the latter case, what is the timing of infection? Line 1296: We now added “persistently” to the legend of Fig. 4A-D. We further added in Methods (line 579ff): “If not otherwise specified, persistently prion-infected S7 cells were used in this manuscript. Prion-permissive S7 or myc-tagged S7 cells were freshly infected in Figs. 5F-J, 6C-F and S7 A-D”.

L608 and following: From the description of the method it is not clear if, and how, the cell lysates were normalized for cell #, or protein concentration, etc, prior to cholesterol determination using the Amplex red kit. Please detail the procedure.

Thus, In Fig. 3D, the “relative cholesterol content” – is that normalized to cellular protein content? Or otherwise normalized? While cholesterol levels were not normalized for cell numbers, please note that the actual experiment, i.e. the study of effects of cholesterol lowering on 6D11/5B2 colabelling, was conducted under conditions in the non-toxic range of the experiment (<1 mM m β CD). In absence of toxicity, the protein content between wells within the expected experimental variance. This information was now added to the

Methods section below.

L663 (added content underlined): While cholesterol levels were not normalised for cell numbers in Fig. 3D, the effects of cholesterol lowering on the level of colocalisation between 5B2 and 6D11 (Fig. 3E, F) were determined for non-toxic m β CD concentrations (< 1 mM) only.

L 1074: "...PrPc extraction from the plasma membrane as specified in Methods." Please specify which paragraph in "Methods". This reviewer, for one, was unable to ascertain this detail. The relevant Methods section "Quantifying relative PrP^d fluorescence intensities" was now added to the legend of Fig. 3G.

L612: Fig. X? We apologise for this error; this is now changed to Fig. 3D (Line 664).

L531: What is "ribolysed"? Is this term used to describe the use of a Ribolyzer grinding instrument? Please mention that. How is it performed? In what buffer? This is crucial as it pertains to how cells are infected in each experiment. In L539, what is "clarified" brain homogenate? Please specify if all infections were performed using "ribolyzed" brains, or clarified homogenates, or whether there were exceptions? In Methods, we have now specified how we improve the homogeneity of brain homogenates (Line 568; added content is underlined) "Crude brain homogenates (10% w/v) were prepared essentially as described in Hill et al. ⁶⁴. To improve homogeneity, samples were ribolysed as reported previously⁶⁵. Briefly, crude brain homogenates were transferred to 2 ml microtubes (Sarstedt Ltd., Leicester, UK), containing zirconium beads, Protease Inhibitor Cocktail Set I (Pierce, Leicester, UK) and 25 U benzonase (Novagen, Madison, WI) using a Ribolyser (Hybaid, Cambridge, UK) at maximum speed for two cycles of 45 seconds."

Clarified brain homogenates were used in Fig. 5F-I only and the procedure was described in Methods section "Minimum contact time for productive infection". A link to the Methods section was now added to the legend of Fig. 5F (Line 1334-5).

L625: Is there really 10% Pen/Strep in the blocking step? Yes, imaging of chamber slides is not under aseptic conditions. Further, chamber slides are kept in a cold room under non-sterile conditions. During long-term storage, lower antibiotic concentrations failed to prevent bacterial contamination.

Fig. S1C: inset is misplaced. We thank this reviewer for the thorough perusal. We have now adjusted the position of the inset.

L122 and L1025, 1028, and more: The use of the terms "neuronal" and "neural" is incorrect. See for instance the Thermofisher explanation of these terms: "There are several types of cells in the nervous system including neurons, astrocytes, and microglia. The term "neural" refers to any type of nerve cell, including a mixture of brain cells, whereas "neuronal" is specifically related to neurons. The term "neural" is used in context of both primary cultures and neural stem cells."

(<https://www.thermofisher.com/il/en/home/life-science/cell-culture/primary-cell-culture/neuronal-cell-culture.html>).

We respectfully disagree that primary cultures of neurons should be termed "primary neural", instead of "primary neuronal". While we fully agree with reviewer #3 that such mixed cultures contain non-neuronal cells, including astrocytes, oligodendrocytes and microglia, neurons outweigh the number of non-neuronal cells. This may be the reason why such cultures are more commonly termed "primary neuronal cells". In fact, in manuscript abstracts, the term "primary neuronal cultures" exceeds the term "primary neural cultures" by a factor 30 (1052 versus 34).

Fig. 2F: Arrows not explained/mentioned in the text or Fig. legend. Thank you. Arrows now mentioned in legend (Lines 1261-2): “Arrows denote intracellular 5B2-positive PrP^d aggregates.”

Fig. 3A: the arrows are not mentioned in the text nor in the legend. The arrow in the original magnification is not followed in the inserts a, b. Do the right panels show another cell? Arrows in original magnification do not point at anything I can see. We apologise for this mistake. We have now explained the arrows in Fig. 3A. We have furthermore added an arrow in the original magnification that is followed in inserts a and b. Added content in Fig. 3A underlines: “Arrows denote punctate 6D11-positive PrP^d in juxtaposition with 5B2-positive PrP^d fibrils.”

Can you please mention the estimated lateral resolution with the SIM attachment? We have now added to the legend of Fig. 3A (Line 1272-3): “Estimated lateral resolution SIM: 100 nM.”

L1061: re panel B. “A schematic for additive colour mixing was added next to the merged image to depict colocalisation patterns.” I don’t see the schematic in the figure. Also, scale bars in B? We have now deleted the sentence “A schematic for additive colour mixing was added next to the merged image to depict colocalisation patterns” (Line 1274-5), since this schematic was removed in the submitted version of the figure. We thank the reviewer for spotting this. We have now added a scale bar in Fig. 3B.

Why do we need this panel here? There is almost no co-localization data to be gleaned from it. Is it SIM as well? Can you show a proper magnification (you did that anyway in Fig. 4). For clarity, Fig. 3 provides evidence for a role of vesicular trafficking in the formation of PrP^d fibrils at the plasma membrane from distinct information. First, the observation of 6D11-positive puncta on the cytosolic side of fibrillar PrP^d (Fig. 3A); second, evidence for overlapping distribution of 6D11 (cytosolic side) and 5B2 (plasma membrane) with Syn1-positive vesicles (Fig. 3B), evidence for 6D11-positive puncta proximal to and integral to integrin b1 positive plasma membranes and finally, inhibition of fibril formation at the plasma membrane under conditions of cholesterol depletion (Fig. 3D-G). I hope this helps to clarify that Fig. 3 combines distinct aspects of the role of vesicular trafficking where colocalisation of Syn1 with both anti-PrP antibodies, 5B2 and 6D11 is critical.

L1027 specify “at 3 wks post infection” Thank you, this information was now added to the legend (added content underlined): Line 1239: Three weeks after infection, cells were ~~Fixed cells were~~ and double-labelled with 5B2 and anti-Map2.

L126 Panel I is not at all explained in the text. In the text and in the legend, how long after the infection? This comment has been dealt with on page 7 of this document in response to the comment of this reviewer. “However, the new panel I is not explained in the text.”

Fig. S5: If the text is followed, then “Asynchronous changes in intra- and extracellular PrP^d pools after cell dissociation” should be Fig. S5, not S6. This is correct and we are sorry for this mistake. We now have swapped Figs S5 and S6 and changed the figure numbers in the main body of the manuscript accordingly.

L 1083: E: all these cells were triple labeled, it’s just that in b, c, you just show two colours. Thank you for noting. The description of the legend in Fig. 4E was changed to (Line 1298):

“SIM image of plasma membrane, triple-labelled (a) or double-labelled (b, c) with Syn1, 6D11 and 5B2. Magnified areas (a-c) are denoted by a dashed box in the first image.”

L1087: Levels of colocalization . specify what: 6D11 vs vesicular markers? Thank you.
Amended as suggested.

Changes to manuscript below (added content underlined):

L1302, Fig. 4 (*HG*) *Levels of colocalisation between 6D11 and organelle/vesicular markers, expressed as Person correlation coefficients. For Representative images and gene names for the abbreviations, ~~are shown in~~ we refer to Fig. S5S6.*

L1087: Pearson *Incomplete sentence.*

L276: “Two pharmacological dynamin inhibitors, dynasore and dynole showed a strong dose-dependent decrease of cellular prion steady state levels by as much as 90% over three days (Fig. 5A). Perhaps specify: ... decrease of infectious cell associated prion ... as determined by SCA”? We thank reviewer #3 for this suggestion.(added content is underlined) L309: dose-dependent decrease of cellular prion steady state levels by as much as 90% over three days, as determined by Scrapie Cell assay (SCA, -Fig. 5A).

Fig. 5 legend (L1100) and tables S3 and S4: please include the time length of treatment with the inhibitors. Thank you for the suggestion. The length of the experiment, i.e. a 3-day incubation with inhibitors was added in Fig. 5 legend (L1321) and Table S3 (L1615). Note that the time frame of inhibitor treatment was specified in Table S4 (L1624).

L1356: Table S3: Here and elsewhere, can you please indicate which of the two medium change regimes (Methods, L510 ff) was used?

We thank reviewer #3 for this suggestion. We have now termed the two medium change regimes “extended TC protocol” and “standard TC protocol”, specified the corresponding Figures in the methods section and in the legends of respective figures referred to the TC protocol type in Methods (added content underlined):

Methods, line 547ff: Extended TC protocol: Where S7 cells were grown for extended culture times (Fig. 1C), the serum concentration was reduced by 50 %, while the culture medium was changed daily from 3 days of culture. Under these conditions, the build-up of α -fibril-like PrP^d aggregates at the plasma membrane is observed from three days onwards. Standard TC protocol: Daily replacement of ½ the conditioned medium with fresh OFCS from 3 days of culture was alternatively used for extended tissue culture times where specified (Figs. 2C and 3A).

Reviewer #4 (Remarks to the Author):

This paper addresses an important and insufficiently explored question in prion biology: where and when are prions generated in the cell? Answering this question is experimentally challenging, even in model cell lines, because of the lack of availability of antibodies or other probes that can distinguish PrPC from PrPSc, making it difficult to track the conversion process kinetically and at subcellular resolution.

This paper attempts to overcome this difficulty using existing PrP antibodies with selective reactivities toward certain conformational and sequence-specific epitopes, as well as an epitope-tagged form of PrPC. A key strategy relies on the use of antibodies against the N-terminal region of PrPC, an approach that builds on the observations of Rouvinski et al. (2014), who first described PrP^{Sc} strings at the plasma membrane of infected cells. The use of myc epitope-tagged PrP derives from earlier work by Goold et al. (2011). By applying these strategies in immunolabeling experiments, the authors identify truncated and full-length forms of PrP^{Sc}, and characterize their cellular localization and growth kinetics. They find that the two pools are connected by exocytic transport of truncated PrP^{Sc} to the surface, and clathrin-independent endocytosis of the full length PrP^{Sc} into the cell.

The work is very rigorous, thorough, and well controlled. The authors have made a serious attempt to address the individual points raised by the previous reviewers, most notably reviewer #2 who had the most extensive criticisms. The authors have undertaken several new experiments, although the manuscript revisions have been relatively modest in scope.

We thank this reviewer for the strong support of our study to further characterise the mode of PrP conversion.

Although it has major strengths, this study also suffers from a significant limitation, which is the exclusive reliance on immunofluorescence detection with a set of antibodies whose selective recognition of PrP^{Sc} is conditional, depending on their relative reactivities under native vs. denaturing conditions, and on their staining of PrP^{Sc} molecules before and after proteolytic truncation of the N-terminus. Thus, the conclusions are, of necessity, somewhat indirect. The addition of another methodology to complement the immunofluorescence approach would help strengthen the arguments the authors are trying to make. For example, isolation and characterization of different forms of cell-associated PrP^{Sc} in conjunction with subcellular fractionation would provide confirmatory evidence, as would pulse-chase labeling to follow the kinetic transformations of the different forms and their trafficking through the cell. Even if some fibrillar PrP^{Sc} remains bound to the substrate (Fig. S6), it should be possible to isolate cell-associated pools of PrP^{Sc} associated with particular organelles. Of course, it would also help to derive new antibodies with more selective reactivities, or to tag the PrP molecules with fluorescent probes that would allow their recognition by real-time imaging.

This reviewer notes that “*Although it has major strengths, this study also suffers from a significant limitation, which is the exclusive reliance on immunofluorescence detection...*” We respectfully disagree with this opinion and point out that this study is based on correlative imaging and prion titre output methodology to investigate the kinetics of *de novo* conversion and formation of truncated and full-length PrP^d species at the plasma membrane. Inclusion of data on *bona fide* prion levels provides access to high-throughput quantitative data for intervention (Figs. 5 and 6). This view is shared by reviewer #3: “*In fact, in my opinion, the authors should be lauded for using bioassays in cell biological studies, which have traditionally utilized primarily microscopy and biochemical assays.*”

In response to the reviewers’ suggestion, we have now conducted IP and Western blot analyses to support the link between intracellular prion gradients and PrP^d truncation.

This reviewer points to several challenges in investigating abnormal PrP species in cells, for instance that “*selective recognition of PrP^{Sc} is conditional, depending on their relative*

reactivities under native vs. denaturing conditions and on their staining of PrP^{Sc} molecules before and after proteolytic truncation of the N-terminus". While we could not agree more with this assessment, we would like to add that the described challenges are inherent to misfolding and processing of PrP molecules and that identification of discriminatory anti-PrP antibodies and replication-compatible modifications of PrP are strategies to overcome these obstacles as shown in this study.

Indeed, the defining advancements of this study are intricately linked to the tools developed or applied here, including binders, prion titre output and imaging methods, resulting in an expansion of our understanding of the prion concept. More precisely, we demonstrate (1) that seed and fibril-forming aggregates are spatially separated and linked via vesicular transport, (2) that prion infection can be effectively blocked at the level of the plasma membrane by the single and combined targeting of dynamins and Cdc42 and (3) that abnormal PrP is segregated into the regulated secretory pathway and reaches the plasma membrane following depolarisation, a process that triggers template-assisted fibril formation at the plasma membrane.

We fully agree that identification of "*new antibodies with more selective reactivities*", in particular disease-specific monoclonal antibodies would greatly advance our understanding of prion conversion and pathogenesis, respectively.

We furthermore agree with reviewer #4 that further advances in other methodologies may expand our findings, like "isolation and characterization of different forms of cell-associated PrP^{Sc} in conjunction with subcellular fractionation" and isolation of "cell-associated pools of PrP^{Sc} associated with particular organelles". In regards to the isolation of fibrillar PrP^d, we refer to the failure of immunoprecipitating ECM-bound PrP^d fibrils (Fig. S5C, D) as well as to our discussion of point #4 on pages 8-10 of this document. Whether there is mileage in isolating truncated PrP^d by subcellular fractionation remains to be determined. Our study advances our knowledge of template-assisted protein aggregation and thus highlights the formation and growth of PrP^d fibrils at the plasma membrane.

Other points:

1. In addition to relying on N-terminal antibodies to follow PrP^{Sc} formation and processing, why don't the authors try monitoring myc-tagged PrP not only at early times after infection, but also at later times, after the cells grow fibrils. This will also allow the authors to unambiguously determine if the fibrillar PrP^{Sc} derives from endogenous PrP^c, or if it results from residual PrP^{Sc} in the inoculum.

We thank this reviewer for suggesting to use myc-tagged PrP at later time points after infection to determine if fibrillar PrP^{Sc} is derived from "endogenous PrP^c, or if it results from residual PrP^{Sc} in the inoculum". However, it turned out that anti-myc is non-discriminatory in regards to PrP^c and PrP^{Sc}, respectively. While PrP^d fibrils at the plasma membrane show higher fluorescence intensities, the plasma membrane is decorated with myc-positive PrP^c. This is why we opted to include the discriminatory anti-PrP antibody 6D11 to identify infected cells (Fig. 6C-E).

2. The paper leaves open the identity of the perinuclear region where truncated PrP^{Sc} accumulates. Co-staining with additional marker proteins would help here.

We thank this reviewer for correctly noting that our study “leaves open the identity of truncated PrP^d at the perinuclear region“. This is an important point and subject to current investigations. Our failure of co-labelling PrP^d with typical marker proteins in the proximity of the Golgi apparatus (not reported in this study) leaves open the question whether biomolecular condensates are formed by virtue of PrP aggregation.

3. As written, the paper will be a bit opaque to the non-prion biologist, since it does not clearly convey the big picture of the results, and fails to engage the reader with an easily comprehensible story line. The use of multiple antibodies makes this especially important.

We respectfully disagree with this reviewer’s view that our manuscript “does not clearly convey the big picture of the results” and refer to a well-structured discussion section of this manuscript. Admittedly, several aspects are subject to ongoing research and we hope that our future work on disease pathways of proteopathic seeds may help to form a framework to identify disease pathways that are common and specific to distinct proteopathic seeds, like A β , tau and α -synuclein.

Minor Points:

4. Figure 1H – The primary astrocytes are infected with 22L and not RML. There is no discussion on strain differences in the paper so it would be best to substitute this for an image of RML infected primary astrocytes to keep this consistent throughout.

We thank this reviewer for the suggestion to keep the used prion strains consistent. We have therefore replaced Fig. 1H, which previously showed 22L-infected primary neuronal cultures with a new image of a primary neuronal culture which was infected with RML or mock (CD1).

Changes to manuscript (added content is underlined):

Line 122: “Primary neuronal cultures from embryonic e17 mouse brains, infected with 22L RML prions showed...”

Figure 1H legend, Line 1232: “Glial fibrillary acidic protein (GFAP)-positive astrocytes in cultures of Primary primary neuronal cultures cells from embryonic e17 mouse brains, infected with α -10⁻⁵ dilutions of the prion strain 22L-RML (10% brain homogenate, w/v) and uninfected CD1 (10% brain homogenate, w/v, mock), triple-labelled with 5B2, 6D11 and anti-GFAP.”

5. Line 519 - Although the paper specifies use of neuronal cultures, the images presented in Figures 1 and 2 are astrocytes (except Figure 1H). Are the astrocytes part of a coculture system or is it a glial culture? This should be clarified to prevent confusion and keep the image legends consistent.

We thank this reviewer for suggesting to clarify whether astrocytes are part of a co-culture system or a glial culture. We have now specified the source of astrocytes and neurons in the Methods section “Primary neuronal cultures”.

Changes to manuscript (added content is underlined):

Line 554: “Primary cortico-hippocampal cultures were prepared from embryonic e17 FVB mouse brains essentially as described previously⁶². These cultures contain GFAP-positive

prion-susceptible astrocytes which, when infected with RML prions, form 5B2-positive PrP^d fibrils (Fig. 1H). To culture neuronal monocultures (Fig. 1I), primary hippocampal cultures were treated with 1 μ M AraC at 6 days after isolation. Following infection with RML prions, PrP^d fibrils were observed after three weeks (Fig. 1I)."

6. The paper bounces around in the naming of the cell cultures as infected S7, iS7, and PK1. It would be best to change all references to S7 or infected S7, instead of using the iS7 designation, and to rename the cells in Figure S6A and S6G as S7 cells, if they are indeed the same thing.

We thank this reviewer for suggesting to omit the abbreviation iS7 and instead to use (prion-) infected S7 cells. We have made this change throughout the manuscript. For clarity, PK1-10 is a specific cell clone used to generate double-knockdown cells (Goold et al., Nat Commun 2, 281 (2011)). These cells have been used to express myc-tagged Prnp chimera. This has been referenced in line 389 below and in Methods (Line 766). Line 389: "Myc-tagged Prnp chimera were expressed in PK1-10 cells where endogenous Prnp is stably silenced with a small hairpin RNA targeting the 3'-UTR of Prnp¹³."

7. References vary in terms of the number of authors mentioned vs. et al.

We apologise for this inconsistency, but we have no control on the output of references. We are using the standard style for Nature Communications in EndNote.

Concluding remarks

This review process was considerably disrupted by the withdrawal of the two initial reviewers from the review process. Despite these difficulties, we have done our utmost to address experimental controls and additional experiments were deemed necessary and helpful. We have also made substantial progress towards the requested biochemical characterization of PrP aggregation during acidification inhibition. In support of our willingness to improve the quality of this manuscript, we have provided 14 additional experiments (with 3-4 independent repetitions per experiment): Figs. 1H, 1I, 3G, 4H-J, 4K, 6G, S1F, S5C,D, S5E,F, S5G-I, S5J-L, S5M-O, S8, Table S5.

References:

1. **Khalili-Shirazi A, et al. Beta-PrP form of human prion protein stimulates production of monoclonal antibodies to epitope 91-110 that recognise native PrP(Sc). *Biochim Biophys Acta* 1774, 1438-1450 (2007).**
2. **Stahl-Meyer J, Holland LKK, Liu B, Maeda K, Jäättelä M. Lysosomal Changes in Mitosis. *Cells* 11, (2022).**
3. **Abu-Remaileh M, et al. Lysosomal metabolomics reveals V-ATPase- and mTOR-dependent regulation of amino acid efflux from lysosomes. *Science* 358, 807-813 (2017).**
4. **Chadda R, Howes MT, Plowman SJ, Hancock JF, Parton RG, Mayor S. Cholesterol-sensitive**

Cdc42 activation regulates actin polymerization for endocytosis via the GEEC pathway. *Traffic* **8, 702-717 (2007).**

- 5. Shapira KE, Ehrlich M, Henis YI. Cholesterol depletion enhances TGF- β Smad signaling by increasing c-Jun expression through a PKR-dependent mechanism. *Mol Biol Cell* **29**, 2494-2507 (2018).**
- 6. Telbisz A, *et al*. Membrane cholesterol selectively modulates the activity of the human ABCG2 multidrug transporter. *Biochim Biophys Acta* **1768**, 2698-2713 (2007).**

REVIEWERS' COMMENTS

Reviewer #3 (Remarks to the Author):

Reviewer #3

"Additional comments":

1. All my 'minor points and typos' seem to have been properly addressed.

2. As to the 'essential point' referring to the fibril/rods terminology (regarding the 5B2 cell surface structures), I have read the authors' response with great interest. That they removed the term 'rods' is great. However, in my opinion, the use of the term 'fibril', and sometimes 'amyloidogenic fibrils', is absolutely confusing because it carries the unwarranted hypothesis that the 5B2 filaments are actually amyloid fibrils.

Concerning points Q1-Q3:

Thus, what makes me uneasy is that the 5B2 fibrils, which are "defined by virtue of their resistance to denaturing agents", are equated by the authors to actual bona fide amyloid fibrils (response to Q1). But is this assumption warranted?

It is true that, for decades, PrPd has been characterized by the recurrent finding that some of its epitopes are natively hidden and are revealed only by various levels of denaturation. But the connection of this property (hidden epitopes revealed by denaturation) with amyloids is not unequivocal. In fact, the formation of 'rods', which are bona fide amyloidic structures based on their tinctorial properties (with Congo red), requires that scrapie brain homogenates be extracted with detergents. This suggests that PrPd may need to be extracted from membranes prior to its forming amyloid structures. But the 5B2 fibrils studied here are (probably) membrane associated. Is there data in the literature showing that membrane-associated PrPd forms amyloids?

The only study I could find in this respect is the Rouvinski 2014 paper. In that study, PrPd also appeared on the cell surface as filamentous filaments or 'strings', and these authors did manage to show that these 'strings' stained with Thioflavin (using isolated plasma membrane sheets). But a closer look using SEM suggested that the filaments were actually composed of a suite of dots containing PrPd. These dots may be local 2D non fibrillar amyloids. How these dots are aligned into filamentous morphology was not determined in that study, but the discontinuous and periodic immunogold staining did not indicate the presence of bona fide amyloid fibrils. Of note, these strings are probably equivalent to the 5B2 fibrils of the current manuscript since they were detected using the same methods.

What's in a word?

In my opinion, then, the fact that the authors equate their 5B2 fibrils to bona fide amyloid fibril (response to Q1) is unwarranted by their own data, or by data in the literature, especially given the findings of Rouvinski (2014). The problem in using the term 'fibrils', especially interchangeably with 'amyloidogenic fibril', is that it leads the reader to assume that the 5B2 fibrils are proven amyloid fibrils, without proper warning or caveats in the text. In these days and times, when amyloid fibrils structure, protofilaments, etc are increasingly understood, using this term to describe the 5B2 filaments conjures up well established structural entities, as the term 'fibrils' is "committed".

I am thus curious as to why the term 'fibril'/'amyloidogenic fibril' was unanimously chosen by the authors and their colleagues. In any case, in my opinion, this state of affairs needs to be discussed in depth in the manuscript, especially if the authors decide to keep the 'fibril' designation.

Regarding point Q4, referring to L462:

"Notably, N-terminally truncated TR-PrPd at perinuclear sites fails to form amyloidogenic aggregates

which suggests that the PrP N-terminus might be necessary for fibril formation. This is in agreement with the notion that the N-terminus mediates higher order aggregation processes. Owing to the lack of amyloidogenic properties, TR-PrPd may represent amorphous aggregates or oligomers."

The authors reply to Q4: "Given that PrPd fibrils are full-length and are only detected at the plasma membrane, we assume that intracellular, truncated PrPd aggregates are amorphous as explained in line 465" appears to be based on the same flawed (in my opinion) equivalence, discussed above, of "fibrils" and "amyloidogenic". The manuscript's argument here is that 1) since one does not see long fibrils in lysosomes, they are not "amyloidogenic", and 2) since the PrPd there is known to be truncated, this may show that only full length PrPd forms fibrils.

1) the use of "amyloidogenic" vs "amyloid" is confusing. When stating that TRPrPd in lysosomes lacks amyloidogenic properties, do you mean that it cannot **generate amyloid**, as opposed to not **being** an amyloid?

How do we know that lysosomes do not contain short fibrils (shorter than the lateral resolutions)?

2) There is no reason to think that TRPrPd cannot form amyloid fibrils. In fact, prion rods are usually formed of strictly truncated PrP27-30.

Reviewer #4 (Remarks to the Author):

The authors have now satisfactorily addressed all of the criticisms raised in my review. They have gone to great lengths to include additional experiments (including biochemical ones), and to thoughtfully address the points raised by all reviewers in an extensive and detailed rebuttal letter. In my view, the paper is ready for acceptance and publication.

Response to reviewers' comments

Reviewer #3 (Remarks to the Author):

Reviewer #3

"Additional comments":

1. All my 'minor points and typos' seem to have been properly addressed.

We thank this reviewer for acknowledging that minor points have been addressed.

2. As to the 'essential point' referring to the fibril/rods terminology (regarding the 5B2 cell surface structures), I have read the authors' response with great interest. That they removed the term 'rods' is great. However, in my opinion, the use of the term 'fibril', and sometimes 'amyloidogenic fibrils', is absolutely confusing because it carries the unwarranted hypothesis that the 5B2 filaments are actually amyloid fibrils.

We thank this reviewer for positively acknowledging that we deleted the term "rods" in this manuscript, if and where it relates to the authors' own data, in response to the reviewer's comment. We respond to this reviewer's concern regarding terminology in the paragraphs below where he/she further elaborates on the context of the previous rebuttal and the manuscript, respectively.

Concerning points Q1-Q3:

Thus, what makes me uneasy is that the 5B2 fibrils, which are "defined by virtue of their resistance to denaturing agents", are equated by the authors to actual bona fide amyloid fibrils (response to Q1). But is this assumption warranted?

It is true that, for decades, PrPd has been characterized by the recurrent finding that some of its epitopes are natively hidden and are revealed only by various levels of denaturation. But the connection of this property (hidden epitopes revealed by denaturation) with amyloids is not unequivocal. In fact, the formation of 'rods', which are bona fide amyloidic structures based on their tinctorial properties (with Congo red), requires that scrapie brain homogenates be extracted with detergents. This suggests that PrPd may need to be extracted from membranes prior to its forming amyloid structures. But the 5B2 fibrils studied here are (probably) membrane associated. Is there data in the literature showing that membrane-associated PrPd forms amyloids?

The concern of this reviewer is the authors' use of the term "amyloid fibrils" and whether this assumption is warranted. The authors observed that fibrils are resistant to denaturing agents, supported by their phenotypic characterisation at subdiffraction resolution (Fig. 2C). While we agree with this reviewer that this property does not equivocally corroborate the amyloid properties of PrP fibrils, we disagree on the experimental approach suggested. On pages 24-26 of the rebuttal, we suggest that "ultrastructural detail is required to further assess the protofilament (or amyloid) structure of fibrils". This reviewer suggests that the "tinctorial" properties of fibrils may be confirmed using direct dyes, like Congo red or Thioflavin T (ThT). However, the lack of specificity of direct dyes poses a considerable concern. A number of reports show that direct dyes, like Congo Red and ThT exhibit false-positive binding to many polymeric substances, including DNA, cellulose proteins and protein aggregates that are not in the amyloid conformation [1-4]. Given these common pitfalls reported for direct dyes, we think that tomographic focused ion beam scanning

electron microscopy (FIB-SEM) and cryo-electron microscopy are more suitable to clarify the amyloid property of fibrils which we hope the reviewers agree are beyond the scope of this manuscript. To address the reviewer's concern, we deleted the term "amyloid" and "amyloidogenic" which was used five times in this manuscript in association with data from this study.

The only study I could find in this respect is the Rouvinski 2014 paper. In that study, PrPd also appeared on the cell surface as filamentous filaments or 'strings', and these authors did manage to show that these 'strings' stained with Thioflavin (using isolated plasma membrane sheets). But a closer look using SEM suggested that the filaments were actually composed of a suite of dots containing PrPd. These dots may be local 2D non fibrillar amyloids. How these dots are aligned into filamentous morphology was not determined in that study, but the discontinuous and periodic immunogold staining did not indicate the presence of bona fide amyloid fibrils. Of note, these strings are probably equivalent to the 5B2 fibrils of the current manuscript since they were detected using the same methods.

The authors are well aware of the Rouvinski et al. (2014) study and have cited this manuscript three times in this study. In line 55-57 of the manuscript, we write: "*A recent study reported the detection of lipid raft-associated amyloid strings and webs of PrP, but the molecular mode of amyloid formation is unknown (Rouvinski et al., 2014).*"

We fully agree with the reasoning of the reviewer that the "strings" described in the Rouvinski et al. (2014) study are "*probably equivalent to the 5B2 fibrils*".

What's in a word?

In my opinion, then, the fact that the authors equate their 5B2 fibrils to bona fide amyloid fibril (response to Q1) is unwarranted by their own data, or by data in the literature, especially given the findings of Rouvinski (2014). The problem in using the term 'fibrils', especially interchangeably with 'amyloidogenic fibril', is that it leads the reader to assume that the 5B2 fibrils are proven amyloid fibrils, without proper warning or caveats in the text. In these days and times, when amyloid fibrils structure, protofilaments, etc are increasingly understood, using this term to describe the 5B2 filaments conjures up well established structural entities, as the term 'fibrils' is "committed".

I am thus curious as to why the term 'fibril'/'amyloidogenic fibril' was unanimously chosen by the authors and their colleagues. In any case, in my opinion, this state of affairs needs to be discussed in depth in the manuscript, especially if the authors decide to keep the 'fibril' designation.

We agree with the reviewer's opinion above and have amended the manuscript as explained above.

In regards to the authors' use of the terms "fibril" and "fibril-like", we would like to refer to the authors' response to this reviewer on page 25 of the rebuttal. For convenience we have copied the relevant paragraph below.

"In additional comments, reviewer #3 requests more information on fibril-like aggregates, reported in this manuscript on the context of previous work in the field. Notably, this reviewer's summary of the state of research on fibrils includes a multitude of terms which may – or may not – describe the same phenomenon, including "rods", "strings", "webs", "fibrils", "amyloid", "classical rods", "amyloid fibrils", "rod-like" and "fibril-like" aggregates. Prior to drafting this manuscript, we discussed the phenomenology of prion-infected cells

with experts in structural biology from the prion and Alzheimer's field and the term "fibril" or "fibril-like" was suggested unanimously to describe the PrP phenotype on the plasma membrane of prion-infected cells, while the term "rod" was considered less helpful or appropriate. The authors here underscore the importance of using language that is broadly used and understood across research fields in neurodegeneration, given the importance of prion-like phenomena. While following the recommendation of peers, we used the term "rod" only in reference to work by Wenborn et al. (2015, reference 17) and Terry et al. (2016, reference 18) (see line 54). As correctly pointed out by this reviewer, there are inconsistencies in the terminology in this manuscript (lines 55, 549, 693 and 1464) and we have thus replaced "rod" with "fibril" for four cases where the term "rod" was not associated with the above mentioned manuscripts. To address the important point of this reviewer on experimental data that justify the term "fibril-like PrPd aggregates", we elaborate on outcomes of our study below".

Regarding point Q4, referring to L462:

"Notably, N-terminally truncated TR-PrPd at perinuclear sites fails to form amyloidogenic aggregates which suggests that the PrP N-terminus might be necessary for fibril formation. This is in agreement with the notion that the N-terminus mediates higher order aggregation processes. Owing to the lack of amyloidogenic properties, TR-PrPd may represent amorphous aggregates or oligomers."

The authors reply to Q4: "Given that PrPd fibrils are full-length and are only detected at the plasma membrane, we assume that intracellular, truncated PrPd aggregates are amorphous as explained in line 465" appears to be based on the same flawed (in my opinion) equivalence, discussed above, of "fibrils" and "amyloidogenic". The manuscript's argument here is that 1) since one does not see long fibrils in lysosomes, they are not "amyloidogenic", and 2) since the PrPd there is known to be truncated, this may show that only full length PrPd forms fibrils.

1) the use of "amyloidogenic" vs "amyloid" is confusing. When stating that TRPrPd in lysosomes lacks amyloidogenic properties, do you mean that it cannot **generate amyloid**, as opposed to not **being** an amyloid?

How do we know that lysosomes do not contain short fibrils (shorter than the lateral resolutions)?

2) There is no reason to think that TRPrPd cannot form amyloid fibrils. In fact, prion rods are usually formed of strictly truncated PrP²⁷⁻³⁰.

In response to the reviewer's concern, we have amended the manuscript by deleting the term "amyloid" which addressed the reviewer's concern, as mentioned above.

In regards to truncated PrP^d (TR-PrP^d) at intracellular sides, this reviewer again takes issue with the authors' use of the term "amyloid" or "amyloidogenic". Given our decision to delete "amyloid" in this manuscript, we consider this concern resolved.

Reviewer #4 (Remarks to the Author):

The authors have now satisfactorily addressed all of the criticisms raised in my review. They have gone to great lengths to include additional experiments (including biochemical ones), and to thoughtfully address the points raised by all reviewers in an extensive and detailed rebuttal letter. In my view, the paper is ready for acceptance and publication.

We thank this reviewer for the very positive and complimentary response, for confirming that the “*authors have now satisfactorily addressed all of the criticisms raised in the review*” and for suggesting “*the paper is ready for acceptance and publication*”.

References:

- [1] A.J. Howie, D.B. Brewer, Optical properties of amyloid stained by Congo red: history and mechanisms, *Micron* 40 (2009) 285–301.
- [2] S.A. Hudson, H. Ecroyd, T.W. Kee, J.A. Carver, The thioflavin T fluorescence assay for amyloid fibril detection can be biased by the presence of exogenous compounds, *FEBS J.* 276 (2009) 5960–5972.
- [3] G.V. De Ferrari, W.D. Mallender, N.C. Inestrosa, T.L. Rosenberry, Thioflavin T is a fluorescent probe of the acetylcholinesterase peripheral site that reveals conformational interactions between the peripheral and acylation sites, *J. Biol. Chem.* 276 (2001) 23282–23287.
- [4] S. Sugimoto, K.I. Arita-Morioka, Y. Mizunoe, K. Yamanaka, T. Ogura, Thioflavin T as a fluorescence probe for monitoring RNA metabolism at molecular and cellular levels, *Nucleic Acids Res.* 43 (2015) 1–12.